# Reviews and syntheses: Expanding the global coverage of gross primary production and net community production measurements using BGC-Argo floats

Robert W. Izett[1], Katja Fennel[1], Adam C. Stoer[1], David. P. Nicholson[2]

[1]Department of Oceanography, Dalhousie University, Halifax, B3H 4R2, Canada
[2]Marine Chemistry and Geochemistry Department, Woods Hole Oceanographic Institution, Woods Hole, 02543, USA

*Correspondence to*: Robert Izett (robert_izett@live.com); Katja Fennel (Katja.Fennel@Dal.Ca)

**Abstract.** This paper provides an overview and demonstration of emerging float-based methods for quantifying gross primary production (GPP) and net community production (NCP) using Biogeochemical-Argo (BGC-Argo) float data. Recent publications have described GPP methods that are based on the detection of diurnal oscillations in upper ocean oxygen or particulate organic carbon concentrations using single profilers or a composite of BGC-Argo floats. NCP methods rely on budget calculations to partition observed tracer variations into physical or biological processes occurring over timescales greater than one day. Presently, multi-year NCP time-series are feasible at near-weekly resolution, using consecutive or simultaneous float deployments at local scales. Results, however, are sensitive to the choice of tracer used in the budget calculations and uncertainties in the budget parametrizations employed across different NCP approaches. Decadal, basin-wide GPP calculations are currently achievable using data compiled from the entire BGC-Argo array, but finer spatial and temporal resolution requires more float deployments to construct diurnal tracer curves. A projected, global BGC-Argo array of 1000 floats should be sufficient to attain annual GPP estimates at 10-degree latitudinal resolution, if floats profile at off-integer intervals (e.g., 5.2 or 10.2 days). Addressing the current limitations of float-based methods should enable enhanced spatial and temporal coverage of marine GPP and NCP measurements, facilitating global-scale determinations of the carbon export potential, training of satellite primary production algorithms, and evaluations of biogeochemical numerical models. This paper aims to facilitate broader uptake of float GPP and NCP methods, as singular or combined tools, by the oceanographic community and to promote their continued development.

## 1 Introduction

Marine primary production (PP), the photosynthetic production of organic carbon and oxygen ($O_2$), is a foundational process for ocean ecosystems. PP sustains marine life, strongly correlates with fisheries yields (e.g., Ware and Thomson, 2005), and influences the planet's climate by contributing to atmospheric carbon dioxide ($CO_2$) sequestration via the biological carbon pump (Volk and Hoffert, 1985; Siegenthaler and Sarmiento, 1993). Climate change is expected to have a heterogeneous, albeit uncertain, effect on the timing, magnitude, and variability of PP across the global ocean (e.g., Polovina et al., 2008; Bopp et

al., 2013; Westberry et al., 2012), with potentially significant impacts on marine food webs and the biological carbon sink (e.g., Hoegh-Guldberg and Bruno, 2010; Ainsworth et al., 2011). To understand and predict these climate-dependent changes with confidence, it is crucial to monitor PP variability on ecologically relevant space and time scales. Autonomous profiling instruments, such as biogeochemical Argo (BGC-Argo) floats, offer great potential to achieve this objective by augmenting traditional PP sampling approaches and enhancing the spatial (horizontal and vertical) and temporal coverage of PP estimates (Chai et al., 2020).

At the ecosystem level, PP can be quantified by the following common metrics: gross primary production (GPP), net primary production (NPP), and net community production (NCP) (Fig. 1). GPP measures community-wide photosynthesis, representing the total production of organic carbon or $O_2$ by autotrophs (e.g., phytoplankton, cyanobacteria) and represents the photosynthetic energy availability to the entire food web. GPP is reported as gross oxygen production (GOP) or gross carbon production (GCP), when defined in $O_2$ or carbon equivalents, respectively. NPP refers to the net production of autotroph biomass when accounting for autotrophic respiration (i.e., organic matter oxidation; AR), and represents the amount of photosynthetically produced organic carbon available to heterotrophs (e.g., bacteria, zooplankton, fish). Lastly, NCP is the difference between GPP and respiration by autotrophs and heterotrophs (i.e., community respiration, CR), and therefore determines if an ocean region is net autotrophic (net production, indicated by NCP > 0) or net heterotrophic (net consumption and NCP < 0). When measured over sufficiently large temporal and spatial scales, NCP quantifies the amount of photosynthetically produced organic matter that is removed from the upper ocean (Laws 1991). GPP, NPP and NCP are often expressed as volumetric equivalents of organic carbon or $O_2$ production (e.g., mol C or $O_2$ $m^{-3}$ $d^{-1}$) and respiration terms are expressed in terms of organic C or $O_2$ consumption. Accordingly, in a closed system, GPP, NPP and CR can only have positive values, while NCP may assume positive or negative quantities.

A variety of approaches and sampling platforms have been used to quantify PP. The earliest method estimates NCP and CR (and thus GOP) by measuring the evolution of $O_2$ in natural seawater samples incubated in light and dark bottles, respectively (Gaarder and Gran, 1927). Other incubation-based approaches involve spiking samples with [14]C- or [13]C-labelled bicarbonate (GPP and NPP; Steeman Nielsen, 1952; Slawyk et al., 1977) or [18]O-labelled water (GOP; Bender et al., 1987; Ferrón et al., 2016) to trace temporal changes in photosynthetic biomass or $O_2$ production under realistic incubation conditions. These incubation approaches, though, are subject to various experimental biases, including containment effects on the plankton community, sensitivity to the incubation duration, and the excretion of labelled dissolved organic carbon (e.g., Pei and Laws, 2013; Cullen, 2001). The $O_2$-to-argon ($O_2$/Ar; Reuer et al., 2007; Spitzer and Jenkins, 1989) and triple $O_2$ isotope (Luz and Barkan, 2000) methods thus emerged as tracer-based techniques for measuring PP from in situ observations and biogeochemical budget calculations. While the original incubation and tracer-based approaches have been applied widely, they require the collection of discrete samples from ships and therefore yield limited data coverage. Fortunately, advances in instrumentation have facilitated underway measurements of $O_2$/Ar and particulates at the surface, giving rise to methods for high-resolution ship surveys of NCP and NPP, respectively (Tortell, 2005; Kaiser et al., 2005; Burt et al., 2018). Sampling via instrumented moorings similarly enabled high temporal resolution GPP and NCP time-series at fixed positions (e.g., Emerson

and Stump, 2010; Johnson, 2010; Weeding and Trull, 2014; Fassbender et al., 2016). Yet, while promising, these ship and mooring-based approaches are subject to trade-offs between temporal, horizontal, and vertical measurement resolution. Moreover, many traditional approaches require expensive instrumentation (underway approaches) or considerable human oversight to collect the necessary data (incubation approaches), making them broadly inaccessible to the oceanography community or impractical for autonomous surveys. As a result of the challenges associated with the traditional PP methods, there are substantial gaps in PP datasets, with many ocean regions being under-sampled or omitted from archived records (Fig. 2a,b). While satellite and statistical algorithms can provide PP estimates (Behrenfeld and Falkowski, 1997; Huang et al., 2021; Li and Cassar, 2016) with enhanced space-time coverage, their utility is constrained by limitations such as the accuracy of satellite ocean colour observations (e.g., Long et al., 2021) and the inability to detect subsurface information (Gordon and McCluney, 1975). Ultimately, the challenges associated with quantifying PP from the various in situ and ex situ methods has resulted in large uncertainties in global estimates of GPP and NCP. Reported estimates of GPP, for example, range from 8 to 14 Pmol C $y^{-1}$ (Westberry and Behrenfeld, 2014; Huang et al., 2021), while estimates of NCP and carbon export range from 250 to 2650 Tmol $y^{-1}$ (Boyd and Trull, 2007; Henson et al., 2011; Siegel et al., 2016; Westberry et al., 2012).

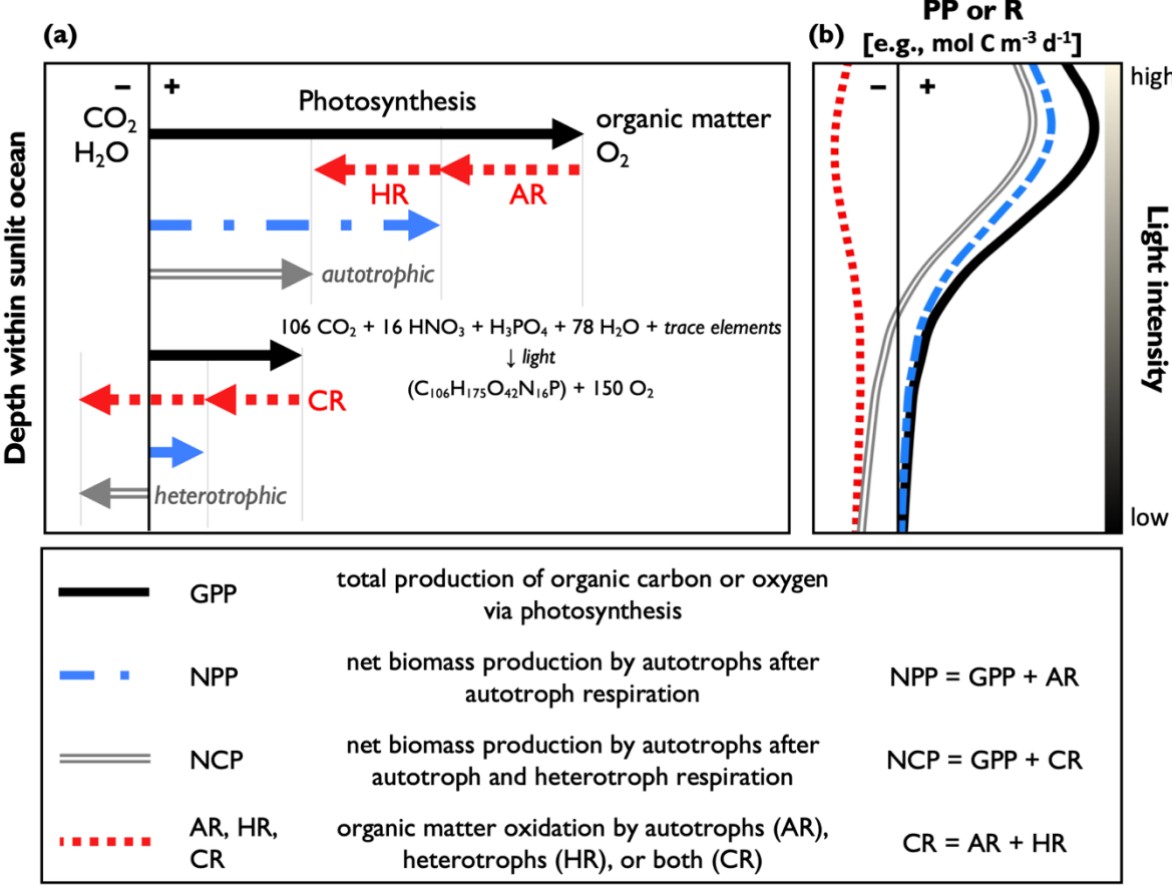

**Figure 1. A conceptual schematic and definitions of the common primary productivity (PP) and respiration (R) metrics: gross primary production (GPP), net primary production (NPP), net community production (NCP), and autotrophic, heterotrophic and community respiration (AR, HR, CR, respectively). Panel (a) shows simplified reaction equations of organic matter production and R. The upper part of the figure represents a region of net autotrophic conditions (NCP > 0), while the lower part represents a region of net heterotrophic conditions (NCP < 0). Panel (b) represents idealized PP and CR profiles, where PP declines with depth due to the light dependency of photosynthesis with a subsurface maximum resulting from photoinhibition. The vertical axis represents water column depth, and the thin black line divides positive and negative rates. The equation represents average oceanic aerobic photosynthesis, following Redfield nutrient stoichiometry. The reverse reaction represents respiration.**

Considering the challenges associated with the above-mentioned traditional PP approaches, emerging methods that use autonomous profiler observations present a significant opportunity to expand the spatial and temporal coverage of PP datasets and improve satellite-based observations via hybrid approaches. The BGC-Argo program, in particular, supports a growing array of profiling floats that provide continuous biogeochemical observations (e.g., $O_2$, pH, nitrate, chlorophyll fluorescence, particle backscatter as a proxy for organic matter) in the upper 2000 m of the global ocean at ~5- or 10-day intervals (Fig. 2d). The BGC-Argo fleet has grown steadily in recent years (>500 operational floats as of Feb. 2023), and the international community is targeting a sustained deployment of 1000 BGC floats distributed equally throughout the global ocean (Roemmich et al., 2021; Biogeochemical-Argo Planning Group., 2016). Several recent studies have quantified PP using BGC-Argo floats

and other autonomous profilers, including gliders (see Table A1 in the appendix, and references therein), demonstrating the potential to derive year-round, depth-resolved PP estimates in remote ocean regions (Fig. 2c).

The primary objective of this paper is to demonstrate the potential of autonomous platforms, exemplified by BGC-Argo floats, for expanding the spatial and temporal coverage of PP estimates in the upper ocean. This paper explores float-based approaches for estimating GPP and NCP, since those methods are more mature than emerging approaches for NPP quantification (Arteaga et al., 2022; Yang, 2021; Estapa et al., 2019; Long et al., 2021). While recent literature has presented float-based methods for quantifying PP metrics in the interior ocean (e.g., Martz et al., 2008; Hennon et al., 2016; Arteaga et al., 2019; Su et al., 2022), the focus of this manuscript is on methods that resolve processes occurring principally within the euphotic zone. To facilitate a full exploitation of these new opportunities, we take stock of the float-based tools currently available to researchers and identify their strengths and limitations. After providing an overview of the emerging float- and glider-based PP approaches, we present quantitative analyses to demonstrate the current application of these methods, as single or combined tools. Overall, this paper is intended as a resource for a broad readership – including researchers who do not normally perform PP calculations – that summarizes the current state of GPP and NCP methods and helps to familiarize the community-at-large with the current benefits, challenges and application of these new tools.

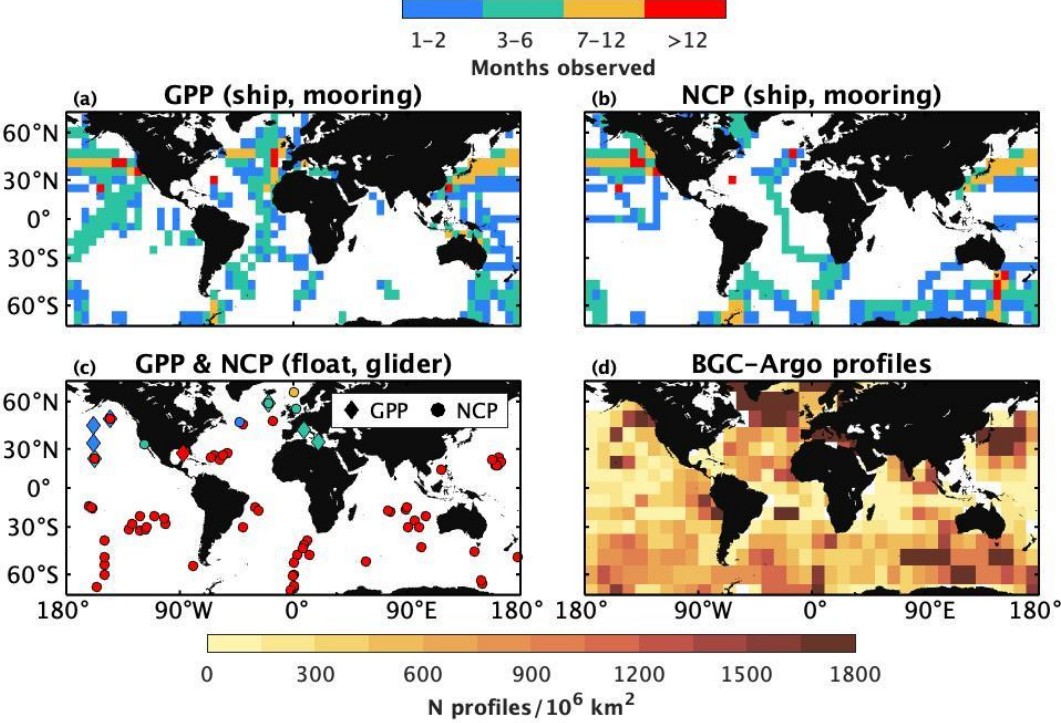

**Figure 2. Coverage of gross primary productivity (GPP) and net community productivity (NCP) datasets, and biogeochemical-Argo (BGC-Argo) profiles. The upper row represents archived GPP and NCP data obtained from ships or moorings, while panel (c) shows the locations and durations of float- or glider-based GPP and NCP studies. Panel (d) shows a heatmap of the distribution of BGC-Argo profiles collected from 2010 through 2022. Data in panels (a) and (b) were binned to a five-by-five-degree grid. Only floats equipped with at least one biogeochemical sensor and registered in the international BGC-Argo program were included in (d). A list of archived data sources is provided in the appendix.**

## 2 Overview of approaches and application details

This section provides an overview of approaches to quantifying GPP (measured as GOP and GCP) and NCP using observations made by BGC-Argo and other autonomous profilers. For each approach, we outline the premise and describe the specific variables used, sampling requirements, assumptions, and variations.

To date, autonomous GPP approaches have relied on measurements of $O_2$ and particulate organic carbon (POC). NCP calculations have relied on $O_2$, POC and nitrate ($NO_3^-$) measurements and estimates of dissolved inorganic carbon (DIC) and total alkalinity (TA). These tracers are selected because their concentrations in the sunlit ocean are impacted by primary production (photosynthesis and respiration). Other sources and sinks, such as exchange across the air-sea interface, vertical mixing, advection, and/or sinking and grazing, also impact the concentrations of these tracers. Accordingly, the temporal change in the concentration of a tracer, T, can be represented by the following general budget equation

$$\frac{d[T(t,z)]}{dt} = GPP(t,z) - CR(t,z) \pm other\ sources/sinks(t,z) \tag{1}$$

where [T(t,z)] is the tracer concentration at time, t, and depth, z, and $\frac{d[T(t,z)]}{dt}$ is its time rate of change, expressed in concentration units per unit time (e.g., mmol C m$^{-3}$ d$^{-1}$). The left-hand side of the equation is measured, while terms on the

right represent estimated quantities. Autonomous GPP methods interpret Eq. 1 over a 24-hr period and are premised on the widespread observation of diurnal cycles in O$_2$ and POC concentrations (Fig. 3). These cycles result from the dependency of photosynthesis on sunlight and are driven by daytime net autotrophic production (GPP–CR) and nighttime CR (e.g., Siegel et al., 1989; Johnson et al., 2006). Assuming that diurnal variability in the other source/sink terms in Eq. 1 is negligible, and that CR is constant over a 24-hr period, Eq. 1 can be approximated by the following equation

$$\frac{d[T(t,z)]}{dt} \approx GPP(t,z) - CR(z) \tag{2}$$

where T is O$_2$ or POC. Given Eq. 2, vertically resolved GCP or GOP estimates can be derived if the diurnal cycles of POC or O$_2$ in the euphotic zone are detectable.

Autonomous NCP approaches, in contrast, seek to interpret temporal changes in the concentration of a photosynthesis-respiration tracer over timescales exceeding one day (typically on the order of one week or more). Over these timescales, variability in the non-photosynthesis/respiration terms in Eq. 1 is not negligible. NCP (i.e., GPP+CR) is thus determined by re-arranging Eq. 1, as follows, and estimating the contributions of the other source/sink terms to the observed tracer time-series,

$$NCP(t,z) = \frac{d[T(t,z)]}{dt} \pm other\ sources/sinks(t,z). \tag{3}$$

Eq. 3 is typically evaluated at discrete time and depth intervals equivalent to the resolution of profiling measurements, or by integrating quantities over coarser depth ranges (e.g., the mixed layer).

As GPP and NCP methods evaluate Eq. 1 over contrasting timescales, different sampling approaches have been employed to obtain the requisite tracer time-series observations. For GPP calculations, multiple measurements per day are necessary to adequately resolve the diurnal cycle. Initially, GPP studies used a single profiling instrument, such as a glider (Nicholson et al., 2015; Barone et al., 2019), Lagrangian surface float (Briggs et al., 2018), or biogeochemical profiling float whose mission cycle was adjusted for frequent upper ocean profiling (Barbieux et al., 2022; Gordon et al., 2020; Henderikx Freitas et al.,

2020) (Fig. 3a,b). Gordon et al. (2020) and Barbieux et al. (2022), for example, used floats with profiling intervals of 3 and 6 hrs, respectively, to obtain diurnal cycle observations. The majority of the BGC-Argo fleet, however, collects a water column

profile every ~5 or 10 days. As a result, a diurnal cycle cannot be resolved using data from a single BGC-Argo float profiling at these intervals. This limitation was resolved by Johnson and Bif (2021) and Stoer and Fennel (2023) who quantified GOP and GCP from daily $O_2$ or POC cycles using a composite of observations from multiple floats within selected geographic regions. To achieve roughly equal coverage of all hours of the day, they compiled data from floats that profiled at non-integer intervals (e.g., 10.2, not 10.0 days). Then, GPP was estimated by fitting the photosynthesis curve through all the resulting data points (as in Johnson and Bif, 2021), or by first calculating hourly median POC or $O_2$ values (Stoer and Fennel, 2023) (Fig. 3c). Importantly, data from floats that do not sample all hours of the day evenly must be removed so that the resulting GPP estimates are not biased to a specific time of day. A non-integer sampling interval of 5.2 or 10.2 has been recommended to achieve approximately equal coverage of all hours over a float's lifecycle (Johnson and Bif, 2021; Stoer and Fennel, 2023). While GOP or GCP estimates derived from rapid profiling may yield daily temporal resolution (i.e., one GPP estimate per daily cycle) in ocean regions with strong diurnal variations, estimates derived from composite curves are more representative of typical conditions over the time and space scales that the data are composited. Sampling for NCP determinations has most commonly been based on nominal BGC-Argo profiling intervals, although high-resolution sampling using rapidly profiling floats is also feasible. Resulting NCP estimates have optimal vertical and temporal resolutions equivalent to those of the sampling profiling observations.

To estimate GOP, $O_2$ is best expressed as a concentration anomaly, $\Delta O_2$, calculated as the difference between observed and equilibrium concentrations (i.e., $\Delta O_2 = O_2 - O_{2,equil}$; all typically mmol $O_2$ m$^{-3}$). Equilibrium concentrations are calculated using corresponding temperature and salinity observations (Garcia and Gordon, 1992). This practice is recommended to minimize potential diurnal solubility effects on $\frac{d[O_2(t,z)]}{dt}$. In NCP calculations, $O_2$ is expressed as its absolute concentration. POC concentrations (typically mg m$^{-3}$) for GCP and NCP calculations are derived from particle backscatter ($b_{bp}$) or beam attenuation ($c_p$, typically at 660 nm) measurements (both m$^{-1}$) using regional algorithms (e.g., Loisel et al., 2011; Cetinić et al., 2012) or those derived from latitudinally distributed datasets (e.g., Graff et al., 2015 based on data obtained from the Atlantic Meridional Transect and equatorial Pacific) (see Table A4 for a list of selected POC algorithms). Many algorithms estimate POC from $b_{bp}$ at 700 nm ($b_{bp,700}$), the wavelength that is most commonly measured by BGC-Argo floats. For algorithms that rely on different $b_{bp}$ wavelengths (e.g., $b_{bp}$ at 470 nm, as in the algorithm of Graff et al., 2015), a power-law equation is required to convert between $b_{bp,700}$ and $b_{bp}$ at other wavelengths (Boss et al., 2013; Boss and Haëntjens, 2016). Only a subset of floats directly measures $b_{bp,470}$ or $c_{p,660}$. Lastly, because TA and DIC are not directly measured by BGC-Argo floats, NCP estimates derived using those variables rely on calculations of their concentrations using float measurements and an empirical TA function (Huang et al., 2022). Total alkalinity is estimated from float pH, $O_2$ and hydrographic observations using a neural network algorithm (e.g., Bittig et al., 2018; Carter et al., 2021), and DIC is subsequently calculated from float-pH and derived-TA based on known seawater carbonate system relationships (Gattuso et al., 2022).

## 2.1 GPP

Given a diurnal POC or $O_2$ time-series, GCP or GOP have been estimated using three different mathematical algorithms that describe the shape of the diurnal curve. Two of the approaches have been applied only using single profilers making multiple measurements of the upper ocean each day; the other has been adapted for composite daily cycles (Fig. 3). Each method yields one daily GPP estimate per diurnal curve, and estimates may be vertically resolved or integrated, depending on the sampling infrastructure used. As a result, the spatial and temporal resolution of the following methods is constrained by the measurement

resolution of the float or glider.

Briggs et al. (2018) described a method that requires estimating tracer sink terms (including CR) by fitting a type I linear regression to nighttime (sunset to sunrise) POC or $O_2$ data (red line in Fig. 3a). Extrapolating the regression line from the POC or $O_2$ value at sunrise (sunset) to noon on the following (preceding) day (dashed line in Fig. 3a) then yields an estimate of the tracer's mid-day concentration in the absence of photosynthesis. The difference between observed noontime concentrations

($[T(t,z)]_{observed}$) and the value predicted by the regression extrapolation ($[T(t,z)]_{predicted}$) is an indication of GPP, so that GPP is calculated as follows

$$GPP(z) = ([T(z)]_{predicted} - [T(z)]_{observed})\ (0.5\ day^{-1}). \tag{4}$$

Daily GPP is taken as the average of morning and afternoon values. This method has been applied by constructing diurnal $O_2$ or $c_p$-POC cycles from continuous, upper ocean observations using a Lagrangian surface float (Briggs et al., 2018), or from a float profiling at 3-hr intervals (Gordon et al., 2020). In both cases, surface layer-integrated GPP estimates were obtained by integrating $O_2$ or POC observations within a density-defined layer. A minimum upper ocean sampling resolution of ~3-4 hr is likely necessary to obtain a robust nighttime regression fit to the data and to derive GPP at daily resolution.

Barbieux et al. (2022), following Claustre et al. (2008), introduced another approach for GCP derivations from a rapidly-profiling BGC-Argo float deployed in the Mediterranean Sea. In their method, GCP is estimated by solving the following differential equation for the time rate of change in depth-resolved POC concentrations

$$\frac{d[POC(t,z)]}{dt} = \mu(t,z)\ POC(t,z) - L(t,z)\ POC(t,z), \tag{5.1}$$

where $\mu$ represents autotrophic growth, and L represents particle losses due to CR, sinking, and grazing (both $d^{-1}$). Eq. 5.1 is a variation of Eq. 2, where $\mu(t,z)\ POC(t,z)$ and $L(t,z)\ POC(t,z)$ are equivalent to GPP(t,z) and CR(t,z), respectively. Integrating Eq. 5.1 between sunset ($SS_0$) and the following sunrise ($SR_1$), when $\mu$=0, yields an estimate for the loss term,

$$L(z) = \frac{ln\left(\frac{POC(z,SS_0)}{POC(z,SR_1)}\right)}{SR_1 - SS_0}.$$ (5.2)

Combining Eqs. 5.1 and 5.2, assuming constant L(z), and integrating over a full day (sunrise to sunrise; $SS_0$ to $SS_1$) produces the following equation for daily GPP

$$GPP(z) = POC(SR_1, z) - POC(SR_0, z) + L(z)\sum_{i=1}^{j}(t_{i+1} - t_i)\frac{POC(t_{i+1}, z) + POC(t_i, z)}{2},$$ (5.3)

where the index $i$ represents time-resolved POC measurements from sunrise on the first day ($SR_0$) to sunrise on the following day ($SR_1$) (Fig. 3b). Barbieux et al. (2022) used a BGC-Argo float profiling at 6-hr intervals, thus enabling GCP calculations with daily resolution. POC quantities were integrated vertically in three upper ocean layers.

A third approach for estimating GPP has been applied successfully using $O_2$ observations from gliders (Nicholson et al., 2015; Barone et al., 2019), a rapidly profiling BGC-Argo float (Henderikx Freitas et al., 2020), and a composite of $O_2$ and $b_{bp}$-POC cycles from BGC-Argo floats (Johnson and Bif, 2021; Stoer and Fennel, 2023). In this method, introduced by Nicholson et al. (2015), Eq. 2 is re-written to describe discrete, time-dependent changes in POC or $O_2$ as a function of time-variable irradiance, E(t),

$$T(t_1, z) = T(t_0, z) + GPP(z)\frac{\int_{t_0}^{t_1} E(t)dt}{\overline{E}} - CR(z)(t_1 - t_0),$$ (6)

given $\frac{d[T(t,z)]}{dt} \approx \frac{[T(t_1,z)] - [T(t_0,z)]}{t_1 - t_0}$, and where $\underline{E}$ and $t_1$-$t_0$ are the mean daily irradiance level and time step, respectively. The

middle term, $GPP(z)\frac{\int_{t_0}^{t_1} E(t)dt}{\overline{E}}$, represents photosynthesis as a function of time-varying irradiance, which is calculated from

245 geospatial (location and time) data. A photosynthesis-versus-irradiance (P-vs-E) relationship, a sinusoidal, and a linear

algorithm have been proposed for $\frac{\int_{t_0}^{t_1} E(t)dt}{\overline{E}}$ (see coloured lines in Fig. 3c), although resulting GPP estimates are not statistically

different across models (Barone et al., 2019; Henderikx Freitas et al., 2020). Given time-resolved $\Delta O_2$ or POC observations, Eq. 6 can be re-expressed as a system of linear equations (see Eq. 4 in Barone et al., 2019), and GPP and CR are approximated

as the least squares coefficients required to fit $\frac{\int_{t_0}^{t_1} E(t)dt}{\overline{E}}$ to the observed diurnal cycle. MATLAB code for solving the system

of linear equations has been provided by Barone et al. (2019) and modified by Johnson and Bif (2021). Stoer and Fennel (2023) modified the code further and adapted it for Python.

To simplify the system of equations, Nicholson et al. (2015), Johnson and Bif (2021) and Stoer and Fennel (2023) assumed equivalency between daily integrated GPP and CR. Although the assumption is physically invalid in many ocean regions since

it may unrealistically constrain daily NCP to zero, it enables calculations of statistically robust GPP estimates in ocean regions where diurnal oscillations are small. Barone et al. (2019), in contrast, calculated separate GPP and CR values, albeit with larger errors in each term. Similarly, Gordon et al. (2020) attempted separate GPP, CR, and NCP estimates by applying the Briggs et al. (2018) method for float data collected from the Gulf of Mexico.

Surface layer-integrated GOP has been derived by applying this approach to observations obtained from gliders (Nicholson et al., 2015; Barone et al., 2019) or rapidly profiling floats (Henderikx Freitas et al., 2020). In principle, these sampling methods can yield daily diurnal curves and GOP estimates. In practice, however, the resulting GOP values may have an effective temporal resolution of ~5-7 days in low-productivity regions, due, in part, to limited detection (i.e., low signal-to-noise ratio) of daily $O_2$ oscillations (Barone et al., 2019). Johnson and Bif (2021) and Stoer and Fennel (2023) extended the present approach for composite sampling exploiting the broader BGC-Argo array. Johnson and Bif (2021) collated float $\Delta O_2$ data in different geographic regions between 2010 and 2020, constructing vertically resolved diurnal cycles by binning the composited datasets in 10-m intervals, and averaging values to the nearest local hour. GPP is calculated for a single composited diurnal curve, as described above. Stoer and Fennel (2023) further extended the approach by calculating GCP from $b_{bp}$-POC and using observations median-binned to each local hour. Using data from a meta-analysis by Moran et al. (2022), they calculated an average percent extracellular release (PER) to account for dissolved organic carbon (DOC) production not detected by the $b_{bp}$ sensor. Accordingly, they scaled their GCP values using the calculated PER value and converted between GCP and GOP using a photosynthetic quotient (PQ) value of 1.4, i.e., $GOP = \frac{GCP}{(1-PER)}PQ$. Finally, Johnson and Bif (2021) and Stoer and Fennel (2023) derived NPP from the diurnal GPP calculations by applying a global empirical GOP:NPP ratio of 2.7 mol $O_2$ (mol C)$^{-1}$ (i.e., NPP = GOP / 2.7).

The horizontal and temporal resolution of the present approach based on composited sampling is limited by the number of floats and profiles in a given geographic region. There must be enough profiles taken equally throughout the day to distinguish a daily signal. Johnson and Bif (2021) estimated that a minimum of 20 and 50 $O_2$ profiles in each hour (equivalent to 480 to 1200 profiles, per day) are required to clearly detect diurnal variability in tropical and high-latitude waters, respectively. For the region 30–70°S, Stoer and Fennel (2023) estimated that at least 2000 $b_{bp}$ and 5000 $O_2$ profiles, per diurnal curve, are required to limit the noise-to-signal ratio of the resulting PP estimates to one, or less.

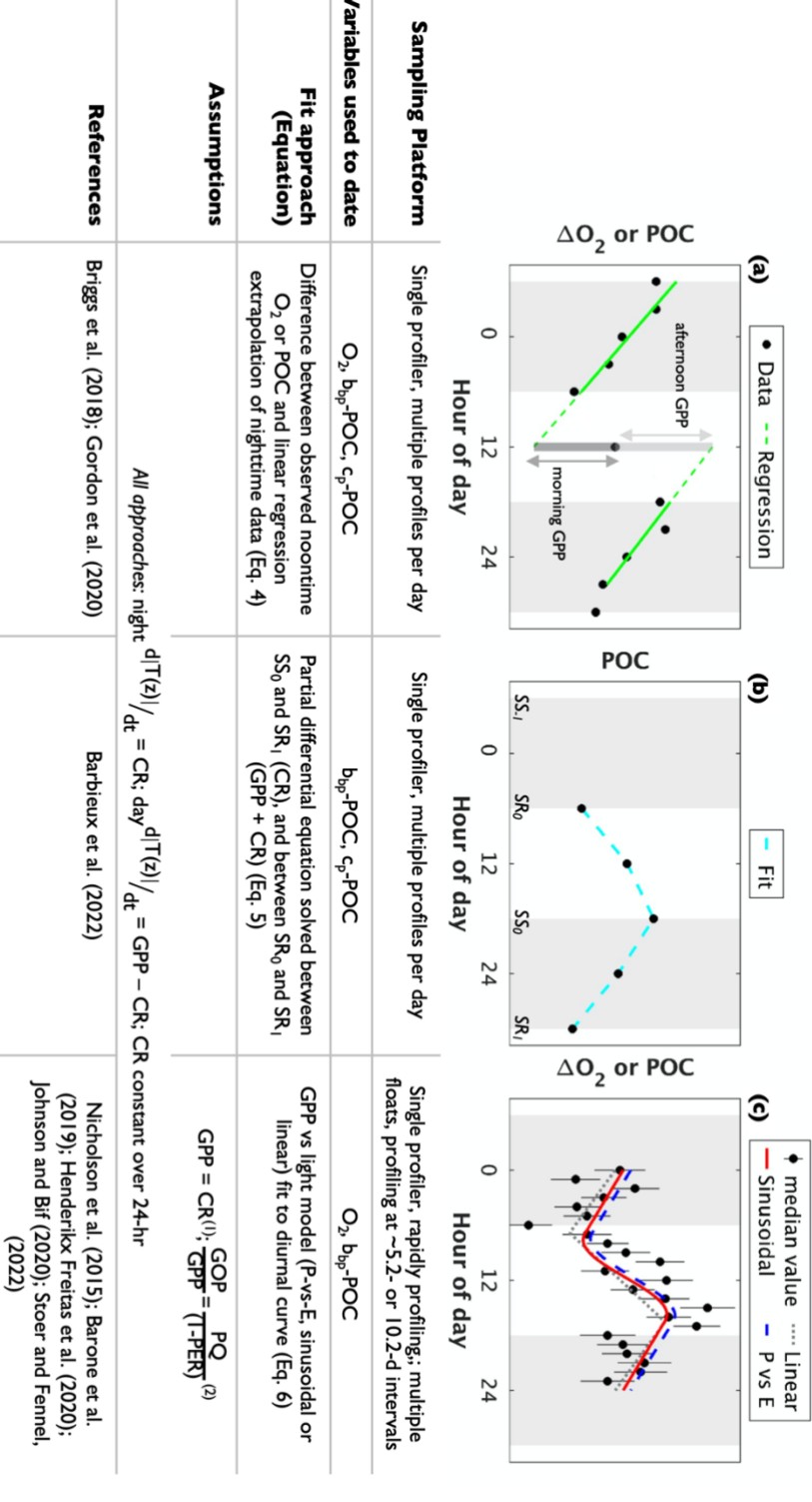

| Sampling Platform | Single profiler, multiple profiles per day | Single profiler, multiple profiles per day | Single profiler, rapidly profiling; multiple floats, profiling at ~5.2- or 10.2-d intervals |
|---|---|---|---|
| Variables used to date (Equation) | $O_2$, $b_{bp}$-POC, $c_p$-POC | $b_{bp}$-POC, $c_p$-POC | $O_2$, $b_{bp}$-POC |
| Fit approach (Equation) | Difference between observed noontime $O_2$ or POC and linear regression extrapolation of nighttime data (Eq. 4) | Partial differential equation solved between $SS_0$ and $SR_1$ (CR), and between $SR_0$ and $SR_1$ (GPP + CR) (Eq. 5) | GPP vs light model (P-vs-E, sinusoidal or linear) fit to diurnal curve (Eq. 6) |
| Assumptions | All approaches: night $\left.\frac{d[T(z)]}{dt}\right\| = CR$; day $\left.\frac{d[T(z)]}{dt}\right\| = GPP - CR$, CR constant over 24-hr | | $GPP = CR^{(1)}$, $\frac{GOP}{GPP} = \frac{PQ}{(1-PER)}$ (2) |
| References | Briggs et al. (2018); Gordon et al. (2020) | Barbieux et al. (2022) | Nicholson et al. (2015); Barone et al. (2019); Henderikx Freitas et al. (2020); Johnson and Bif (2020); Stoer and Fennel, (2022) |

**Figure 3. Conceptual schematic of autonomous gross primary production (GPP) methods. The black markers in the figures represent oxygen anomaly (ΔO₂) or particulate organic carbon (POC). In (a) and (b) markers represent data obtained using a single profiling platform, while those in (c) represent median (± standard deviation) data values during each hour of the day. The grey bars represent approximate nighttime periods between sunset (SS) and sunrise the following day (SR). The lower part of the schematic summarizes the approach requirements. The "Variables used to-date" row identifies the tracers that have been used successfully, so far, under each method applied using autonomous profilers. It does not necessarily limit the respective float-based methods to those tracers, alone. Notes: (1) assumption applied in Nicholson et al. (2015), Johnson and Bif (2021), Stoer and Fennel (2023); (2) assumption applied in Stoer and Fennel (2023) only.**

## 2.2 NCP

Autonomous NCP methods invoke a different set of calculations and assumptions than GPP methods. Namely, the sum of non-biological terms (i.e., physical fluxes) is estimated and subtracted from observed tracer changes in discrete time and depth intervals (as in Eq. 3). Equation 3 is commonly solved using a one- or two-dimensional box model approach by partitioning the water column into layers (e.g., mixed layer, euphotic zone) or by discretizing in depth intervals (Table 1; Fig. 4), and performing calculations between consecutive profiles (e.g., dt in Eq. 3 is the float profiling interval) or as seasonally integrated quantities (e.g., Baetge et al., 2020). The following equation describes the calculations performed at each timestep and in each depth layer

$$NCP(t,z) \; = \; (h_{i+1} - h_i)\frac{[T(t_1,z)]-[T(t_0,z)]}{t_1-t_0} \pm \; \Sigma F(t,z) \, . \tag{7.1}$$

NCP(t,z) (typically mol T m$^{-2}$ d$^{-1}$) represents NCP integrated over the depth range $h_{i+1} - h_i$ (m). [T(t,z)] is the average tracer concentration between $h_i$ and $h_{i+1}$, and $\frac{[T(t_1,z)]-[T(t_0,z)]}{t_1-t_0}$ is the observed change in the tracer's concentration between time intervals (both mol T m$^{-3}$ d$^{-1}$). Lastly, ΣF is the sum of the estimated physical fluxes and non-NCP biological terms (mol T m$^{-2}$ d$^{-1}$). Integrating the resulting NCP values over one year provides an estimate of annual net community production (ANCP; mol T m$^{-2}$ yr$^{-1}$), which is equivalent to carbon export when integrated to the depth of the maximum annual mixed layer (Yang et al., 2017). However, the depth to which NCP and ANCP estimates are integrated impacts the interpretation and magnitude of the resulting NCP values and metabolic state of the system. Haskell et al. (2020), for example, reported ~10-20% variability in climatological ANCP and monthly NCP estimates calculated down to the seasonal mixed layer depth (MLD), euphotic zone, 100 m, and annual maximum MLD. Pelland et al. (2018) noted ~50% variation in ANCP values when integrating to the seasonal MLD versus 120 m. Ship-based work has also demonstrated the sensitivity of export estimates to the depth of wintertime ventilation, with regions of deep winter MLDs experiencing greater ventilation, and therefore, reduced export or ANCP calculated to that depth (Palevsky et al., 2016).

The general approach represented by Eq. 7.1 has been applied using float-based O$_2$, NO$_3^-$, DIC, TA, and POC, although there is significant variability in how and which physical fluxes are included when calculating ΣF, and in how the box model is discretized in time and space (Table 1). Air-sea gas exchange (gases only), vertical and lateral exchange or transport and evaporation/precipitation (excluding O$_2$) are important processes that modify tracer concentrations over daily to monthly timescales (Bushinsky and Emerson, 2015; Emerson and Stump, 2010; Huang et al., 2022; Pelland et al., 2018). Accordingly, ΣF is estimated by calculating some or all of the terms in the following equation,

$$\Sigma F(t,z) = F_{AS}(t,z) + F_{EP}(t,z) + F_{vmix}(t,z) + F_{vadv}(t,z) + F_{ent}(t,z) + F_{horiz}(t,z) + F_{bio}(t,z) \tag{7.2}$$

$F_{AS}$ represents gas exchange via bubbles ($F_{bub}$) and diffusion ($F_{diff}$) at the air-sea interface, $F_{EP}$ is the evaporation/precipitation flux at the surface, $F_{vmix} + F_{vadv} + F_{ent}$ are vertical transport via diapycnal mixing, advection, and entrainment, respectively, and $F_{horiz}$ is horizontal transport. $F_{bio}$ represents biological processes, not including NCP, such as particulate inorganic C production/consumption, DOC production, or POC sinking, which are reflected in the DIC, TA, and POC budgets (Huang et al., 2022). The general equations for the physical terms in Eq. 7.2 are as follows

$$F_{diff}(t, z = 0) = k(t)\big([T(t,0)] - [T(t,0)]_{eq}\big) \tag{7.3}$$

$$F_{bub}(t, z = 0) = \beta(F_C(t) + F_p(t)) \tag{7.4}$$

$$F_{EP}(t, z = 0) = T{:}S\left(\frac{d[S(t,0)]}{dt} - \frac{d[S(t,0)]}{dt}\bigg|_{phys}\right) \tag{7.5}$$

$$F_{vmix}(t, z) = \kappa_Z(t, z)\frac{d[T(t,z)]}{dZ} \tag{7.6}$$

$$F_{vadv}(t, z) = w(t, z)\,\Delta[T]_z(t, z) \tag{7.7}$$

$$F_{ent}(t, z) = \begin{cases} \Delta[T]_z\frac{dh}{dt}; & \frac{dh}{dt} > 0 \\ 0; & \frac{dh}{dt} \leq 0 \end{cases} \tag{7.8}$$

$$F_{horiz}(t, z) = u(t, z)\,\Delta[T]_x(t, z) + v(t, z)\Delta[T]_y(t, z) \tag{7.9}$$

where k is the wind speed-dependent diffusive gas transfer coefficient (m d$^{-1}$), $[T]_{eq}$ is the temperature- and salinity-dependent equilibrium concentration at ambient sea level pressure (mol T m$^{-3}$), and $F_C + F_p$ represent bubble-mediated gas transfer via small and large bubbles, respectively. The $\beta$ term is a bubble-flux tuning coefficient between 0 and 1. The evaporation/precipitation term (Eq. 7.5) is typically estimated by normalizing tracer concentrations to the observed salinity during each time step, and multiplying by the measured time-dependent change in salinity (Fassbender et al., 2016; Huang et al., 2022). In Eq. 7.5, T:S is the ratio of tracer T to salinity, $\frac{d[S(t,0)]}{dt}$ is the observed change in salinity over time, and $\frac{d[S(t,0)]}{dt}\big|_{phys}$ is the change due to physical processes. $F_{diff}$, $F_{bub}$, and $F_{EP}$ are zero below the surface box. The transport terms $\kappa_Z$ (m$^2$ d$^{-1}$), w, dh/dt, u and v (all m d$^{-1}$) represent the diapycnal eddy diffusivity coefficient, vertical advection velocity, the rate

of change of a given depth layer, and the lateral advection velocities, respectively. d[T]/dZ (mol m$^{-4}$) is the vertical gradient between consecutive depth bins, while $\Delta[T]_z$, $\Delta[T]_x$, and $\Delta[T]_y$ (all mol m$^{-3}$) represent concentration differences in vertical and horizontal directions. Importantly, when evaluating NCP following Eq. 7, it is assumed that the float remains in a single water mass, such that tracer changes strictly represent temporal variations due to NCP and the processes described in Eqs. 7.3-7.9. In reality, however, this may not always be the case, and the resulting effect on NCP calculations remains a source of uncertainty that is difficult to constrain.

As summarized in Table 1 and Table A3, different studies have represented the terms in Eqs. 7.3-7.9 in different ways. Parameterizations of air-sea exchange (Eqs. 7.3-7.4) and diapycnal mixing (Eq. 7.6) vary most widely across studies, and those fluxes typically contribute the largest source of uncertainty in budget-based NCP and ANCP calculations, up to ~40% and 20%, respectively (Bushinsky and Emerson, 2015; Yang et al., 2017; Huang et al., 2022). Different $F_{diff}$ + $F_{bub}$ parameterizations, for example, have been employed, and efforts have been made to tune those terms for local conditions using a scaling coefficient (β). Yang et al. (2017) and Emerson et al. (2019) tuned $F_C$ + $F_P$ for Ocean Station Papa (OSP) by minimizing differences between observed mixed layer $N_2$ concentrations and values predicted by the same mass balance used for their $O_2$-based ANCP calculations. Plant et al. (2016) tuned $F_{bub}$ by scaling the magnitude of that flux to minimize differences between $O_2$- and $NO_3^-$-based ANCP estimates. Most recently, Yang et al. (2022) introduced a correction for air-sea flux estimates that relies on reanalysis data products to account for small temperature differences in the ocean skin (the ~500 µm thick layer over which gas exchange occurs) and mixed layer which impact the magnitude of diffusive and bubble-mediated gas exchange. Only that paper and a subsequent one by Emerson and Yang (2022) have applied the correction, but its influence on ANCP estimates may be as large as ~40%. Approaches to estimating the diapycnal mixing flux also differ widely across studies. Most invoke values from the literature, either selecting constant or time-varying climatological $\kappa_Z$ values for the study region. Bushinsky and Emerson (2015) and Huang et al. (2022) used an average OSP $\kappa_Z$ time-series from Cronin et al. (2015) for the base of the mixed layer, and scaled values vertically to a background of 10$^{-5}$ m$^2$ s$^{-1}$ below the thermocline, following Sun et al. (2013). Haskell et al. (2020) scaled the Cronin et al. (2015) $\kappa_Z$ climatology for their NCP model by minimizing differences between $NO_3^-$- and DIC-based ANCP estimates. These approaches, however, are somewhat problematic as they likely neglect significant spatial and temporal variability in upper ocean mixing rates. Pelland et al. (2018) derived independent estimates of all the transport terms ($\kappa_Z$, w, u, v) by using their glider observations to close heat and salt budgets for OSP, while Plant et al. (2016) estimated the physical transport terms by running locally forced simulations of a Price-Weller-Pinkel (PWP) mixed layer model (Price et al., 1986). Other studies have estimated vertical advection velocities (u) by calculating the Ekman pumping velocity from local wind stress data. Most float-based approaches neglect horizontal transport, suggesting its influence on NCP estimates would be small away from boundary currents, eddies, or frontal zones, and over seasonal timescales, or longer (e.g., Yang et al., 2017; Huang et al., 2018). Emerson and Bushinsky (2015) is the only float-based study to have calculated that term, and found it to be small relative to the vertical physical fluxes, contributing <7% to uncertainty in their ANCP estimates. In a glider-based study, however, horizontal advection fluxes were larger than the sum of all vertical fluxes in the upper 120 and 200 m of the water column (e.g., Pelland et al., 2018). Lastly, entrainment

terms, which are often estimated from observed changes in the mixed layer depth or other depth horizons between time intervals, tend to be small, except during periods of rapid mixed layer depth changes.

Different approaches to setting up the vertical discretization have been also applied. For example, Bushinsky and Emerson (2015), Plant et al. (2016) and Pelland et al. (2018) divided the upper water column into multiple depth layers with ~1.5-5 m vertical resolution. Other studies have employed coarser one- or two-box model frameworks, partitioning the upper water column into layers defined by the seasonal or winter maximum mixed layer depth (MLD), euphotic depth, or a fixed density or depth horizon (e.g., Yang et al., 2017; Haskell et al., 2020; Huang et al., 2022). In all cases, the vertical transport and mixing flux terms are evaluated by measuring the depth-dependent change in T (dT/dZ or $\Delta T_Z$) across the base of each box (Fig. 4), and air-sea exchange and/or evaporation are quantified at the top of the surface box, only. There is no consensus on the optimal vertical discretization scheme, and no estimates of the (A)NCP sensitivity to the approach have been reported.

By performing simultaneous NCP calculations using multiple tracers, it is possible to partition biological productivity into distinct biogenic pools, and to estimate other non-NCP biological terms ($F_{bio}$ in Eq. 7.2; Haskell et al., 2020; Huang et al., 2022). For example, while calculations based on $O_2$ and $NO_3^-$ target particulate and dissolved organic C cycling, those based on DIC or TA are also influenced by inorganic C cycling associated with non-NCP production of calcareous shells by some organisms (Fassbender et al., 2016). Calculations from POC represent only the particulate organic fraction, as well as POC sinking. As a result, differences between DIC, TA, POC, and $O_2$ or $NO_3^-$ based estimates can be used to quantify sinking rates, and the relative importance of particulate organic, dissolved organic and particulate inorganic production within a system (see details in Huang et al., 2022).

Finally, while most NCP studies to date have performed the above calculations at the approximate resolution of the profiling instrument, a handful of studies have evaluated NCP by integrating tracer changes over seasonal timescales (Table A1). Johnson et al. (2017), Bif and Hansell (2019), and Baetge et al. (2020) all estimated NCP as the winter-to-summer drawdown of $NO_3^-$ in the upper 100 m, 75m and 200 m, respectively, neglecting any other $NO_3^-$ sources or sinks (i.e., $h_i$-$h_{i+1}$ = 100 m, $T(t) = \int_0^{100} \Delta NO_3^- dz$ and $\Sigma F = 0$ in Eq. 7.1). A reference winter profile is taken from float observations, and $NO_3^-$ drawdown is converted to C- or $O_2$-equivalents using Redfield stoichiometry.

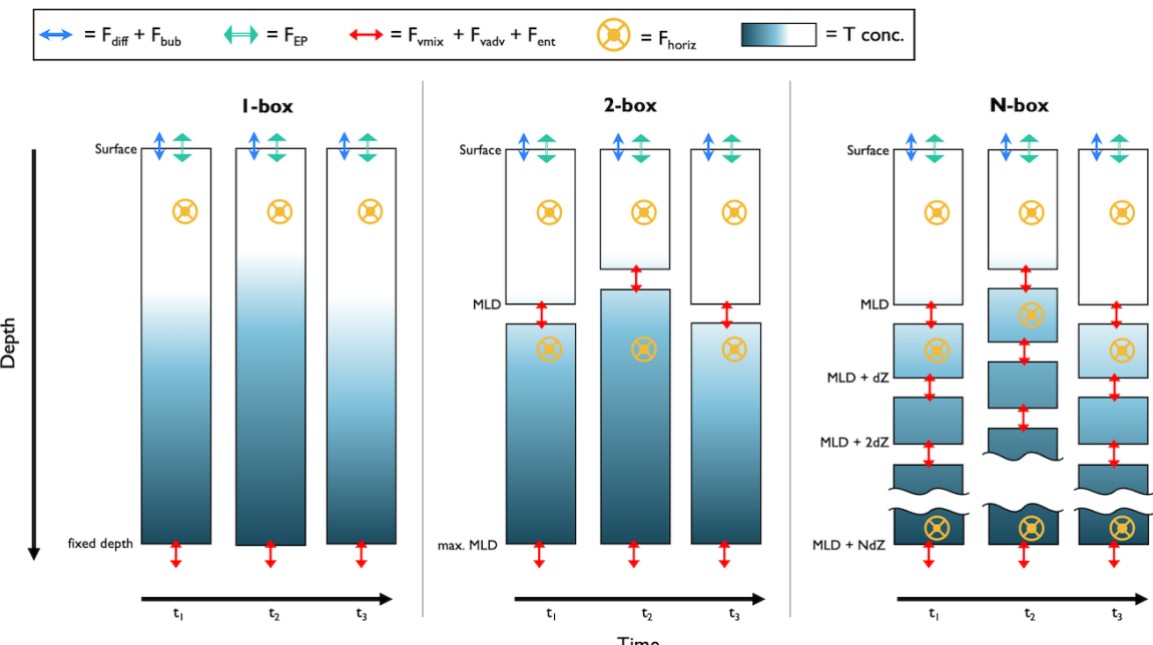

Figure 4 **Schematic of box model setups for float-based NCP approaches. The columns represent a profile of tracer concentration at discrete time intervals ($t_n$, where n is the n-th point in time) and the rows are divided into vertical layers (dZ). T conc = tracer concentration; MLD = mixed layer depth; $F_{diff}$ = air-sea flux via diffusion; $F_{bub}$ = air-sea flux via bubbles; $F_{EP}$ = evaporation or precipitation flux; ; $F_{vmix}$ = vertical transport flux via diapycnal mixing; $F_{vadv}$ = vertical transport flux via advection; $F_{ent}$ = vertical transport flux via entrainment; and $F_{horiz}$ = horizontal transport flux.**

**Table 1 Variations in budget terms used in float- and glider-based net community productivity (NAP) calculations at Ocean Station Papa.**

| Study | Study Abbrev. | Platform | T | Vertical resolution | $F_{AS} + F_{EP}$ (surface only) | $\kappa_z$ [m² s⁻¹] | w [m s⁻¹] | dh/dt [m d⁻¹] | u + v [m d⁻¹] |
|---|---|---|---|---|---|---|---|---|---|
| Bushinsky and Emerson (2015) | BE15 | Float | $O_2$ | N box (0-MLD; MLD-150 m, Δh = 1.5m) | $F_{AS}=k_{O2} (O_2-O_{2,eq}) + \beta(F_{bub.})$; Liang et al. (2013) | Cronin et al. (2015) [surface]; Sun et al. (2013) [profile] | Ekman pumping velocity ; > 0 only | derived from observations ; > 0 only | NCEP/NCAR reanalysis |
| Plant et al. (2016) | P16 | Float | $O_2$, $NO_3^-$ | N box (0-180 m, Δh = 2m) | $F_{AS}=k_{O2} (O_2-O_{2,eq}) + \beta(F_{bub.})$; Liang et al. (2013); scaled to $NO_3^-$ ANCP | $1.5 \times 10^{-5}$ | PWP | PWP | 0 |
| Yang et al. (2017) | Y17 | Float | $O_2$ | 2 box (0-MLD; MLD-max. MLD) | $F_{AS}=k_{O2} (O_2-O_{2,eq}) + \beta(F_{bub.})$; $\beta = 0.53$; Liang et al. (2013) | $1.5 \times 10^{-5}$ (box 2 only) | 0 | derived from observations ; > 0 only | 0 |
| Pelland et al. (2018) | P18 | Glider | $O_2$ | 86 boxes (0-150 m, Δh = 2m; 150-200 m, Δh = 5m) | $F_{AS}=k_{O2} (O_2-O_{2,eq}) + \beta(F_{bub.})$; $\beta = 0.29$; Liang et al. (2013) | T/S budget | T/S budget | 0 | T/S budget |
| Haskell et al. (2020) | H20 | Float | $NO_3^-$ | 1 box (0- MLD, EuZ, 100 m or max. MLD) | $F_{EP}=(dS/dt - dS/dt_{phys}) (T{:}S)$ | Cronin et al., (2015); scaled to DIC budget | Ekman pumping velocity ; > 0 only | derived from observations ; > 0 only | 0 |
| Huang et al (2022) | H22 | Float | $O_2$, $NO_3^-$, POC, DIC, TA | 1 box (0-56 m) | $F_{EP}= k_{CO2} (k_H) (\Delta pCO_2) + (dS/dt-dS/dt) (C{:}S)$; $F_{AS}=k_{O2} (O_2-O_{2,eq}) + (F_{bub.})$; $k_{CO2}$ from Wanninkhof (2014) ; $F_{AS,O2}$ from Liang et al. (2013) | Cronin et al. (2015) [surface]; Sun et al. (2013) [profile] | Ekman pumping velocity | derived from observations ; > 0 and MLD > 56 m, otherwise 0 | 0 |

$O_2$ = oxygen concentration (mol m⁻³); TA = total alkalinity (mol m⁻³); POC = particulate organic carbon concentration (mol m⁻³); $NO_3^-$ = nitrate concentration (mol m⁻³); DIC = dissolved inorganic carbon concentration (mol m⁻³); MLD = mixed layer depth; $F_{AS}$ = air-sea gas exchange (mol m⁻² d⁻¹); $F_{bub.}$ = air-sea bubble flux (mol m⁻² d⁻¹); $F_{EP}$ = evaporation or precipitation (mol m⁻² d⁻¹); $k_{O2}$ and $k_{CO2}$ = air-sea gas transfer coefficient for $O_2$ and $CO_2$; $\kappa_z$ = eddy diffusivity coefficient [m² d⁻¹]; w = vertical advection velocity [m d⁻¹]; dh/dt = change in layer depth [m d⁻¹]; u + v = horizontal advection velocities [m d⁻¹]; Δh = box vertical displacement ($h_{i-1}$ – $h_i$ in Eq. 7); β = bubble-mediated transfer scaling coefficient [unitless]; PWP = Price-Weller-Pinkel mixed layer model (Price et al., 1986); C:S = observed DIC:salinity ratio [mol C:S]. "Surface" refers to values derived in the mixed layer only, while "profile" is for values derived over the full water column.

## 3 Overview of the current capacity to derive GPP and NCP estimates from BGC-Argo floats

Here, we summarize and demonstrate, through examples, the current capacity to determine GPP and NCP using the BGC-Argo array. The main goal of this section is to provide readers with an overview of how the emerging float-based methods are applied. Sections 3.1 and 3.2 demonstrate the methods' applications at local and basin-to-global scales, respectively. In section 4, we discuss the current challenges and opportunity to further broaden the scope of GPP and NCP calculations using floats.

### 3.1 GPP and NCP calculations at local scales

To date, a handful of studies have examined GPP and NCP dynamics at relatively small spatial scales, using data from one or several floats deployed within a single geographic region. Targeted GPP studies employing single BGC-Argo (or BGC-Argo-like) floats have only occurred in the Mediterranean Sea (Barbieux et al., 2022), N Pacific (Henderikx Freitas et al., 2020), and Gulf of Mexico (Gordon et al., 2020). Gordon et al. (2020), however, were unable to reliably determine GOP from their diurnal $O_2$ curves due to low biological productivity and confounding signals from physical $O_2$ fluxes. While Barbieux et al. (2022) successfully derived an approximately-four-month euphotic-zone integrated GCP time-series in two locations in the Mediterranean Sea using $c_p$-POC data, diurnal variations in the $b_{bp}$-to-POC relationship precluded the same calculations using $b_{bp}$-POC data.

Float-based NCP studies are somewhat more numerous than GPP studies (Table A2) but are similarly limited in their geographic extent. NCP has been well-studied around Ocean Station Papa (OSP; 50ºN, 145ºW) in the subarctic NE Pacific (sect. 3.1.1), and only a handful of localized studies have occurred elsewhere, such as in the S. China Sea (Huang et al., 2018) and the NW Atlantic (Alkire et al., 2014; Yang et al., 2021) (Fig. 2c). These studies have spanned from about one year to several, and have employed single floats, or multiple floats clustered within the same region. Plant et al. (2016), for example, used float data from six floats that were deployed independently and consecutively between 2008 and 2013.

Several float-based studies have quantified ACNP in the Southern Ocean, however, that work has principally focused on processes occurring below the euphotic zone (e.g., Martz et al., 2008; Hennon et al., 2016; Arteaga et al., 2019; Su et al., 2022) No single study has examined NCP and GPP dynamics simultaneously, although Alkire et al. (2014, 2012) and Briggs et al. (2018) studied NCP and GPP during the same NW Atlantic spring bloom in their respective papers.

### 3.1.1 NCP case study at OSP

Ocean Station Papa is one of longest running time series with sustained oceanographic observation. In the past 20 years, the monitoring site has seen several deployments of Biogeochemical-Argo floats and profiling gliders, allowing for various studies to estimate NCP. To demonstrate the current abilities to calculate NCP studies at a local scale, we performed a case study

analysis of studies that utilized float/glider data to estimate NCP at OSP. A similar analysis is not presently feasible for GPP, owing to the small number of localized studies using floats and gliders, and the currently insufficient number of profiles available to conduct GPP calculations from composite diurnal cycles. Indeed, there have not been enough published float-based GPP studies to date in a single region to compile those data and perform an analysis similar to the present NCP analysis. Moreover, we could not perform our own local GPP calculations due to the high number of profiles required to make those calculations. These factors currently preclude an analogous analysis of GPP methods at localized scales.

We compiled all available published float- and glider NCP data collected from OSP between 2008 and 2020. The published data constitute five independent studies, each employing slightly different approaches to quantifying NCP and ANCP (Table 1). For comparison with the profiler data, we also compiled independent NCP estimates from ship-board sampling, moorings, and satellites collected over the same timeframe as the float/glider data. We present time-explicit, seasonal average, and annual integrated NCP values integrated to the depth of the annual maximum winter mixed layer (typically ~120 m at OSP), and depth-resolved seasonal average NCP. All values were converted to $O_2$ equivalents using a PQ of 1.4, and $O_2:NO_3^-$ ratio of 150:-16. Data sources and a detailed description of our data handling are provided in the appendix (Table A1).

The compilation of float and glider data from OSP yields a nearly continuous, 12-year time-series of NCP and ANCP estimates, and a shorter, seven-year time-series of depth-resolved estimates (Fig. 5). The temporal resolution of estimates ranges from 10 days (float profiling interval; Plant et al., 2016; Huang et al., 2022) to one month (Pelland et al., 2018). Yang et al. (2017) provided NCP estimates interpolated to one-day resolution, while data provided by Haskell et al. (2020) were averaged over six years. The depth-resolved data from Plant et al. (2016) and Pelland et al. (2018) had vertical resolutions of 2 and 2-5 m, respectively.

There is general consistency in the magnitude (NCP, ANCP) and seasonal patterns (NCP) across the float and glider studies. Most datasets, for example, reveal peak productivity and autotrophy (NCP > 0) between June and August, and minimum values and heterotrophy (NCP < 0) between November and February (Fig. 5a,b). These patterns are also broadly consistent with those of the independent data records. Indeed, the average seasonal float NCP cycle is very similar to the average of ship-based measurements between January and July (compare white and red markers in Fig. 5b), and the seasonality is similar to the average estimates derived from moorings and satellites. Notably, while all float/glider approaches consistently predict periodic net heterotrophic conditions, the satellite-based approaches only ever produce positive NCP estimates, reflecting how those algorithms are trained using only positive PP data (Li and Cassar, 2016; Westberry et al., 2008; Behrenfeld and Falkowski, 1997).

The float/glider ANCP estimates are typically within one standard deviation of one another (Fig. 5d). Exceptions to this result are the Huang et al. (2022) $O_2$-based estimate and the Haskell et al. (2020) $NO_3^-$-based estimate. It is, however, somewhat unsurprising that the Huang et al. estimate exceeds the others because ANCP values from that publication were integrated only to 50 m depth (i.e., calculations integrated to the annual maximum MLD were not available) and may thus exclude subsurface regions of net heterotrophy which occur during the fall and winter (Fig. 5c). For the same reason, it is not surprising that the

float- and glider ANCP estimates are typically lower than estimates derived from moorings (Fassbender et al., 2016; Emerson and Stump, 2010), satellites and ships, which only resolve a narrow depth range in the upper ocean.

     Despite the general agreement across float- and glider NCP approaches, there are some important differences, which are particularly apparent in the full, time-resolved NCP record (Fig. 5a). For example, NCP estimates made at the same time diverge by up to ~50 mmol $O_2$ $m^{-2}$ $d^{-1}$, and in extreme cases, ~100 mmol $O_2$ $m^{-2}$ $d^{-1}$ across different approaches (Fig. 5a).

Likewise, the spread in average seasonal NCP values is ~50 $O_2$ $m^{-2}$ $d^{-1}$ (Fig. 5b). The most notable difference across studies is the anomalous phenology of the Pelland et al. (2018) record, which identifies peak NCP in March, and net heterotrophy in September and October, only. These differences are also seen in the depth-resolved record from that publication. Interestingly, however, the anomalies in the seasonal record of Pelland et al. (2018) do not correspond with anomalous ANCP.

     Despite these differences, our analysis demonstrates strong agreement across different float-based NCP studies and illustrates
the capacity to derive NCP time-series using consecutive float deployments. In section 4.2, we discuss the factors that contribute to differences in the NCP results presented in Fig. 5.

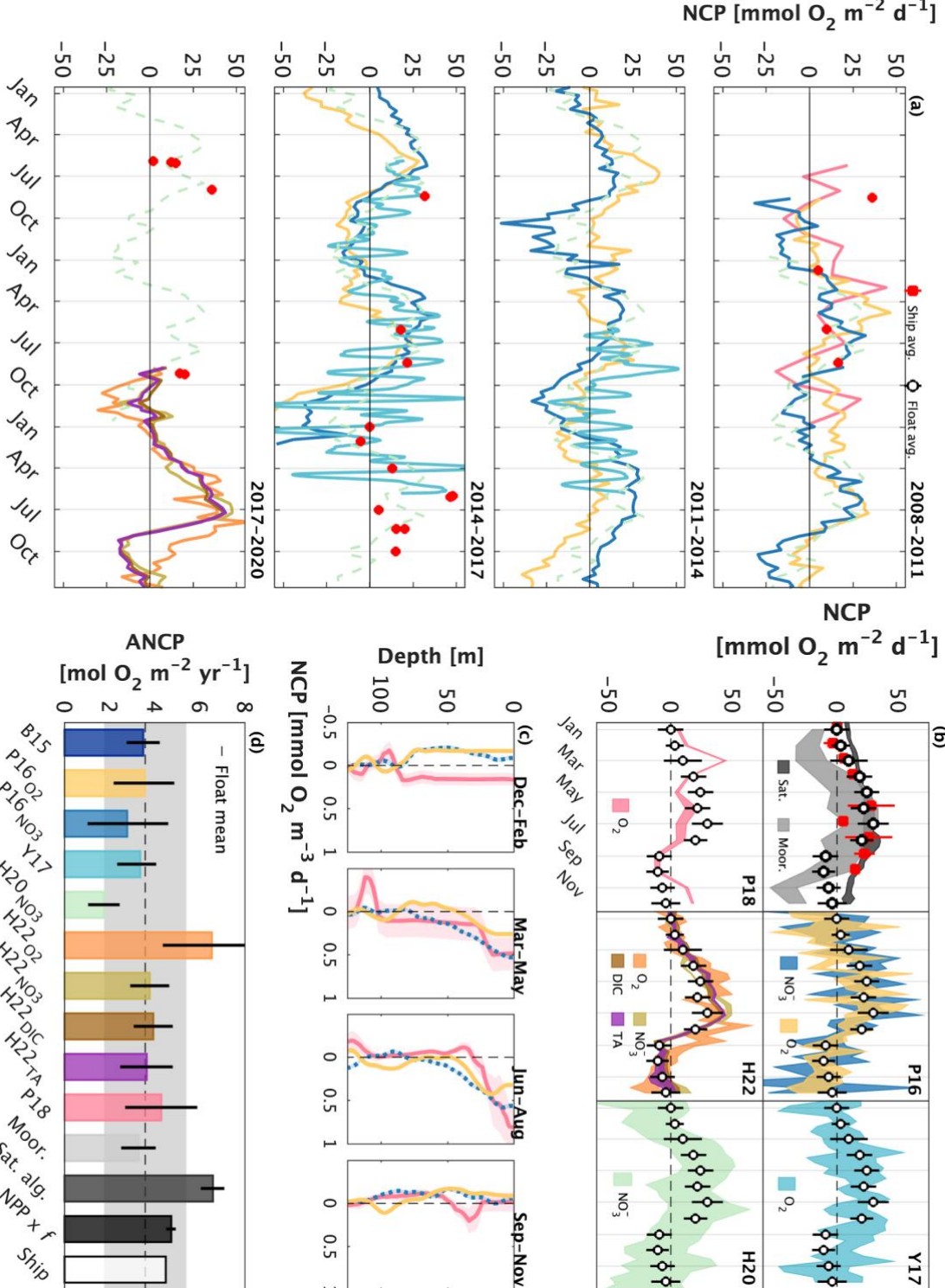

**Figure 5 (previous page).** Published float- and glider-based net community production (NCP) estimates from Ocean Station Papa (OSP) located at 50°N and 145°W. (a) Full time-series NCP. Red markers are ship-based estimates. The Haskell et al. (2020; H20) record (light green) is dashed because it represents an average annual cycle between 2009 and 2018. (b) Average seasonal cycles, presented at the temporal resolution of each study. Shading around the mean represents the reported approach uncertainty or the standard deviation of estimates derived over multiple years. The black and red markers and error bars represent the average ± one standard deviation annual cycle derived from float and ship sampling, respectively. Depth-resolved NCP estimates in (c) are from Plant et al. (2016; P16) and Pelland et al. (2018; P18). (d) Annual integrated NCP (ANCP), including data from mooring studies (Emerson 2010; 2014; Fassbender et al. 2016; Haskell et al., 2020), a satellite algorithm (Li and Cassar, 2016), a combination of satellite net primary productivity (NPP) and an empirical estimate of the ratio of carbon export to NPP (or the f-ratio) (Westberry et al., 2008; Laws et al., 2011) and ship-based sampling. Colours in (a)-(c) correspond with labels in (b) and (d). Values in (a), (b), and (d) represent quantities integrated to the annual maximum mixed layer depth. The subscripts (e.g., H20$_{NO3}$) denote the tracer used in each study. Y17 = Yang et al. (2017); H22 = Huang et al. (2022).

## 3.2 GPP and NCP calculations on basin and global scales

Few studies have examined PP at basin or global scales using float data. Johnson and Bif (2021) provided the first global assessment of decadal GOP and NPP derived from a compilation of float observations, while Stoer and Fennel (2023) presented float-based GPP and NPP estimates of the southern hemisphere ocean. Both studies performed depth-resolved and euphotic zone-integrated calculations by subsetting all available BGC-Argo $O_2$ and/or $b_{bp}$-POC data into different geographic regions. Johnson and Bif (2022) performed calculations in 10-degree latitude bands in the Northern and Southern hemispheres, subdividing the data into annual and bi-monthly segments. They also performed calculations at 2-monthly intervals around the Bermuda Atlantic Time-series Station and Hawaii Ocean Time-series sites. Stoer and Fennel (2023), in contrast, performed calculations between 30° and 70°S, only, due to an insufficient number of $b_{bp}$ profiles north of that region at the time.

No studies to date have estimated global NCP from floats. Two Southern Ocean studies (Johnson et al., 2017; Huang et al. 2023) and two subtropical ocean studies (Yang et al. 2019; Emerson and Yang, 2022) have, however, provided extensive assessments of (A)NCP from a compilation of multiple floats. Johnson et al. (2017) used BGC-Argo data to characterize ANCP in the Southern Ocean by compiling $NO_3^-$ data from 24 floats deployed between 2009 and 2016. Similarly, Huang et al. (2023) provided basin-scale estimates of NCP in different biogenic carbon pools in the Southern Ocean, derived using a compilation of floats and multiple tracers (DIC, TA, $NO_3^-$, POC). Yang et al. (2019) and Emerson et al. (2022) compiled $O_2$ data from multiple floats to estimate ANCP in the North and South Hemisphere Subtropical Ocean. Lastly, some recent work (e.g., Martz et al., 2008; Hennon et al., 2016; Arteaga et al., 2019; Su et al., 2022) compiled data from subsets of the Southern Ocean BGC-Argo array to quantify ANCP and respiration below the euphotic zone.

No work has simultaneously characterized NCP and GPP at global or regional scales using BGC-Argo data.

### 3.2.1 Global GPP case study

Building on recent work by Johnson and Bif (2021) and Stoer and Fennel (2023), we performed new global GOP and GCP calculations using the available BGC-Argo array. We summarize those calculations here and provide further details in the

appendix. Presently, a similar analysis is not feasible for NCP, as global scale NCP calculations have not yet been attempted by the community, and only a small handful of studies have calculated NCP at basin scales (see section 3.1). As a result, intercomparisons of published results at these scales are not feasible, and new calculations of global NCP are beyond the scope of the present paper.

For our GPP calculations, we followed Stoer and Fennel (2023), by compiling all available high-quality BGC-Argo $\Delta O_2$ and $b_{bp}$-POC data collected between January 2010 and December 2022. We subset the data into 10 m depth bins, from 0 to 200 m, and different spatial groups, representing 10° latitudinal bands (70°S to 70°N) or Longhurst Biogeographical Provinces (Longhurst, 2006; Flanders Marine Institute, 2009). We constructed composite diurnal curves in each spatial subset by calculating the median $\Delta O_2$ or $b_{bp}$-POC value at each hour of the day. We subsequently calculated GPP by fitting a sinusoidal function to the resulting diurnal curves (sect. 2.1). We accounted for DOC production by scaling $b_{bp}$-GPP estimates by a global mean PER value of 33% (Moran et al., 2022), and converted GCP to $O_2$ equivalents using a photosynthetic quotient of 1.4 (Laws, 1991) (i.e., $\frac{b_{bp}-GCP}{1-0.33} 1.4$).

These calculations yield spatially explicit, depth-resolved $\Delta O_2$-GOP and $b_{bp}$-GCP estimates, representing a median snapshot from 2010 to 2021. Our calculations extend the work of Johnson and Bif (2021) and Stoer and Fennel (2023) by 1) attempting simultaneous $\Delta O_2$-GOP and $b_{bp}$-GCP calculations in different biogeochemical provinces and latitude bands of northern and southern hemisphere waters, 2) comparing the float-based data to archived GOP datasets (Table A1), and 3) assessing the availability of float profiles to perform GPP calculations.

We compiled a total of ~222,300 $O_2$ and ~103,800 $b_{bp}$ profile observations. After discarding data from floats that did not profile all hours of the day evenly (i.e., floats that sampled at integer intervals, 5- or 10-day) only ~23% ($O_2$) and 24% ($b_{bp}$) of the original datasets were available for our GPP calculations (compare dashed and solid lines in Fig. 6a). This processing also resulted in significantly more $O_2$ and $b_{bp}$ profiles in the Southern Ocean, and typically fewer than 1000 $b_{bp}$ profiles in each latitude band or province in the northern hemisphere.

We were able to derive GOP estimates in 26/32 non-coastal provinces and 12/14 latitude bands, and GCP in 11/32 provinces and 4/14 latitude bands (Fig. 6b). GCP calculations were not feasible in most northern latitude regions due to an insufficient number of profiles, based on thresholds estimated in Johnson and Bif (2021) and Stoer and Fennel (2023). Among the regions with sufficient profiles, ~32% and 20% of the dataset had negative or unrealistic $O_2$- or $b_{bp}$-GPP values, resulting from poor detection of a diurnal curve. In waters shallower than 60 m, these values decrease to ~19 and 17%, respectively, owing to more pronounced photosynthesis in surface waters.

There is generally good agreement between float $O_2$- and $b_{bp}$-based GPP and between the float estimates and independent GOP estimates derived from bottle sampling (Fig. 6b,c). These results are best seen in surface waters and in vertical profiles of the Southern Ocean. We did directly not compare the vertical profile float-GPP values against independent bottle samples due to the increasing errors in float GPP with depth. There is also reasonable agreement between our $O_2$-GOP calculations in surface waters (<20 m) and those reported in Johnson and Bif (2021) (yellow line in Fig. 6b). The median difference between our

estimates and those of Johnson and Bif (2021) was ~-0.2 mmol $O_2$ $m^{-3}$ $d^{-1}$, on average (range -0.7-1.6 mmol $O_2$ $m^{-3}$ $d^{-1}$), excluding latitude bands centred at between 5° and 15°S, where there were too few profiles for Johnson and Bif (2021) to derive estimates. At those latitudes, we were able to derive GOP estimates, but the resulting values have high uncertainty (shading in Fig. 6b), owing to the small number of profiles (~600 at both latitude bands) in that region. The low number of profiles and high uncertainty in the low-latitude regions likely also explain the offset between our float-based GOP, and the archived data in that region. We suspect that once more profiles are collected, we will see stronger agreement between the float- and ship-based estimates.

It is also noteworthy that depth-resolved GPP values derived using the sinusoidal, linear, and P-vs-E algorithms agree within one standard error of the approach for both $O_2$ and $b_{bp}$-based estimates (Fig. 6c). In the upper 100m for the region of 30-70°S, the average range of GPP values derived using the three algorithms was only 0.4 and 0.1 mmol $O_2$ $m^{-3}$ $d^{-1}$ for $O_2$- and $b_{bp}$-based estimates, respectively.

Overall, the histogram distributions of the float-based GPP estimates demonstrate broad agreement between float and bottle-sample GPP estimates, at all depths shallower than 100m (Fig. 6d). The distributions suggest that float-based, decadal estimates are within the range of expected values derived from bottle sampling, albeit with a slight tendency for lower estimates in the float dataset (median float-based $O_2$- $b_{bp}$- and archived-GPP values of 0.7, 0.5, 1.3 mmol $O_2$ $m^{-3}$ $d^{-1}$, respectively). This result, however, is unsurprising as diurnal cycles derived from a composite of observations obtained over multiple years will also have dampened amplitude relative to daily cycles observed over a single day or composited over a single season. This result may also reflect a high proportion of negative or undetectable GPP values in the float dataset, and a summertime (i.e., high-GPP) sampling bias in the bottle sample record (~65% of the dataset).

In summary, our GPP case study results demonstrate 1) the general insensitivity of calculated GPP values to how the diurnal cycle is constructed (i.e., median binned, as in Stoer and Fennel, 2023, or unbinned as in Johnson and Bif, 2021); 2) that different GPP algorithms give similar results, although the sinusoidal fit tends to have the smallest error; 3) the robustness of the decadal GPP estimates to the addition of new profiles since calculations were performed by Johnson and Bif (2021) using data available up to 2021; and 4) that float-based GPP estimates are within the range of expected values.

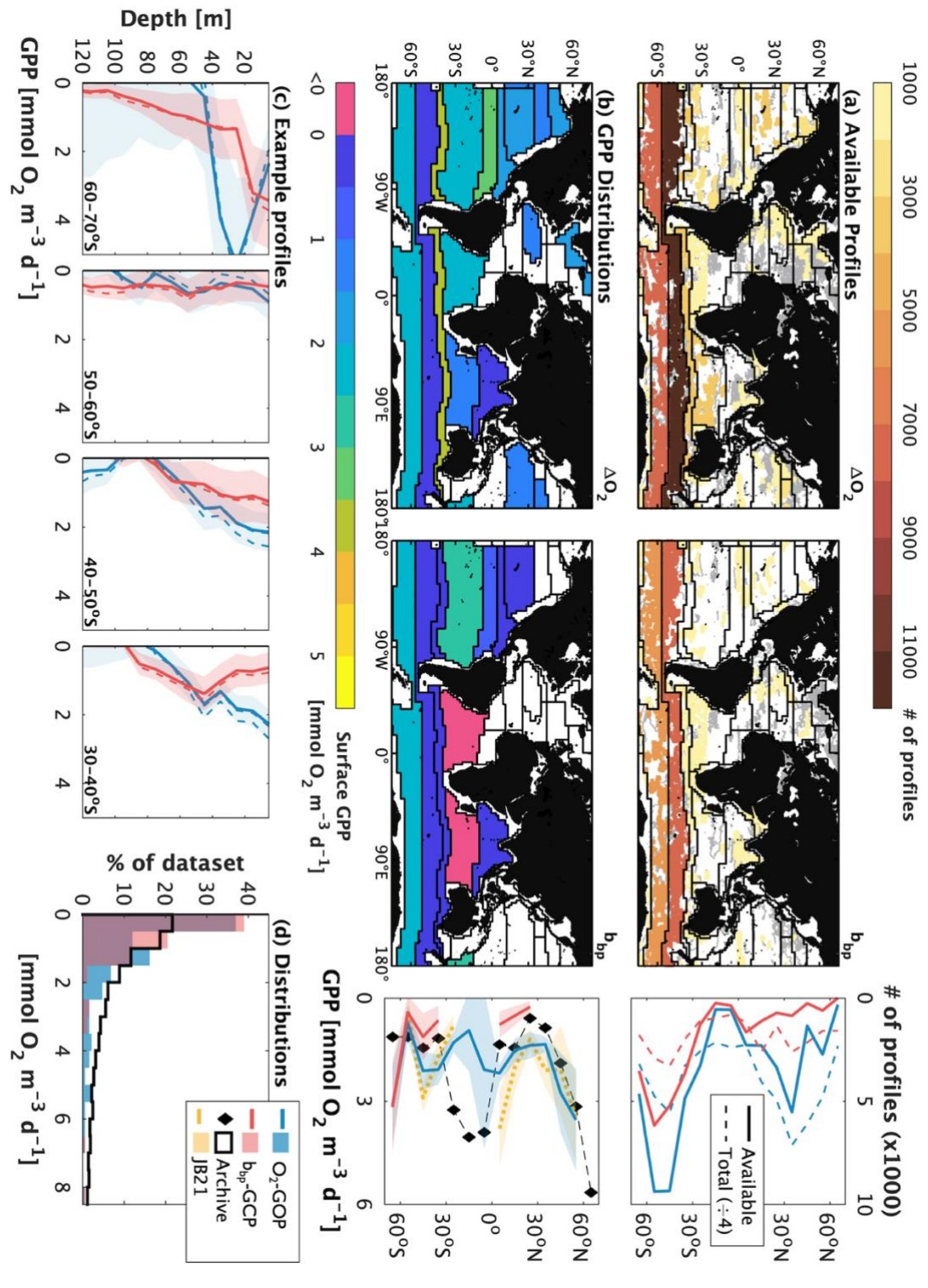

**Figure 6 (previous page). Demonstration of global-scale float-based gross oxygen productivity (GOP) and gross carbon productivity (GCP) estimates. (a) shows the distribution of biogeochemical-Argo (BGC-Argo) profiles, collected between January 2010 to December 2022 that are available for GOP or GCP calculations in Longhurst Biogeographical Provinces or by latitude. The coloured markers in the maps represent the profile locations of floats that sample all hours of the day evenly, while the grey markers represent profiles obtained from floats that do not. The colour identifies the total number of profiles in each province, whose boundaries are identified by the black lines. In the latitudinal distribution, the solid line represents the number of profiles available for GPP calculations, and the dashed lines represent the total number of profiles collected (divided by four, for comparison), including those from floats that did not sample all hours evenly. Panel (b) shows the distribution of surface (0-20 m average) GPP estimates by province or latitude band. Regions without data reflect an insufficient number of profiles available for calculations. Panel (c) shows an example of vertical GPP profiles in the southern hemisphere, and (d) shows the histogram distribution of float-based, and archived GOP data, derived from ship bottle sampling at all latitudes in waters shallower than 100 m. In panels (b) and (d), the black markers/lines represent archived bottle-sample GOP data, median-binned by latitude band, and the yellow line represents $\Delta O_2$-GOP estimates from Johnson and Bif (2021; JB21). The thin dashed lines in (c) are GOP estimates derived using the linear and P-vs-E algorithms; the solid lines are from the sinusoidal algorithm. Throughout, POC-based GCP estimates were converted to $O_2$ equivalents using a photosynthetic quotient of 1.4 and dissolved primary production of 33%.**

## 4 Discussion

### 4.1 Constraints on GPP accuracy and coverage

Float-based GPP estimates have been shown to compare well with independent data, and $O_2$ and $b_{bp}$-based estimates generally correlate with one another (p-value < 0.05 and $R^2$ = 0.47 through paired data in upper 60 m; Fig. 7). With some exceptions (e.g., surface waters between 0-30°N) offsets between $O_2$ and $b_{bp}$-based estimates are often within the standard error of the diurnal cycle approach (Fig. 6b-c, and see results from Johnson and Bif, 2021; Stoer and Fennel, 2023). However, when compared directly, the ratio between $\Delta O_2$-GOP and $b_{bp}$-GCP is not always consistent with the expected relationships based on documented PQ and PER variability (Fig. 7). For example, given an estimated range of ~18-47% DOC production during photosynthesis (median PER value of 32.5% ± 14.4% standard deviation calculated from Moran et al., 2022), and a PQ range of 1-1.45 (Laws, 1991), the ratio between $\Delta O_2$-GOP and $b_{bp}$-GCP uncorrected for PER should be between ~1.2 and 2.6 (shaded region in Fig. 7). Considering an even broader PER range of ~2-50% (global confidence interval from Baines and Pace, 1991) results in an expected GOP:GCP ratio of ~1-2.9. In our depth-resolved, global GPP dataset, we derived a median ratio of ~3.1 ± 0.2 (median ± confidence interval) for estimates derived in the upper 60 m. When considering all depths (up to 200 m), the median ratio is ~4.1 ± 0.6, reflecting the lower signal-to-noise ratio of diurnal $O_2$ or $b_{bp}$ variability at depth. For comparison, Briggs et al. (2018) calculated a ratio of ~2.6 between mixed layer $O_2$-GOP and $c_p$-GCP during a NW Atlantic spring bloom. These results imply higher PQ values and/or DOC production rates and may indicate that these terms are non-uniform across the global ocean. Using static PQ or PER values in GPP calculations (as in Stoer and Fennel, 2023 and in our global GPP case study) likely contributes to the uncertainty in the resulting GPP datasets, and partially explains the offsets we observed between $O_2$- and POC-based GPP estimates, and differences between the float- and bottle sample GPP values. Other sources of uncertainty and causes for potential and apparent offsets between $O_2$- and POC-based estimates are discussed in the following paragraphs. We note that our analysis presented in Fig. 7 is, unfortunately, unable to discern geographic patterns in or predictors of the GOP:GCP relationship due to an insufficient number of floats available for calculations in most geographic regions (see

next section). However, future work should use float data to explore potential relationships between the GOP:GCP ratio and $NO_3^-$ concentrations (a predictor of the fractional contribution of DOC-to-total carbon production) or latitude.

Diurnal cycle GPP methods are based on the presumption that day-night variations in photosynthesis are the primary driver of diurnal variations in upper ocean $O_2$ or POC concentrations. Other than accounting for potential diurnal solubility impacts on $O_2$ (through expressing $O_2$ as its concentration anomaly, $\Delta O_2$) no attempts have been made to reconcile for additional diurnal variations in float estimates of $O_2$ or POC that are not caused by photosynthesis. For $O_2$, these include potential impacts due to air-sea exchange or vertical mixing, and for POC, sinking, diel vertical migration and grazing, or PER. Yet, these processes vary throughout the day, and the extent to which they do changes seasonally and geographically. Diurnal variability in solar heating and wind forcing influence mixed layer dynamics on hourly, or longer, timescales, with impacts on air-sea gas exchange (Briggs et al., 2018) and near-surface vertical mixing (Price et al., 1986). Moreover, particle sinking, grazing, or DOC production, have been implicated as a mechanism for decoupling $O_2$- and POC-based PP estimates, particularly in high-productivity (e.g., diatom-dominated) regions (e.g., Rosengard et al., 2020). For example, regions of high POC sinking rates, grazing or PER will decouple $O_2$ and POC concentrations, leading to high-$O_2$ and low-POC in upper ocean waters, with implications for resulting GPP and CR estimates (White et al., 2017; Rosengard et al., 2020; Briggs et al., 2018). Similarly, day-night variations in grazing, resulting from diel vertical migrations, could amplify the nighttime decline in POC, thereby artificially inflating nighttime respiration estimates, and decoupling $O_2$- and POC-based GPP calculations. Independently or in combination, these processes likely imprint on the daily signals detected by BGC-Argo floats, whether by single assets or the composite of the array, and therefore constitute a source of uncertainty to the resulting GPP estimates.

The use of POC to estimate GPP also requires the assumption that gross community production is equal to autotrophic gross carbon production (White et al., 2017; Henderikx Freitas et al., 2022; Stoer and Fennel, 2023), and that daily cycles of non-algal particles are negligible. Often, however, this may not be the case. Moran et al (2022) suggested that bacterial carbon production contributes a small, but highly variable, fraction to particulate PP, equal to ~13±19% (mean ± one standard deviation), or <10% of total PP if PER is ~30%. For the size range relevant to $b_{bp}$, Martinez-Vincente et al. (2012) further suggested that the variability in $b_{bp}$ largely results from variability in phytoplankton between 2 and 20 μm in diameter, despite the majority of the $b_{bp}$ signal coming from highly abundant bacteria. Thus, if diurnal variability in $b_{bp}$ is mainly attributed to phytoplankton, then the $b_{bp}$ daily signal may still be a close proxy of GPP. Nonetheless, it is important to consider other potential sources of variability in $b_{bp}$ attributed to non-algal particles.

Variations in the $b_{bp}$-to-POC relationship, both in space and in time, also contribute a key source of uncertainty in the POC-based GPP estimates. Several algorithms between $b_{bp}$ and POC exist, including the algorithm of Graff et al. (2015), which was derived using a latitudinally-distributed dataset obtained from the Atlantic Meridional Transect and equatorial Pacific , and several regional ones (e.g., Loisel et al., 2011; Cetinić et al., 2012). We, and Stoer and Fennel (2023) used a $b_{bp}$-to-POC relationship based on a globally distributed dataset, which may not be appropriate for all ocean regions or depths (Bol et al., 2018). Moreover, diurnal variations in the $b_{bp}$-to-POC relationship have been implicated in the uncertainties in $b_{bp}$-POC-based GPP estimates in the Mediterranean and NW Atlantic (Briggs et al., 2018; Barbieux et al., 2022). Such variations may be

attributed to changes in the phytoplankton carbon-to-$b_{bp}$ ratio (Poulin et al., 2018) or refractive index (Henderikx-Freitas et al., 2022), which will confound interpretations of particulate productivity. Beam attenuation-based GCP estimates ($c_p$-GCP), however, appear to be more reliable than those derived from $b_{bp}$ due to the dampened diurnal variability in the $c_p$-to-POC relationship (Briggs et al., 2018; Barbieux et al., 2022). At this time, though, $c_p$ is not widely measured on BGC-Argo floats, and a far greater proportion of BGC-Argo floats already measure $b_{bp}$.

Differences in sampling time and location, including offsets in the number and locations of $O_2$ versus $b_{bp}$ profiles, will also contribute to uncertainty in GPP comparisons. This includes differences between the timing and locations of independent bottle samples (see markers in Fig. 1) and float profiles, as well as differences in the timing and location of float $O_2$ and $b_{bp}$ profiles. For these reasons, it is not surprising that the relationship between $\Delta O_2$-GOP and $b_{bp}$-GCP is less robust when considering the non-co-located float profiles (data not shown).

Finally, a critical number of profiles are needed to accurately estimate GPP from daily cycles of composite float profiles. As mentioned here and in previous studies (Johnson and Bif, 2021; Stoer and Fennel, 2023), a large number of floats are discarded from calculations because they do not sample all hours of the day evenly, presently reducing the number of profiles available for GPP calculations by ~75%. As a result, calculations are precluded in many regions or latitude bands, particularly those based on $b_{bp}$, and the resulting values are likely less robust. In the N Atlantic Ocean, for example, many floats currently do not

sample all hours of the day evenly (compare grey and coloured markers in Fig. 6a), preventing GPP calculations in a number of provinces in that region. For this method to be applied more broadly, floats need to cycle at all hours of the day. To achieve this, float manufacturers should ensure that the sampling protocols can be readily adjusted to the recommended profiling interval of 5.2 or 10.2 days by users via the float firmware. We discuss, in more detail, the minimum number of floats required for robust GPP calculations in the following section.

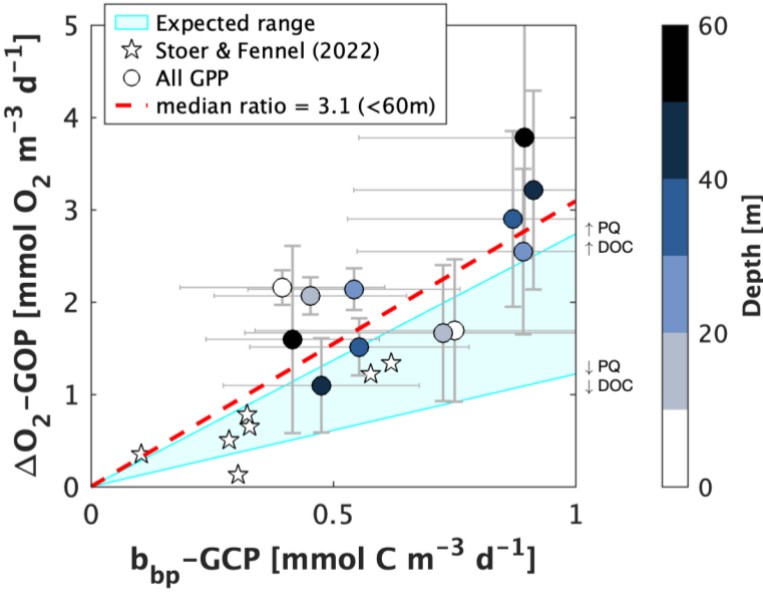

**Figure 7. A comparison of depth-resolved oxygen-based estimates of gross oxygen production ($\Delta O_2$-GOP) and particle backscattering-based estimates of gross carbon production ($b_{bp}$-GCP) in waters shallower than 60 m. Data points represent values derived from co-located profiles in latitude bands or Longhurst Provinces with enough profile measurements to obtain statistically consistent GPP estimates (sect. 4.1.1). Error bars represent one standard error. Star markers represent GPP estimates from Stoer and Fennel (2023) which were obtained from co-located $O_2$ and $b_{bp}$ profile measurements below the euphotic depth in the latitude range 30-60 ºS. GCP estimates were not converted to $O_2$ equivalents, nor were they adjusted for potential PER. The light blue shading represents the expected range for the relationship between GOP and GCP, given a percent extracellular release (PER) range of 18-47% (Moran et al., 2022) and photosynthetic quotient (PQ) range of 1-1.45 (Laws, 1991). The dashed line shows the median GOP:GCP ratio below 60 m.**

### 4.1.1 How many floats are required for consistent, annual GPP estimates?

Following Stoer and Fennel (2023), we performed a bootstrapping analysis to determine the number of $O_2$ and $b_{bp}$ profiles required to obtain stable GOP or GCP estimates in different latitude bands. We performed the analysis in the 0-30º and 30-60ºN/S latitude bands for $O_2$-GOP and in the 0-30ºN and 30-60ºS regions for $b_{bp}$-GCP. There are not enough $b_{bp}$ profiles currently available to perform the calculations outside of those regions. In each band, we calculated GPP from diurnal cycles constructed from a random subset of data, repeating calculations 1000 times for subset sizes between 500 and 12,000 profiles. As above, we did not sub-sample the profiles in time, such that our GPP estimates reflect an ensembled median value over the period of 2010-2022. From the resulting GPP estimates, we calculated the 0-100 m integrated quantities, and we derived a signal-to-noise ratio by dividing the standard deviation by the mean value. Unlike Stoer and Fennel (2023), who used a threshold ratio of one, we determined the minimum number of profiles required as the first subset size with a ratio less than 0.5.

Our calculations suggest that between 500 (0-30ºN) and 6500 (30-60ºS) $O_2$ profiles, and between 1100 (0-30ºN) and 4500 (30-60ºS) $b_{bp}$ profiles are required to obtain robust annual GPP estimates from composite diurnal cycles (Fig. 8a,b). Previous estimates are somewhat lower: 20 or 50 $O_2$ profiles per hour (480 or 1200 per day composite day) in tropical and high-latitude waters, respectively (Johnson and Bif, 2021), or 5000 $O_2$ and 2000 $b_{bp}$ profiles south of 30ºS (Stoer and Fennel, 2023). Regardless, these results imply that the horizontal and/or temporal resolution of GPP estimates derived from composite sampling is presently constrained by the number of floats available to attain the requisite number of profiles. While the total number of profiles collected by the BGC-Argo array since 2010 is sufficient to derive decadal $O_2$-GOP, but not $b_{bp}$-GCP, from composite daily cycles in most 10º latitude bands (compare solid lines and shaded region in Fig. 8a,b), more floats will be required to perform similar calculations in narrower latitude bands, or biogeographic provinces. More floats are also necessary to yield GPP estimates with better-than ~10-year temporal resolution.

Notably, our results indicate that the projected array of 1000 BGC-Argo floats (Roemmich et al., 2021; Biogeochemical-Argo Planning Group., 2016) should be sufficient to obtain annual, or better, GPP snapshots at most latitude bands. Assuming, for example, that the projected 1000-float array is deployed evenly in proportion to ocean surface area in each latitude band, and that floats profile every 10.2 days, then the number of profiles obtained per year (dashed black lines in Fig. 8a,b) will be greater than the minimum threshold that we calculated in our bootstrapping analysis at many latitudes. Given these assumptions, there would be enough profiles to obtain sub-annual GPP estimates in regions equatorward of ~30ºN/S (dashed lines in Fig. 8c). More floats will be required towards the poles, although the achievable temporal resolution may still be less than two years in high-latitude Southern Ocean waters. This resolution cannot be achieved if floats are set to cycle at integer intervals (sect. 3.2.1), but, in theory, if all floats are set to profile every 5.2 days (rather than 10.2 days), the duration to achieve the minimum profile threshold should be halved. Given the current BGC-Argo array, on the other hand, the best-available temporal resolution is typically greater than one year at all temperate or sub-polar latitude bands, but may be less than one year in the tropics and sub-tropics (solid lines in Fig. 8c).

It is also noteworthy that our estimates of the minimum number of profiles required for consistent GPP estimates are based on the compilation of $\Delta O_2$ or $b_{bp}$ data obtained during all months of the year. Towards the poles, the amplitude and phase of diurnal productivity or biomass cycles differ between seasons, due, in part, to light constraints on productivity. The diurnal cycles constructed from a composite of measurements obtained throughout the year reflect somewhat conflicting signals from sampling at different times of year, making it more difficult to resolve a clear diurnal signal. As a result, it is likely that our threshold estimates represent an overestimate of the number of profiles required to obtain consistent seasonal GPP values in some regions. Unfortunately, however, there are an insufficient number of profiles presently available in a given season to repeat the analysis at higher temporal resolution.

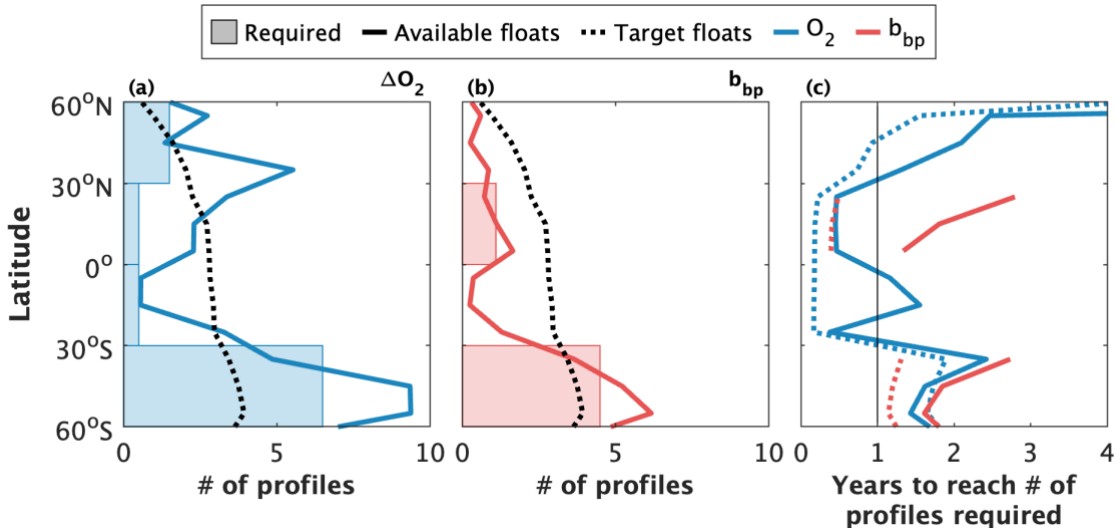

**Figure 8. Estimates of the number of profiles and time required to derive statistically consistent gross oxygen production (GOP) and gross carbon production (GCP) estimates at different latitude bands. The shaded regions in (a) and (b) represent the estimated number of profiles required for dissolved oxygen (O₂) and particle backscattering (b_bp), respectively. The minimum number of profiles required was calculated from a bootstrap analysis with a signal-to-noise threshold of 0.5. The solid lines represent the current number of profiles available for gross primary production calculations since 2010. The dashed lines represent estimates of the number of profiles obtained, per year, by a target biogeochemical-Argo array of 1000 floats deployed ocean-wide proportionally with ocean surface area, and profiling at 10.2-day intervals. Panel (c) shows an estimate of the time required to attain the minimum number of required profiles if the current active (Jan. 2023; solid lines) and target (dashed lines) biogeochemical-Argo array profiles at 10.2-day intervals. The time required was calculated as** $\frac{(profiles\ required)\times(10.2\ days\ per\ profile)}{(current\ or\ target\ \#\ of\ floats\ in\ region)\ x\ (365\ days\ per\ year)}$.

**4.2 Constraints on NCP accuracy and coverage**

The compiled OSP NCP time-series (Fig. 5, sect. 3.1) identified important differences between float-based NCP studies. Those differences can be attributed to one of the following: 1) real, interannual NCP variability, 2) the tracer used to evaluate the NCP budget, or 3) the budget setup and parameterizations. We used the compiled OSP results to assess the potential role of each of those factors on time-resolved and annual-integrated NCP (Fig. 9a). To assess the natural interannual variability, we calculated the mean range of monthly NCP or annual ANCP across studies spanning multiple years (Plant et al., 2016; Yang et al., 2017). To determine the impact of tracer selection, we calculated the mean monthly range of values across studies that performed calculations using more than one tracer (Plant et al., 2016; Huang et al., 2022). To determine the role of the parameterization approach, we calculated the mean monthly range of values across all O₂-based studies (Plant et al., 2016; Yang et al., 2017; Pelland et al., 2018) occurring within the same year.

Our analysis suggests that interannual NCP or ANCP variability is the largest contributor to differences between the float-based OSP NCP studies (Fig. 8a,b). Between-tracer and between-approach differences are similar in magnitude for time-resolved NCP estimates, but between-approach differences are smallest across ANCP estimates. Interannual differences are

largest in the early spring, which may reflect year-to-year differences in the onset of the spring bloom, or the end of wintertime heterotrophy. There are no apparent seasonal patterns in the between-tracer or between-approach differences, although between-tracer differences are somewhat smaller during the summer.

As described in section 2.1, between-tracer differences reflect how the tracers target different components of the carbon pool and system (Huang et al., 2022). Calculations based on $O_2$ and $NO_3^-$ reflect particulate and dissolved organic C cycling, while POC-based calculations only reflect the fraction of suspended POC. DIC or TA budgets, meanwhile, are influenced by organic and inorganic C cycling. Differences between $O_2$ and $NO_3^-$-based estimates, moreover, are sensitive to the relative importance of new production (based on $NO_3^-$) versus recycled production (based on $NH_4^+$), and, to a lesser degree, denitrification and $N_2$-fixation. For example, under fully recycled production, $O_2$-based NCP estimates would reflect $O_2$ production during photosynthesis, while $NO_3^-$ concentrations would be unchanged. As a result, $O_2$-based estimates would exceed $NO_3^-$based values. Similarly, denitrification and $N_2$-fixation would affect the decoupling between $NO_3^-$-based estimates and estimates derived using other tracers if the consumption/production of $NO_3^-$ during those processes is unaccounted for in the NCP budget calculations. Indeed, if the $NO_3^-$ source of $N_2$-fixation is unaccounted for, the resulting NCP estimated will be biased low. This bias is particularly problematic oligotrophic waters (e.g., Huang et al., 2023).

It is possible to partition some of the processes and carbon pools by performing simultaneous NCP calculations using multiple tracers (Huang et al. 2022), but in the absence of such calculations, it is important to consider how the tracer selection influences the interpretation of NCP results. In addition, the between-tracer differences also somewhat reflect the importance of different flux parameterizations used in the budget calculations. For example, calculations based on $O_2$ require estimates of the air-sea flux term, while those based on $NO_3^-$ do not. As a result, those estimates based on $NO_3^-$ may be perceived to be somewhat more accurate, due to the large air-sea flux uncertainties (e.g., Bender et al., 2011; Emerson and Bushinsky, 2016).

The between-method differences reflect differences in the flux parameterizations and NCP budget setup between studies, which are summarized in section 2.2 and Table 1. We examined the contributions of different fluxes to the overall differences between approaches by calculating the range of physical fluxes (air-sea, vertical mixing, entrainment, and vertical advection) applied in the different studies at OSP (Fig. 9c,d). To estimate the range of air-sea fluxes represented in the OSP studies, we calculated monthly average surface water $O_2$ and $O_{2,eq}$ using BGC-Argo observations collected from OSP between 2008 and 2020. We then applied the different $O_2$ air-sea flux parameterization schemes (Table 1) and calculated the resulting range of values. Similarly, we used BGC-Argo observations to the range of vertical fluxes by determining the average monthly subsurface vertical $O_2$ gradient (d[T]/dZ in Eq. 7.6) and concentration difference ($\Delta[T]_z$ in Eqs. 7.7, 7.8), and multiplied those values by the different eddy diffusivity ($\kappa_Z$), vertical advection (u) and entrainment (dh/dt) values applied in the OSP studies. Our analyses indicate that the air-sea flux and vertical mixing fluxes are the most variable across studies, contributing large uncertainty in time-resolved and annual integrated NCP (Fig. 9c,d). Previous work has similarly identified air-sea flux and eddy diffusive mixing as two of the most important sources of uncertainty in their ANCP calculations, up to ~0.7 and 0.3 mol $O_2$ m$^{-2}$ yr$^{-1}$, respectively (Bushinsky and Emerson, 2015; Plant et al., 2016; Yang et al., 2017). Moreover, Plant et al. (2016) estimated an ANCP range of nearly 2 mol $O_2$ m$^{-2}$ yr$^{-1}$ when applying different air-sea flux parametrizations to their calculations,

and a range of ~1 mol $O_2$ m$^{-2}$ yr$^{-1}$ was calculated between ANCP estimates derived using regionally tuned versions of the Liang et al. (2013) air-sea flux model, and an un-tuned version (Plant et al., 2016; Yang et al., 2017).

Another important constraint on the accuracy of float-based NCP estimates is the measurement accuracy of the BGC variable. A ±1% error in $O_2$, for example, can contribute between 0.3 and 2 mol $O_2$ m$^{-2}$ yr$^{-1}$ uncertainty to ANCP estimates (Bushinsky and Emerson, 2015; Yang et al., 2017; Huang et al., 2018), comparable in magnitude to uncertainties resulting from air-sea flux and diffusive mixing. Plant et al. (2016) also found that a ±1% $O_2$ error results in ~±10 mmol $O_2$ m$^{-2}$ d$^{-1}$ error in time-resolved NCP, and, in some cases, causes a shift in the apparent upper ocean metabolic state (i.e., a shift between net heterotrophy and net autotrophy), particularly during the transition seasons. Moreover, $NO_3^-$ budget calculations may be subject to considerable uncertainty in oligotrophic regions when the $NO_3^-$ concentration is close to the sensor's signal-to-noise ratio. In some cases, erroneous float data should preclude NCP calculations altogether (Plant et al., 2016), and, in general, NCP calculations cannot be performed reliably on unadjusted BGC-Argo data. Another potential source of NCP uncertainty resulting from the air-sea flux parameterization is the impact of sea-level pressure on gas solubility. In the diffusive air-sea flux equation described by Eq. 7.3, the term $[T(t,0)]_{eq}$ refers to the gas saturation concentration at ambient sea level pressure ($P_{SLP}$), which can be calculated from empirical solubility algorithms (e.g., Garcia & Gordon, 1992). These algorithms describe the saturation concertation at one atmosphere, yet, the saturation concentration in situ is impacted by $P_{SLP}$, such that $T_{SLP}(P_{SLP})=T_{SLP}(1 \text{ atm.})\frac{P_{SLP}-P_{H2O}}{1 \text{ atm. } -P_{H2O}}$, where P is pressure and $P_{H2O}$ is the pressure due to water vapour. In temperate and high-latitude regions where $P_{SLP}$ is typically lower than one atmosphere, neglecting to account for this effect may lead to an overestimate in the importance of the diffusive air-sea flux term, and a corresponding underestimate in NCP. Such impacts will only be relevant to gas-based budget calculations, and will be most important for those based on $O_2$. Future work should thus endeavour to address this important detail.

Similarly, it is important to note that budget-based NCP calculations assume that the float follows the same water mass over the duration of the calculation period. However, floats may often transition into adjacent water masses, making the interpretation of observed tracer changes somewhat challenging. The resulting uncertainty in NCP calculations may be important, but is difficult to constrain. In some cases, if floats are judged to transition between different water masses (e.g., by assessing water mass temperature and salinity properties), NCP calculations may be precluded altogether.

It is noteworthy that our analysis does not reveal which methods are most accurate. Rather, our analyses were intended to identify sources of variability across NCP studies. Moreover, our case study focused exclusively on OSP, which is well studied with respect to upper ocean mixing fluxes (Cronin et al., 2015), air-sea exchange (e.g., Emerson and Bushinsky, 2016; Emerson et al., 2019; Steiner et al., 2007; Vagle et al., 2010) and NCP. That many other ocean regions are not so well characterized may ultimately limit the current capacity to derive accurate float-based NCP estimates. Future work should thus endeavour to better understand the relative importance and magnitude of the physical fluxes in a variety of ocean regions. In doing so, efforts to tune air-sea flux parameterizations for regional conditions (e.g., Plant et al., 2016; Yang et al., 2017; Emerson et al., 2019; Haskell et al., 2020), or to identify the most accurate parametrization in different basins (e.g., Atamanchuk et al., 2020) should

be undertaken. Approaches like the one employed by Pelland et al. (2018) to evaluate the physical mixing terms from temperature or salinity budget calculations based on in situ profiler data should also be made alongside corresponding NCP budget calculations.

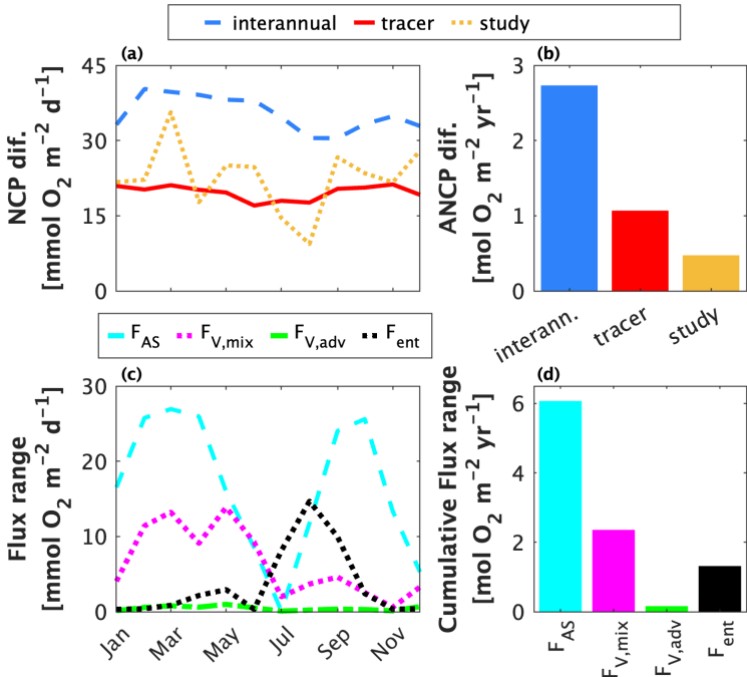

**Figure 9. Contributions to differences between float-based net community production (NCP) and annually integrated NCP estimates at Ocean Station Papa (OSP). Panels (a) and (b) show estimates of the contributions of different factors to differences in published mixed layer-integrated NCP and ANCP estimates from OSP. The blue line/bar represent differences due to real, interannual NCP and ANCP variability; the red line/bar shows differences resulting from the choice of tracer; and the yellow line/bar represents differences between approach occurring between studies within the same year. Panels (c) and (d) represent estimates of the range of**
**dissolved oxygen (O$_2$) flux parameterizations across studies (Table 1). For each flux, values were calculated as the absolute range of estimates after applying the different parameterizations for each term (Table 2). F$_{AS}$ = air-sea flux via diffusion and bubbles; F$_{vmix}$ = vertical transport flux via diapycnal mixing; F$_{vadv}$ = vertical transport flux via advection; F$_{ent}$ = vertical transport flux via entrainment. Panel (d) shows the cumulative flux range over one year.**

**5 Conclusion**

The BGC-Argo fleet offers global observations of real-time ocean biogeochemistry, enabling widespread PP measurements that are independent of, yet complementary to satellite and ship-based approaches. However, compared with PP methods that rely on traditional sampling infrastructure, float-based methods confer significant advantages in detecting PP. Float-based methods, for example, provide simultaneous horizontal, vertical, and temporal PP coverage, presenting the opportunity to fill
key gaps in the existing PP data record (Fig. 1). Moreover, while recent efforts towards FAIR data principles (Tanhua et al.,

2019) have improved the availability of ship and bottle data, resulting PP datasets remain generally inaccessible (e.g., spread over disconnected repositories) and non-standardized (e.g., datasets are often published individually with a single paper/project, and therefore follow no archiving or metadata guidelines). Float data, in contrast, are generally made available within 24 hours of collection, are publicly available and are archived following agreed-upon guidelines (Bittig et al., 2019), enabling cost-effective, open-source PP calculations that can be independently verified and applied by the entire science community, including those without the resources to perform traditional PP methods. Lastly, float-based methods facilitate enhanced detection of the biological response to unpredictable or episodic events like wildfires, volcanic eruptions, or bloom periods, which often cannot be sufficiently characterized using traditional in-situ datasets (Tang et al., 2021).

As float-based techniques mature, the BGC-Argo fleet can be used to extend our current understanding of the marine GPP, NPP, NCP, and C-export, particularly at scales that have so far only been achieved through satellite-based algorithms (e.g., Behrenfeld and Falkowski, 1997; Laws et al., 2011). For example, by compiling the data discussed and derived in this paper, we can calculate independent, global estimates of the carbon export ratio (equivalent to ANCP divided by NPP, where NPP is derived from float-GOP; Figure 10). Notwithstanding the regional and temporal biases in current float-based PP estimates, these C-export ratio estimates are consistent with the commonly used satellite models of Laws et al. (2011) and Henson et al. (2012). Simultaneous estimates of GPP, NCP, and C-export are rarely made, let alone comparisons between them. Thus, the export ratio we derived here could be an important tool for improving our understanding of the ocean carbon cycle. Moving forward, the extent to which float-based PP calculations can be applied will depend, to a large degree, on the availability of float data (sect. 4.1.1), and our capacity to better constrain key sources of uncertainty in biogeochemical budget interpretations (sect. 4.1 and 4.2). Indeed, to increase the availability of float-based PP data, expansion of the Argo fleet should be prioritized, particularly in under-sampled ocean regions. Floats will need to be deployed with sampling intervals set to 5.2 or 10.2 days (rather than 5.0 or 10.0 days) to properly detect diurnal variability. Finally, fully exploiting floats for PP measurements will rely on the open availability of PP datasets, including processed data and relevant software.

Ultimately, continued efforts towards expanding and refining float-based PP datasets will reduce uncertainties in the present methods, yielding widespread, in-situ PP estimates in most ocean basins. As uncertainties are further constrained, the resulting estimates will convey significant tangential benefits, like the ability to improve numerical model predictions through data assimilation (e.g., Wang et al., 2020a) and to train and/or validate satellite PP algorithms, as has been done previously using ship data (e.g., Li and Cassar, 2016; Huang et al., 2021). Given the on-going expansion of the BGC-Argo array and the continued generation of significant amounts of biogeochemical data, the resulting products can be continually re-trained and evaluated using new methods and datasets. Achieving these milestones will enable unprecedented, in situ classification of the response and variability of marine PP to various environmental perturbations over a range of space and time scales.

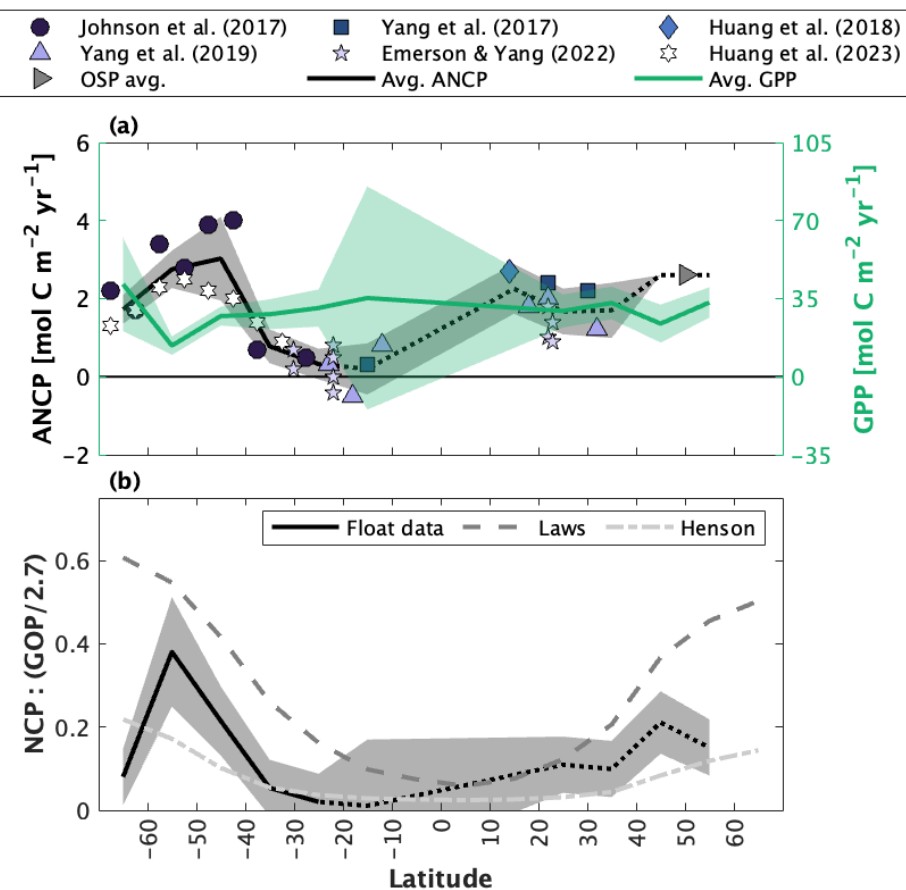

**Figure 10.** The latitudinal distribution of float-derived annual-average gross primary production (GPP), annually-integrated net primary production (ANCP), and the export ratio (equal to GPP divided by ANCP). GPP estimates in (A) are gross oxygen productivity estimates from oxygen measurements ($\Delta O_2$-GOP) integrated within the euphotic zone and converted to carbon equivalents using a photosynthetic quotient value of 1.4. ANCP values are from various data sources, as indicated in the figure legend or from the compilation of Ocean Station Papa data in section 3.1 (Fig. 5). Note that values from Johnson et al. (2017) represent NCP calculated from $NO_3^-$ drawdown over the austral productive period; we did not perform any corrections to adjust those values to represent annually-integrated NCP. The data from Huang et al. (2023) represent the average of $NO_3^-$ and DIC-based ANCP estimates. The black line and shading represent average ± one standard deviation values in 10° latitude bands. In (B), a float-based estimate of the export ratio was derived by dividing average float-based ANCP by float-based net primary productivity (NPP), using an GOP-to-NPP ratio of 2.7, as in Johnson and Bif (2021) and Stoer and Fennel (2023). Independent estimates of the export ratio from Laws et al. (2011) and Henson et al. (2012) are also shown. The dotted black lines north of 30°S indicate poorer latitudinal representation of float-based ANCP, and therefore lower confidence in the derived export ratio.

## Appendix A

Data handling and calculations for the OSP NCP Case Study

We compiled NCP and ANCP data from six published float/glider studies at Ocean Station Papa (OSP) in the Subarctic NE Pacific (Table A1). Time-explicit NCP and ANCP values were obtained from Plant et al. (2016), Yang et al. (2017), Pelland

et al. (2018), Haskell et al. (2020) and Huang et al. (2022). We also obtained an estimate of ANCP from Bushinsky and Emerson (2015). Yang et al. and Haskell et al. provided NCP data that were integrated to the depth of the annual maximum mixed layer (ML), while Plant et al. and Pelland et al. provided depth-resolved estimates. Data from Huang et al. were integrated to 56 m. We present NCP and ANCP values integrated to the annual maximum mixed layer depth (MLD), scaling values from Huang et al. to maximum MLD (i.e., NCP estimates from Huang et al. were scaled by dividing values from that publication by 56 m, and then multiplying by an annual maximum MLD of 120 m for OSP). We appreciate that this approach may result in an over-estimate in maximum MLD-integrated NCP values from Huang et al. (2022) as it assumes constant NCP between 56 m and the maximum MLD, which is likely not the case.

We also obtained NCP estimates from ship-board sampling, moorings, and satellites, collected over the past two decades (Table A1). We obtained two satellite-based NCP estimates: one from a global machine learning algorithm (Li and Cassar, 2016), and the other derived as the product of satellite-NPP (average of the VGPM and CbPM models; Behrenfeld and Falkowski, 1997; Westberry et al., 2008) and a commonly used global export-ratio algorithm (e-ratio; Laws et al., 2011) (i.e., NCP = NPP x e-ratio). The ship- and mooring estimates represent NCP values integrated in the seasonal ML, while satellite-based estimates detect approximately one optical depth below the surface. Accordingly, we scaled all independent NCP estimates to the annual average maximum MLD at OSP, as described above, using MLD estimates obtained from the Argo Mixed Layers climatology (Holte et al., 2017).

We calculated ANCP as the sum of annual maximum MLD-integrated values from January through December for each full year of data. We determined ship-based ANCP by integrating average monthly ML-integrated NCP values over a 12-month cycle, after linearly interpolating values between months without data. All units were converted to $O_2$ equivalents using a PQ value of 1.4, and $O_2:NO_3^-$ ratio of 150:16.

Data handling and calculations for the global GPP Case Study

Following Stoer and Fennel (2023), we compiled all available BGC-Argo $O_2$ and $b_{bp,700}$ data collected between January 2010 and December 2022, selecting only the high-quality (Argo quality flags 1 and 2 representing "good" and "probably good"), "adjusted" (flag 5) and "estimated" (flag 8) $O_2$ data and high-quality $b_{bp}$ data. $b_{bp}$ profiles were de-spiked using a five-point running minimum filter followed by a five-point running maximum filter. Profile measurements were then binned into 10-m intervals from 0 to 200 m depth. We applied linear interpolation between up to two data points when data were missing. We calculated $\Delta O_2$ (mmol m$^{-3}$) using the corresponding float hydrographic data (Garcia and Gordon, 1993, 1992) and POC (mmol m$^{-3}$) following the $b_{bp}$-to-POC algorithm of Graff et al. (2015), after converting $b_{bp,700}$ to $b_{bp,470}$ using a power law relationship with a slope of 0.78 (Boss and Haëntjens, 2016; Boss et al., 2013).

Treating $O_2$ and $b_{bp}$ separately, we excluded a selection of floats with oceanographically inconsistent data or unrealistic $O_2$ and $b_{bp}$ values (see lists in Johnson and Bif, 2021 and Stoer and Fennel, 2023). We discarded any floats that did not sample at least 21 unique hours of the day evenly over their life cycles. Profiles were sub-divided into different spatial groups, representing

10º latitudinal bands (70ºS to 70ºN) or Longhurst Biogeographical Provinces. We constructed a composite diurnal curve in each spatial subset by calculating the median $\Delta O_2$ or $b_{bp}$-POC value at each hour of the day.

We performed two sets of GPP calculations only when at least 21 hours of the day were represented in each subset: 1) using all available $\Delta O_2$ and $b_{bp}$-POC profiles, treating $O_2$ and $b_{bp}$ independently and 2) using co-located data obtained from floats containing both $O_2$ and $b_{bp}$ sensors. GOP and GCP were estimated by fitting the sinusoidal GPP-vs-light function to the resulting diurnal curves. We did not consider the influence of fluxes due to air-sea exchange, vertical mixing, POC sinking or grazing on our calculated GPP estimates. We used each data subset's average location and midpoint date to determine the daily light cycle and sunrise/sunset times. We accounted for DOC production by scaling $b_{bp}$-GPP estimates by a percent extracellular release (PER) value of 0.33, calculated from the global meta-analysis of Moran et al. (2022), and converted GCP values (units mmol C) to $O_2$ equivalents using a photosynthetic quotient of 1.4 (Laws, 1991) (i.e., $\frac{b_{bp}-GCP}{1-0.3}1.4$). Finally, we discarded unrealistic GOP and GCP rates by removing values exceeding three-standard deviations of the mean of a climatological GOP dataset (references listed in Table S1 of the SI). We did not specifically discard negative values, following the recommendation by Barone et al. (2019), but recognize those estimates as representing undetectably low GPP.

**Table A1. List of data sources and archived primary productivity (PP) datasets referenced in the manuscript.**

| Sources | PP type | Platform | Use in manuscript |
|---|---|---|---|
| Bushinsky & Emerson (2015) | ANCP | Float | OSP NCP case study (section 3.1, Fig. 5) |
| Haskell et al. (2020); Huang et al. (2022); Plant et al. (2016); Yang et al. (2017) | NCP, ANCP | Float | OSP NCP case study (section 3.1, Fig. 5) |
| Pelland et al. (2018) | NCP, ANCP | Glider | OSP NCP case study (section 3.1, Fig. 5) |
| Li & Cassar (2016) | NCP | Satellite | OSP NCP case study (section 3.1, Fig. 5) |
| Emerson (2014); Emerson & Stump (2010); Fassbender et al. (2016) | ANCP, NCP | Mooring | Archived PP map (Fig. 1) OSP NCP case study (section 3.1, Fig. 5) |
| Giesbrecht et al. (2012); Hamme et al. (2010); Howard et al., (2010); Izett et al. (2018, 2021); Juranek et al. (2012); Kavanaugh et al. (2014; Lockwood et al. (2012); Palevsky et al., (2016); Timmerman & Hamme (2021) | NCP | Ship | Archived PP map (Fig. 1) OSP NCP case study (section 3.1, Fig. 5) |
| Cynar et al. (2021); Hamme et al. (2012); Izett & Tortell (2021); L. Juranek (2020); Li & Cassar (2016)*; Ouyang et al. (2021); Qin et al., 2021, (2021); Seguro et al. (2019); Wang et al. (2020) | NCP | Ship | Archived PP map (Fig. 1) |
| Johnson (2010); Körtzinger et al. (2008); Weeding & Trull (2014) | NCP, ANCP | Mooring | Archived PP map (Fig. 1) |
| Alkire et al. (2012); Baetge et al. (2020); Huang et al. (2018); Yang (2021); Emerson and Yang, (2022); Yang et al. (2019) | NCP | Float | Archived PP map (Fig. 1) |
| Alkire et al. (2014); Binetti et al. (2020); Haskell et al. (2019); Hull et al. (2021); Possenti et al. (2021) | NCP | Glider | Archived PP map (Fig. 1) |
| Barbieux et al. (2022); Briggs et al. (2018); Gordon et al. (2020); Henderikx Freitas et al. (2020); Johnson & Bif (2021) | GPP | Float | Archived PP map (Fig. 1) |
| Barone et al. (2019); Nicholson et al. (2015) | GPP | Glider | Archived PP map (Fig. 1) |
| Huang et al. (2021)* | GPP | Ship | Archived PP map (Fig. 1) Global GPP case study (section 3.2, Fig. 6) |

OSP = Ocean Station Papa; GPP = gross primary productivity; NCP = net community production; ANCP = annually-integrated NCP; *Data compiled by Li & Cassar (2016) and Huang et al. (2021).

**Table A2. Summary of published float-based GPP and NCP studies. (g) denotes glider-based studies.**

| Method | Variables | PP fraction | Reference |
|--------|-----------|-------------|-----------|
| Diurnal | $O_2$ | GPP | Barone et al. (2019) [g]; Briggs et al. (2018); Gordon et al. (2020); Henderikx Freitas et al. (2020); Nicholson et al. (2015) [g] |
| Diurnal | POC | GPP | Barbieux et al. (2022); Briggs et al. (2018) |
| Diurnal | $O_2$ (composite) | GPP, NPP | Johnson and Bif (2021); Stoer and Fennel (2023) |
| Diurnal | POC (composite) | GPP, NPP | Stoer and Fennel (2023) |
| Budget | $O_2$ | NCP, ANCP | Alkire et al. (2012, 2014) [g]; Binetti et al. (2020) [g]; Bushinsky and Emerson (2015); Haskell et al. (2019) [g]; Huang et al. (2018, 2022); Pelland et al. (2018) [g]; Plant et al. (2016); Possenti et al. (2021) [g]; Yang (2021); Yang et al. (2017, 2018, 2019); Emerson and Yang (2022). |
| Budget | $NO_3^-$ | NCP, ANCP | Haskell et al. (2020); Huang et al. (2022); Plant et al. (2016) |
| Budget | POC, TA, DIC | NCP, ANCP | Huang et al. (2022) |
| Seasonal change | $O_2$, $NO_3^-$, DIC | NCP | Baetge et al. (2020); Hull et al. (2021) [g]; Johnson et al. (2017) |

$O_2$ = dissolved oxygen; POC = particulate organic carbon; $NO_3^-$ = nitrate; DIC = dissolved inorganic carbon; TA = total alkalinity; GPP = gross primary productivity; NCP = net community production; ANCP = annually-integrated NCP; *Data compiled by Li & Cassar (2016) and Huang et al. (2021).

**Table A3. Variations in budget terms used in float- and glider-based NCP calculations.**

| Study | Platform | T | Vertical resolution | $F_{AS} + F_{EP}$ (surface only) | $\kappa_z$ [m² s⁻¹] | w [m s⁻¹] | dh/dt [m d⁻¹] | u + v [m d⁻¹] |
|---|---|---|---|---|---|---|---|---|
| Alkire et al. (2012) | Float | $O_2$ | 1 box (0-27.3 kg m⁻³) | $F_{AS} = k_{O2} (O_2 - O_{2,eq}[1+\Delta_{eq}])$ | 0 | 0 | 0 | 0 |
| Alkire et al. (2014) | Glider | $O_2$ | 1 box (0-27.3 kg m⁻³) | $\Delta_{eq}$ from Woolfe & Thorpe (1991) $k_{O2}$ from Wanninkhof (1992) | $10^{-4}$ | 0 | 0 | estimated from glider displacement between profiles |
| Bushinsky & Emerson (2015)* | Float | $O_2$ | N box MLD-150 m; $\Delta h = 1.5$ m | $F_{AS} = k_{O2} (O_2-O_{2,eq}) + \beta(F_c+F_p)$; $\beta = 1$, 0.29 | Cronin et al. (2015) [surface]; Sun et al. (2013) [profile] | Ekman pumping velocity | derived from observations; > 0 only | NCEP/NCAR reanalysis |
| Plant et al. (2016)* | Float | $O_2$, $NO_3^-$ | N box (0-180 m; $\Delta h = 2$ m) | $F_{AS}=k_{O2} (O_2-O_{2,eq}) + F_{bub}$ | $1.5 \times 10^{-5}$ | PWP | PWP | 0 |
| Yang et al. (2017*, 2018, 2019, 2021); Yang (2021) | Float | $O_2$ | 2 box (0-MLD; MLD- max. MLD) | $F_{AS} = k_{O2} (O_2-O_{2,eq}) + \beta(F_c+F_p)$; $\beta = 1$, 0.53 | $1.5 \times 10^{-5}$ (box 2 only) | 0 | derived from observations; > 0 only | 0 |
| Huang et al. (2018) | Float | $O_2$ | 2 box (0-MLD; MLD- max. MLD) | $F_{AS}$ from Liang et al. (2013) | $10^{-5}$ m² s⁻¹ (box 2 only) | 0 | derived from observations; > 0 only | 0 |
| Pelland et al. (2018)* | Glider | $O_2$ | 86 boxes (0-150 m, $\Delta h = 2$ m; 150-200 m, $\Delta h = 5$m) | $F_{AS}$ from Liang et al. (2013) $F_{AS} = k_{O2} (O_2-O_{2,eq}) + \beta(F_c+F_p)$; $\beta = 0.29$ | T/S budget | T/S budget | 0 | T/S budget |
| Haskell et al. (2019) | Glider | $O_2$ | 2 box (0-MLD, MLD-EuZ) | $F_{AS}$ from Liang et al. (2013) $F_{AS} = k_{O2} (O_{2,eq}) (\Delta O_2/Ar)$ $k_{O2}$ from Nightingale et al. (2000) | Haskell et al. (2016) | Bakun upwelling index | 0 | 0 |

| Study | Platform | T | Vertical resolution | $F_{AS} + F_{EP}$ (surface only) | Kz [m² s⁻¹] | w [m s⁻¹] | dh/dt [m d⁻¹] | u + v [m d⁻¹] |
|---|---|---|---|---|---|---|---|---|
| Binetti et al. (2020) | Glider | $O_2$ | 1 box (0-60m) | $F_{AS} = k_{O2}(O_2 - O_{2,eq}[1+A_{eq}])$ | 0 | 0 | derived from observations; > 0 only | 0 |
| Haskell et al. (2020)* | Float | $NO_3^-$ | 1 box (0-Z, where Z = MLD, EuZ, 100 m or max. MLD) | $F_{EP}=(dS/dt - dS/dt_{phys})(T:S)$; $F_{AS} = k_{O2}(O_2 - O_{2,eq}[1+A_{eq}])$; Δeq from Woolfe & Thorpe (1991) k_O2 from Nightingale et al. (2000) | Cronin et al. (2015); scaled to DIC budget | Ekman pumping velocity | derived from observations; > 0 only | 0 |
| Possenti et al. (2021) | Glider | $O_2$ | 1 box (0-45 m) | $F_{AS} = k_{O2}(O_2 - O_{2,eq}[1+A_{eq}])$; Δeq from Woolfe & Thorpe (1991) k_O2 from Wanninkhof (1992) | 10⁻⁵ | 0 | derived from observations; > 0 only | 0 |
| Huang et al. (2022)* | Float | $O_2$, $NO_3^-$, POC, TIC, TA | 1 box (0-56 m) | $F_{EP} = k_{CO2}(\Delta pCO_2)(K_H)$ + $(dS/dt-dS/dt_{phys})(C:S)$; $F_{AS}=k_{O2}(O_2-O_{2,eq}) + (F_{bub.})$ | Cronin et al. (2015) [surface]; Sun et al. (2013) [profile] | Ekman pumping velocity | derived from observations; > 0 & MLD > 56 m, otherwise 0 | 0 |

| | | | | $k_{CO_2}$ from Wanninkhof (2014) $F_{AS,O_2}$ from Liang et al. (2013) $F_{AS} = k_{O_2}$ $(O_2 - O_{2,eq}[1+\Delta_{eq}])$ | | | | |
|---|---|---|---|---|---|---|---|---|
| Hull et al. (2022) | Float | $O_2$, $NO_3^-$ | 1 box (0-EUZ) | $\Delta_{eq}$ from Liang et al. (2013) $k_{O_2}$ from Nightingale et al. (2000) | 0 | 0 | 0 | 0 |

\* = Studies at OSP; $O_2$ = oxygen concentration (mol m$^{-3}$); POC = particulate organic carbon concentration (mol m$^{-3}$); $NO_3^-$ = nitrate concentration (mol m$^{-3}$); DIC = dissolved inorganic carbon concetration (mol m$^{-3}$); TA = total alkalinity (mol m$^{-3}$); MLD = mixed layer depth; $F_{AS}$ = air-sea gas exchange (mol m$^{-2}$ d$^{-1}$); $F_{bub}$ = air-sea bubble flux (mol m$^{-2}$ d$^{-1}$); $F_{EP}$ = evaporation or precipitation (mol m$^{-2}$ d$^{-1}$); $k_{O_2}$ and $k_{CO_2}$ = air-sea gas transfer coefficient for $O_2$ and $CO_2$; $\kappa_z$ = eddy diffusivity coefficient [m$^2$ d$^{-1}$]; w = vertical advection velocity [m d$^{-1}$]; dh/dt = change in layer depth [m d$^{-1}$]; u + v = horizontal advection velocities [m d$^{-1}$]; $\Delta$h = box vertical displacement (h$_{t+1}$ − h$_t$ in Eq. 7); β = bubble-mediated transfer scaling coefficient [unitless]; PWP = Price-Weller-Pinkel mixed layer model (Price et al. 1986); C:S = observed DIC:salinity ratio [mol C:S]. "Surface" refers to values derived in the mixed layer only, while "profile" is for values derived over the full water column.

**Table A4. A comparison of algorithms for estimating particulate organic carbon (POC, mg m$^{-3}$) from the beam attenuation coefficient ($c_p$, m$^{-1}$) and particle backscattering coefficient ($b_{bp}$, m$^{-1}$) . The wavelength of the $c_p$ and $b_b$ measurements is indicated with a subscripted number (e.g., $c_{p,660}$ indicates measurements at 660 nm). This table is not a complete list; the equations were selected to illustrate variability in POC relationships.**

| POC Equation | Region | Reference |
|---|---|---|
| POC = 367 $c_{p,660}$ + 31.2 | N. Atlantic | Marra et al. (1995) |
| POC = 391 $c_{p,660}$ - 5.8 | N. Atlantic | Cetinić et al. (2012) |
| POC = 35422 $b_{bp,700}$ - 14.4 | N. Atlantic | Cetinić et al. (2012) |
| POC = 48811 $b_{bp,470}$ - 24 | N. and S. Atlantic, Equatorial Pacific | Graff et al. (2015) |
| POC = 841 $b_{bp,532}^{0.395}$ | N. and S. Atlantic | Balch et al. (2010) |
| POC = 39418 $b_{bp,470}$ - 13 | S. Atlantic; Southern Ocean | Thomalla et al. (2017) |
| POC = 501.81 $c_{p,660}$ + 5.33 | Equatorial Pacific | Claustre et al. (1999) |
| POC = 585.2 $c_{p,660}$ + 7.6 | Equatorial Pacific | Behrenfeld and Boss (2006) |
| POC = 661.9 $c_{p,660}$ - 2.168 | Pacific and Atlantic (incl. upwelling) | Stramski et al. (2008) |
| POC = 71002 $b_{bp,555}$ - 5.5 | Pacific and Atlantic (incl. upwelling) | Stramski et al. (2008) |
| POC = 458.3 $c_{p,660}$ + 10.713 | Pacific and Atlantic (excl. upwelling) | Stramski et al. (2008) |
| POC = 53932.4 $b_{bp,555}$ - 5.049 | Pacific and Atlantic (excl. upwelling) | Stramski et al. (2008) |
| POC = 574 $c_{p,555}$ - 7.4 | Mediterranean | Oubelkheir et al. (2005) |
| POC = 404 $c_{p,660}$ + 29.25 | Mediterranean | Loisel et al. (2011) |
| POC = 37550 $b_{bp,555}$ + 1.3 | Mediterranean | Loisel et al. (2011) |
| POC = 31200 $b_{bp,700}$ + 3.04 | Southern Ocean | Johnson et al. (2017) |
| POC = 977760 $b_{bp,770}^{1.166}$ | Southern Ocean | Johnson et al. (2017) |
| POC = 17069 $b_{bp,555}^{0.859}$ | Antarctic Polar Frontal Zone | Stramski et al. (1999) |
| POC = 476935.8 $b_{bp,555}^{1.277}$ | Ross Sea | Stramski et al. (1999) |
| POC = 381 $c_{p,660}$ + 9.4 | Global Ocean | Gardner et al. (2006) |

**Code and Data Availability**

The GPP data in our global case study analysis were derived by modifying the code provided by Stoer and Fennel (2023). The code was only modified to perform GPP calculations in the geographic regions ($10^o$ latitude bands and Longhurst Biogeographic Provinces) described in the main text. Shape files for the Longhurst Biogeographic Province boundaries are available from Flanders Marine Institute (2009). NCP data from our OSP case study analysis is available from Izett et al. (2023). Additional GPP and NCP data included in this manuscript were compiled from the publications listed in Table A1 of the appendix. BGC-Argo data were collected and made freely available by the International Argo Program and the national programs that contribute to it Argo (2023). The Argo Program is part of the Global Ocean Observing System. Float data are available from the Argo Global Data Assembly Centers in Brest, France (ftp://ftp.ifremer.fr/ifremer/argo) and Monterey, USA (ftp://usgodae.org/pub/outgoing/argo).

**Author contributions**

All authors contributed to the planning and preparation of the manuscript. RWI wrote the manuscript, with significant contributions and feedback from all authors. ACS performed the global GPP calculations. RWI compiled the OSP NCP data and performed all analyses on the GPP and NCP data.

**Competing interests**

The authors declare that they have no conflict of interest.

**Acknowledgements**

We would like to thank the BGC-Argo community for supporting the float array through funding acquisitions, deployments, and data management, and for making the resulting data freely available. We also thank many colleagues for making their processed data available for analysis in this paper. In particular, we thank A. Fassbender, W. Haskell, Y. Huang, N. Pelland, J. Plant and B. Yang for their assistance. This work was supported by the Ocean Frontier Institute (OFI) and Canada First Research Excellence Fund (CFREF) through an International Postdoctoral Fellowship to RWI. KF received support from the Natural Sciences and Engineering Research Council of Canada (NSERC) through a Discovery Grant (RGPIN-2014-03938). AS was supported by a Nova Scotia Graduate Student scholarship and by NSERC through a Canada Graduate Scholarship. DN was supported by the National Science Foundation's (NSF) Global Ocean Biogeochemistry Array (GO-BGC) Project under the NSF Award 1946578 with operational support from NSF Award 2110258, as well as support from NSF OCE# 2023080.

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
