# Peer review of "Reviews and syntheses: Expanding the global coverage of gross primary production and net community production measurements using BGC-Argo floats"

_Biogeosciences, 2023_

## Referee Comment (RC2)

Review of "Expanding the global coverage of gross primary production and net community production measurements using BGC-Argo floats" by Izett et al.,

**General comment:** This manuscript aims to provide an overview of available estimates of gross primary production (GPP) and net community production (NCP) obtained from the analysis of BGC-Argo float data. The manuscript starts with a detailed description of the assumptions driving GPP and NCP measurements based on float data, intertwined with a description of some of the existing studies reporting such productivity estimates for different ocean regions (and globally). In the second part of this manuscript (section 3 onwards), the authors further review available NCP estimates for OSP, and conduct their own novel analysis to derive global GPP estimates. Overall, I believe there is a lot of useful information compiled in this work, but the different sections of the manuscript feel disconnected from one another. Moreover, regarding the review of NCP estimates, the authors omit several studies that have inferred respiration rates in the mesopelagic layer from float data and have been used to obtain NCP estimates in the Southern Ocean. My initial recommendation is to divide the present manuscript into two separate works, dealing with GPP and NCP, separately. At this stage, I find the work related on GPP to be potentially more robust than that on NCP. Please see more detailed comments below:

**Specific comments:**

The introductory section is quite complete and provides a good description of the main productivity estimates that one can find in the literature (PP, GPP, NPP, NCP, etc.). For the most part, I like Figure 1, but I do not quite understand the second set of lines below the grey line indicating "autotrophic". I would recommend simplifying panel (a) by removing the last three lines.

Figure 2d does not make sense to me given the large deployment of floats and profiles made available through the SOCCOM program. Based on this panel, it seems as if the Southern Ocean is one of the least sampled regions in terms of BGC-Argo profiles, which is not the case. I have attached below a map from the GO-BGC website showing that the Southern Ocean is the region with the largest quantity of floats (and thus BGC profiles) (https://www.go-bgc.org/array-status#locations). Is Figure 2d perhaps yielding a misleading picture based on the way the data was binned? Furthermore, Figure 2d seems to be inconsistent with Figure 6a, where the largest number of profiles is indeed observed in the Southern Ocean.

[Figure]

Figure from Go-BGC Map Room: https://www.go-bgc.org/array-status#locations

Section 2 provides a good description of the rationale behind the derivation of productivity rates based on tracer budgets and observations. However, community respiration (CR) should be preceded by a negative sign in all the equations where it is present (Eq 1, 2). In most cases, for budgets that include the effect of photosynthesis and respiration on tracers, respiration should always have the opposite sign to that of productivity/photosynthesis, as these chemical redox reactions "flow" in opposite directions.

Near line 115 "…are driven by daytime net autotrophic production (GPP + CR)". Net autotrophic production is NPP (GPP-AR).

Near line 125 "NCP (i.e., GPP + CR)". Should be GPP – CR.

Near line 160 "…derived from particle backscatter (b$_{bp}$) or beam attenuation (c$_p$, typically at 660 nm) measurements (both m$^{-1}$) using regional (e.g., Loisel et al., 2011; Cetinić et al., 2012) or global (e.g., Graff et al., 2015) algorithms.". The Graff et al, 2015 algorithm is not global, it is based on samples from an Atlantic Meridional Transect (AMT-22) and a subsection the Equatorial Pacific.

Section 2.1 is well documented and informative, but it has similar sign-problems in eq 5.3 (loss processes should be negative, as in eq. 5.1), and eq. 6 (again CR should be negative). This same correction applies for Figure 3, under assumptions, "GPP + CR" should be corrected to "GPP – CR".

Section 2.2 for NCP. As this manuscript aims to provide a complete overview of all float-based NCP estimates/methods available, it should also include those applied to the mesopelagic layer, mostly conducted in the Southern Ocean to infer respiration rates, and thereby NCP, from oxygen drawdown:

- Martz, Todd R., Johnson, Kenneth S., Riser, Stephen C., (2008), Ocean metabolism observed with oxygen sensors on profiling floats in the South Pacific, *Limnology and Oceanography*, 53, doi: 10.4319/lo.2008.53.5_part_2.2094.

- Hennon, T. D., Riser, S. C., and Mecking, S. (2016), Profiling float-based observations of net respiration beneath the mixed layer, *Global Biogeochem. Cycles*, 30, 920– 932, doi:10.1002/2016GB005380.
- Arteaga, L. A., Pahlow, M., Bushinsky, S. M., & Sarmiento, J. L. (2019). Nutrient controls on export production in the Southern Ocean. *Global Biogeochemical Cycles*, 33, 942– 956. https://doi.org/10.1029/2019GB006236
- Su, J., Schallenberg, C., Rohr, T., Strutton, P. G., & Phillips, H. E. (2022). New estimates of Southern Ocean annual net community production revealed by BGC-Argo floats. *Geophysical Research Letters*, 49, e2021GL097372. https://doi.org/10.1029/2021GL097372

Section 3 suggest that examples of GPP and NCP will be showed at local and global scales. However, a local/regional example is shown for only NCP, and a global example is shown for only GPP. These GPP and NCP analyses seem therefore disconnected between them and from the previous sections of the manuscript.

Near line 400 " *Float-based NCP studies are somewhat more numerous than GPP studies (Table A2) but are similarly limited in their 400 geographic extent. NCP has been well-studied around Ocean Station Papa (OSP; $50^O$N, $145^O$W) in the subarctic NE Pacific (sect. 3.1.1), and only a handful of localized studies have occurred elsewhere, such as in the S. China Sea (Huang et al., 2018) and the NW Atlantic (Alkire et al., 2014; Yang et al., 2021) (Fig. 2c).* ". This is incorrect, as it omits the Southern Ocean studies mentioned above.

Section 3.1.1. I think this analysis would be better presented in a manuscript dedicated exclusively to NCP or productivity fluxes at OSP. This way, the methodology could be better explained and expanded in a section of its own.

Section 3.2. Again, this section hints at the presentation of global NCP and GPP estimates, but results are presented only for GPP. This type of inconsistency could be addressed by having separate manuscripts on GPP and NCP.

Near line 475: " *No studies to date have estimated global NCP from floats. Johnson et al. (2017) (Southern Ocean), Yang et al. (2019), and 475 Emerson and Yang (2022) (both Subtropical Ocean) have, however, provided extensive assessments of (A)NCP from a compilation of multiple floats. Johnson et al. (2017) used BGC-Argo data to characterize ANCP in the Southern Ocean by compiling $NO_3^-$ data from 24 floats deployed between 2009 and 2016. Similarly, Yang et al. (2019) and Emerson et al. (2022) compiled $O_2$ data from multiple floats to estimate ANCP in the North and South Hemisphere Subtropical Ocean.* ". Again, the studies listed above also used a compilation of floats to infer NCP in large regions of the Southern Ocean and should be referenced here.

Near line 490 "*Our calculations, we extend the work of ..*". This sentence needs correction.

Near line 505 " *There is generally good agreement between float O2- and bbp-based GPP and between the float estimates and independent GOP estimates derived from bottle sampling (Fig. 6b,c)*". Also line 555 "*Float-based GPP estimates have been shown to compare well with independent data, and well between O2- and POC-based estimates (see our global GPP case study, sect. 3.2, also Johnson and Bif, 2021; Stoer and Fennel, 2022).*" I do not agree with these statements. On the contrary, I see a considerably disagreement between the zonally-averaged estimates presented in Figure 6b. From here on, most of the subsequent analyses are based on the premise of an agreement between independent GPP estimates, which is not supported by the presented analysis. The design and focus of the GPP and NCP analyses presented in section 3 do not seem to converge well together within one single manuscript. Therefore, I would strongly recommend having two different works for each topic. Overall, the review and novel analyses conducted with respect to GPP seem to be more mature than those for NCP. Perhaps the authors could consider approaching the topic of float-based GPP estimates first in a more concise manner.

---

## Author Comment (AC1)

Dear Reviewer,

We are grateful for your thoughtful and thorough evaluation of our manuscript. Below, we respond to each of your comments and concerns, identifying how we intend to address them in a revised version of the manuscript. Our responses are indicated with a blue font. We have endeavoured to address all of your comments as you recommended. Overall, we believe that our manuscript will be improved by addressing your comments and we thank you again for your time reviewing this document.

Sincerely,
R. Izett & co-authors.

Specific comments:
Line 44: choose a different word than 'sinks' as not all export is a sinking flux

Thank you for the suggestion. We will change the sentence from "*When measured over sufficiently large temporal and spatial scales, NCP quantifies the amount of photosynthetically produced organic matter that sinks from the upper ocean (Laws 1991)*" to "*When measured over sufficiently large temporal and spatial scales, NCP quantifies the amount of photosynthetically produced organic matter that is removed from the upper ocean (Laws 1991)*"

Equation 1,2,3: it seems counterintuitive to have CR added to GPP in equations 1 & 2 rather than subtracted. I think the reason it is shown this way is because CR is assumed to have a negative value. However, on page 8 it is stated that the first term on the right of equation 5.1 = GPP and the second term on the right = CR, making the relationship GPP – CR, which is inconsistent with equations 1 & 2. It also seems nonintuitive in equations 1 and 3 to have a + sign in front of the last term on the right for equation 1 and a minus sign for the last term of equation 3. Should these both be '+/-' since they represent source and sink terms?

Equation 7.1: should '+/-' be used in front of the final term on the right rather than '-' ?

Thank you for looking closely at the equations. In response to your suggestions, we will change all respiration terms (AR, HR, CR) to have positive notation throughout the manuscript, such that NPP = GPP-AR and NCP = GPP-CR. To clarify this notation, we will change the last sentence of the second introductory paragraph (around L45) from "*All PP fractions are often expressed as volumetric equivalents of organic carbon or O2 production (e.g., mol C or O2 m-3 d-1), such that respiration has negative values*" to "*GPP, NPP and NCP are often expressed as volumetric equivalents of organic carbon or O2 production (e.g., mol C or O2 m-3 d-1), and respiration terms are expressed in terms of organic C or O2 consumption. Accordingly, GPP, NPP and CR can only have positive values, while NCP may assume positive or negative quantities*"

We will also change the "- other source/sinks" term in the equations to "± other sources and sinks"

As a result of these changes, equations 1-3 and 7 will become:

$$d[T(t,z)]/dt = GPP(t,z) - CR(t,z) \pm \text{other sources/sinks}(t,z) \qquad (1)$$
$$d[T(t,z)]/dt \approx GPP(t,z) - CR(t,z) \qquad (2)$$

$$NCP(t,z) = d[T(t,z)]/dt \pm \text{other sources/sinks}(t,z) \qquad (3)$$
$$NCP(t,z) = (h_{i+1}-h_i)[T(t_1,z)]-[T(t_0,z)]\ t_1-t_0 \pm \Sigma F(t,z) \qquad (7)$$

Line 160 – 164: Here it is stated that POC is estimated from published relationships (Loisel et al, Cetinic et al, Graff et al.). I would suggest explicitly giving these relationships in a table in the appendix. The Graff et al. paper, for example, is primarily focused on estimating phytoplankton carbon from bbp and the POC relationship is a secondary result. Explicitly providing the equations used will prevent any confusion.

Thank you for the suggestion. We will include the following supplementary table, and associated references, in the appendix of the manuscript.

**Table A3**. A comparison of selected $c_p$- and $b_{bp}$-to-POC algorithms. Resulting POC units are mg m$^{-3}$. Units of $c_p$ and $b_{bp}$ are both in m$^{-1}$, and the wavelength of the $c_p$ and $b_b$ measurements is indicated with a subscripted number (e.g., $c_{p,660}$ indicates measurements at 660 nm). This table is not a complete list; the equations were selected to illustrate variability in POC relationships.

| POC Equation | Region | Reference |
|---|---|---|
| $POC = 367\ c_{p,660} + 31.2$ | N. Atlantic | Marra et al. (1995) |
| $POC = 391\ c_{p,660} - 5.8$ | N. Atlantic | Cetinić et al. (2012) |
| $POC = 35422\ b_{bp,700} - 14.4$ | N. Atlantic | Cetinić et al. (2012) |
| $POC = 48811\ b_{bp,470} - 24$ | N. and S. Atlantic, Equatorial Pacific | Graff et al. (2015) |
| $POC = 841\ b_{bp,532}^{0.395}$ | N. and S. Atlantic | Balch et al. (2010) |
| $POC = 39418\ b_{bp,470} - 13$ | S. Atlantic; Southern Ocean | Thomalla et al. (2017) |
| $POC = 501.81\ c_{p,660} + 5.33$ | Equatorial Pacific | Claustre et al. (1999) |
| $POC = 585.2\ c_{p,660} + 7.6$ | Equatorial Pacific | Behrenfeld and Boss (2006) |
| $POC = 661.9\ c_{p,660} - 2.168$ | Pacific and Atlantic (incl. upwelling) | Stramski et al. (2008) |
| $POC = 71002\ b_{bp,555} - 5.5$ | Pacific and Atlantic (incl. upwelling) | Stramski et al. (2008) |
| $POC = 458.3\ c_{p,660} + 10.713$ | Pacific and Atlantic (excl. upwelling) | Stramski et al. (2008) |
| $POC = 53932.4\ b_{bp,555} - 5.049$ | Pacific and Atlantic (excl. upwelling) | Stramski et al. (2008) |
| $POC = 574\ c_{p,555} - 7.4$ | Mediterranean | Oubelkheir et al. (2005) |
| $POC = 404\ c_{p,660} + 29.25$ | Mediterranean | Loisel et al. (2011) |
| $POC = 37550\ b_{bp,555} + 1.3$ | Mediterranean | Loisel et al. (2011) |
| $POC = 31200\ b_{bp,700} + 3.04$ | Southern Ocean | Johnson et al. (2017) |
| $POC = 977760\ b_{bp,770}^{1.166}$ | Southern Ocean | Johnson et al. (2017) |
| $POC = 17069\ b_{bp,555}^{0.859}$ | Antarctic Polar Frontal Zone | Stramski et al. (1999) |

| | | |
|---|---|---|
| POC = 476935.8 $b_{bp,555}^{1.277}$ | Ross Sea | Stramski et al. (1999) |
| POC = 381 $c_{p,660}$ + 9.4 | Global Ocean | Gardner et al. (2006) |

Line 493: Check the wording of the sentence beginning 'Our calculation…', something is wrong here

Thanks for pointing this out. We will change the sentence from "*Our calculations, we extend the work of Johnson and Bif (2021) and Stoer and Fennel (2022)*" to "*Our calculations extend the work of Johnson and Bif (2021) and Stoer and Fennel (2022)*"

General: When calculations of production are made where nighttime changes in a given tracer are assumed to be applicable to daytime rates, what error might be introduced because of impacts of diel vertical migrators?

In section 4.1, we discuss potential uncertainty in diurnal-cycle based GPP calculations. In the second paragraph (beginning L572), we discuss uncertainty resulting from non-photosynthetic processes that vary diurnally, such as air-sea gas flux, grazing, and sinking. In response to your feedback, we will modify this paragraph to specifically include grazing and diel vertical migration. A revised paragraph is as follows (new text is underlined, with additional re-phrasing throughout to improve clarity):

*Diurnal cycle GPP methods are based on the presumption that day-night variations in photosynthesis are the primary driver of diurnal variations in $O_2$ or POC concentrations in the upper ocean. Other than accounting for potential diurnal solubility impacts on $O_2$ (through expressing $O_2$ as its concentration anomaly, $\Delta O_2$) no attempts have been made to reconcile for additional diurnal variations in float $O_2$ or POC observations that are not caused by photosynthesis. For $O_2$, these include potential impacts due to air-sea exchange or vertical mixing, and for POC, sinking, diel vertical migration and grazing, or PER. Yet, these processes vary throughout the day, and the extent to which they do depends on the season and region. Diurnal variability in solar heating and wind forcing influence mixed layer dynamics on hourly, or longer, timescales, with impacts on air-sea gas exchange (Briggs et al., 2018; Barone et al., 2019) and near-surface vertical mixing (Price et al., 1986). Moreover, particle sinking, grazing, and DOC production, have been implicated as a mechanism for decoupling $O_2$- and POC-based PP estimates, particularly in high-productivity (e.g., diatom-dominated) regions (e.g., Rosengard et al., 2020). For example, regions of high POC sinking rates, grazing or PER will decouple $O_2$ and POC concentrations, leading to observations of high-$O_2$ and low-POC in upper ocean waters, with implications for resulting GPP and CR estimates (White et al., 2017; Rosengard et al., 2020; Briggs et al., 2018). Similarly, day-night variations in grazing, resulting from diel vertical migrations, could amplify the nighttime decline in POC, thereby artificially inflating nighttime respiration estimates, and decoupling $O_2$- and POC-based GPP calculations. Independently or in combination, these non-photosynthesis diurnal processes likely imprint on the daily signals detected by BGC-Argo floats, whether by single assets or the composite of the array, and therefore constitute a source of uncertainty to the resulting GPP estimates.*

Line 760:  Since the previous statements include assessments of satellites, it is not clear what is implied by stating that float data are 'publicly available' since satellite data are also publicly available.

Our intention in this statement was to compare the availability of float data versus ship/bottle data. While recent efforts towards FAIR data principles have improved the availability of ship/bottle data, they remain less accessible (e.g., spread over multiple, disconnected repositories) and not standardized (e.g., bottle/ship PP datasets are often published individually with a single paper/project, and therefore follow no archiving or metadata guidelines). We will clarify these points in the opening sentences of the conclusions, as in the following revised paragraph (new text is underlined).

*The BGC-Argo fleet offers global observations of real-time ocean biogeochemistry, enabling widespread PP measurements that are independent of, yet complementary to satellite and ship-based approaches. However, compared with PP methods that rely on traditional sampling infrastructure, float-based methods confer significant advantages in detecting PP. Float-based methods, for example, provide simultaneous horizontal, vertical, and temporal PP coverage, presenting the opportunity to fill key gaps in the existing PP data record (Fig. 1). Moreover, while recent efforts towards FAIR data principles (Tanhua et al., 2019) have improved the availability of ship and bottle data, resulting PP datasets remain generally inaccessible (e.g., spread over disconnected repositories) and non-standardized (e.g., datasets are often published individually with a single paper/project, and therefore follow no archiving or metadata guidelines). Float data, in contrast, are generally made available within 24 hours of collection, are publicly available and are archived following agreed-upon guidelines (Bittig et al., 2019), enabling cost-effective, open-source PP calculations that can be independently verified and applied by the entire science community, including those without the resources to perform traditional PP methods. Lastly, float-based methods facilitate enhanced detection of the biological response to unpredictable or episodic events like wildfires, volcanic eruptions, or bloom periods, which often cannot be sufficiently characterized using traditional in-situ datasets (Tang et al., 2021).*

Bittig, H. C., Maurer, T. L., Plant, J. N., Schmechtig, C., Wong, A. P. S., Claustre, H., Trull, T. W., Udaya Bhaskar, T. V. S., Boss, E., Dall'Olmo, G., Organelli, E., Poteau, A., Johnson, K. S., Hanstein, C., Leymarie, E., Le Reste, S., Riser, S. C., Rupan, A. R., Taillandier, V., Thierry, V., and Xing, X.: A BGC-Argo Guide: Planning, Deployment, Data Handling and Usage, Frontiers in Marine Science, 6, https://doi.org/10.3389/fmars.2019.00502, 2019.

Tanhua, T., Pouliquen, S., Hausman, J., O'Brien, K., Bricher, P., de Bruin, T., Buck, J. J. H., Burger, E. F., Carval, T., Casey, K. S., Diggs, S., Giorgetti, A., Glaves, H., Harscoat, V., Kinkade, D., Muelbert, J. H., Novellino, A., Pfeil, B., Pulsifer, P. L., Van de Putte, A., Robinson, E., Schaap, D., Smirnov, A., Smith, N., Snowden, D., Spears, T., Stall, S., Tacoma, M., Thijsse, P., Tronstad, S., Vandenberghe, T., Wengren, M., Wyborn, L., and Zhao, Z.: Ocean FAIR Data Services, Frontiers in Marine Science, 6, https://doi.org/10.3389/fmars.2019.00440, 2019.

Line 775: I'm not sure I would advocate using BioArgo production products to train satellite algorithms as my guess is that there is more error/uncertainty in the former than in the latter. I do not see evidence in the current manuscript to conclusively demonstrate otherwise.

We believe that as float-based PP methods mature - and their uncertainties become reduced or better constrained - it will become feasible to train and validate satellite algorithms using float PP data. The resulting algorithms would constitute an entirely independent method to quantifying PP that does not rely on ship-based observations. Similarly, given the on-going expansion of the BGC-Argo array and the continued generation of significant amounts of biogeochemical data, the algorithms can be continually re-trained and evaluated using new methods and datasets. We will incorporate these comments in the final paragraph of the conclusions as follows (new text is underlined):

*Ultimately, continued efforts towards expanding and refining float-based PP datasets will reduce uncertainties in the present methods, yielding widespread, in-situ PP estimates in most ocean basins. As uncertainties are further constrained, the resulting estimates will convey significant tangential benefits, like the ability to improve numerical model predictions through data assimilation (e.g., Wang et al., 2020a) and to train and/or validate satellite PP algorithms, as has been done previously using ship data (e.g., Li and Cassar, 2016; Huang et al., 2021). Given the on-going expansion of the BGC-Argo array and the continued generation of significant amounts of biogeochemical data, the resulting products can be continually re-trained and evaluated using new methods and datasets. Achieving these milestones will enable unprecedented, in situ classification of the response and variability of marine PP to various environmental perturbations over a range of space and time scales.*

Grammar

Line 223: add 'relationship' after photosynthesis-versus-irradiance

Line 229: Define OSP on first use

Line 348: replace 'are' with 'is'

Line 445:  add 'is' after 'values'

Figure 5: define 'Y17', 'H22' and 'H20'

Line 532: replace 'of' with 'our'

Line 569: add 'between' after 'observed.'

Line 690:  Delete 'And' at the beginning of the sentence and just begin with 'To'

Line 771: add 'be' after 'can'

Thank you for identifying these mistakes. We will make all of these changes as you have suggested, including defining Y17, H22 and H20 in the figure 5 caption.

---

## Author Comment (AC2)

Dear Reviewer,

We are grateful for your thoughtful and thorough evaluation of our manuscript. Below, we respond to each of your comments and concerns, identifying how we intend to address them in a revised version of the manuscript. Our responses are indicated with a blue font. We have endeavoured to address all of your comments following your recommendations. In some cases, however, we feel the recommended changes are unnecessary, and explain why we think so. We believe that our manuscript will be improved by addressing your comments and we thank you again for your time reviewing this document.

Sincerely,
R. Izett & co-authors.

General:

This manuscript aims to provide an overview of available estimates of gross primary production (GPP) and net community production (NCP) obtained from the analysis of BGC-Argo float data. The manuscript starts with a detailed description of the assumptions driving GPP and NCP measurements based on float data, intertwined with a description of some of the existing studies reporting such productivity estimates for different ocean regions (and globally). In the second part of this manuscript (section 3 onwards), the authors further review available NCP estimates for OSP, and conduct their own novel analysis to derive global GPP estimates. Overall, I believe there is a lot of useful information compiled in this work, but the different sections of the manuscript feel disconnected from one another. Moreover, regarding the review of NCP estimates, the authors omit several studies that have inferred respiration rates in the mesopelagic layer from float data and have been used to obtain NCP estimates in the Southern Ocean. My initial recommendation is to divide the present manuscript into two separate works, dealing with GPP and NCP, separately. At this stage, I find the work related on GPP to be potentially more robust than that on NCP.

Thank you for your assessment of our manuscript. We appreciate and value your feedback; however, we elect to not divide the manuscript into separate contributions for GPP and NCP. Our reasons are the following:

- GPP and NCP are important metrics, which have both benefitted from recent efforts to quantify them using float observations. While the methods to estimate both metrics are different, there are some consistencies, such as the requirement to interpret upper ocean biogeochemical budgets (albeit over different timescales) and in the variables used (O2, bbp, in particular). For these reasons, we feel it makes sense to streamline descriptions of their respective methods, as we have done in the opening paragraphs of section 2.

- Despite the recent attention to developing GPP and NCP methods, the approaches are still new, and not widely known or consistently applied. While promising, both sets of methods require continued attention to limit their uncertainties, and promote their uptake by the community. We feel that presenting GPP and NCP methods together is most appropriate for summarizing these shared challenges and efforts to address them.

- NCP and GPP are rarely compared. However, our manuscript offers an early opportunity to do this using float data (as in Fig. 10). While our GPP vs NCP analysis is fairly simple, we feel that alongside the other GPP and NCP examples in the manuscript it offers an example of how the methods can be used as powerful tools - independently *and* together - to resolve PP over a range of scales.

Overall, our goal with this manuscript is to provide a single, comprehensive, and accessible reference on emerging float-based GPP and NCP methods. We are targeting a broad readership, including researchers who do not normally perform PP calculations, with the intention of summarizing the current state of GPP and NCP methods, and helping to familiarize the community-at-large with these new tools. Thus, we elected to describe both methods in a singular resource that serves as both an overview to people unfamiliar with float-based PP methods, and as a resource to those familiar with the methods who may wish to understand, at a higher level, the current main benefits and challenges. Ultimately, we hope that this resource facilitates broader uptake of the methods, as singular or combined tools, and promotes their continued development.

We will clarify this intention by modifying the following sections of text (revised text underlined).

We will add the following text to the end of the abstract: *This paper is intended as a resource to the oceanographic community to facilitate broader uptake of float GPP and NCP methods, as singular or combined tools, and to promote their continued development.*

We will add the following text to the end of the final introduction paragraph: *This paper is intended as a resource for a broad readership — including researchers who do not normally perform PP calculations — that summarizes the current state of GPP and NCP methods and helps to familiarize the community-at-large with the current benefits, challenges and application of these new tools.*

We will modify the second paragraph of the conclusions as follows:
*As float-based techniques mature, the BGC-Argo fleet can be used to extend our current understanding of the marine GPP, NPP, NCP, and C-export, particularly at scales that have so far only been achieved through satellite-based algorithms (e.g., Behrenfeld and Falkowski, 1997; Laws et al., 2011). For example, by compiling the data discussed and derived in this paper, we can calculate independent, global estimates of the carbon export ratio (equivalent to ANCP divided by NPP, where NPP is derived from float-GOP; Figure 10). Notwithstanding the regional and temporal biases in current float-based PP estimates, these C-export ratio estimates are consistent with the commonly used satellite models of Laws et al. (2011) and Henson et al. (2012). Simultaneous estimates of GPP, NCP, and C-export are rarely made, let alone comparisons between them. Thus, the export ratio we derived here could be an important tool for improving our understanding of the ocean carbon cycle. Moving forward, the extent to which float-based PP calculations can be applied will depend, to a large degree, on the availability of float data (sect. 4.1.1), and our capacity to better constrain key sources of uncertainty in biogeochemical budget interpretations (sect. 4.1 and*

*4.2). Indeed, to increase the availability of float-based PP data, expansion of the Argo fleet should be prioritized, particularly in under-sampled ocean regions. Floats will need to be deployed with sampling intervals set to 5.2 or 10.2 days (rather than 5.0 or 10.0 days) to properly detect diurnal variability. Finally, fully exploiting floats for PP measurements will rely on the open availability of PP datasets, including processed data and relevant software.*

Please also see below our responses to your specific comments regarding the omitted Southern Ocean literature. In brief, our intention in this manuscript was to highlight upper ocean (e.g., euphotic zone) processes. However, following your recommendations, we have included references to, and brief descriptions of, the sources that you provide.

Specific comments:

The introductory section is quite complete and provides a good description of the main productivity estimates that one can find in the literature (PP, GPP, NPP, NCP, etc.). For the most part, I like Figure 1, but I do not quite understand the second set of lines below the grey line indicating "autotrophic". I would recommend simplifying panel (a) by removing the last three lines.

Thank you. In Figure 1, our intention was to illustrate a case where CR > GPP, giving rise to net heterotrophic (negative NCP) conditions. The lower part of the figure supports text in the secondary instruction paragraph (starting L45) describing units and notation for the PP fractions. Note that we will change the notation around the respiration terms (see comment below), such that AR, HR and CR have positive notation, and NPP = GPP-AR and NCP = GPP-CR. Accordingly, the text beginning on L45 will become "*GPP, NPP and NCP are often expressed as volumetric equivalents of organic carbon or O2 production (e.g., mol C or O2 m-3 d-1), and the respiration terms are expressed in terms of organic C or O2 consumption. Accordingly, GPP, NPP and CR can only have positive values, while NCP may assume positive or negative quantities*", and equations 1-3 will become:

$$d[T(t,z)]/dt = GPP(t,z) - CR(t,z) \pm \text{other sources/sinks}(t,z) \qquad (1)$$

$$d[T(t,z)]/dt \approx GPP(t,z) - CR(z) \qquad (2)$$

$$NCP(t,z) = d[T(t,z)]/dt \pm \text{other sources/sinks}(t,z) \qquad (3)$$

While we appreciate your feedback, for the reasons described above, we will maintain the lower part of Figure 1, as is. We will, however, clarify in the figure caption what the second set of lines represent. A revised figure caption is as follows, with new text underlined.

[Figure]

**Figure 1. A conceptual schematic of PP definitions. Panel (a) shows simplified reaction equations of organic matter production and respiration. The upper part of the figure represents a region of net autotrophic conditions (NCP > 0), while the lower part represents a region of net heterotrophic conditions (NCP < 0). Note that net heterotrophic conditions do not necessarily always occur deeper in the water column than net autotrophy. Panel (b) represents idealized PP and CR profiles, where PP declines with depth due to the light dependency of photosynthesis. The vertical axis represents water column depth, and the thin black line divides positive and negative rates.**

Figure 2d does not make sense to me given the large deployment of floats and profiles made available through the SOCCOM program. Based on this panel, it seems as if the Southern Ocean is one of the least sampled regions in terms of BGC-Argo profiles, which is not the case. I have attached below a map from the GO-BGC website showing that the Southern Ocean is the region with the largest quantity of floats (and thus BGC profiles) (https://www.go-bgc.org/arraystatus#locations). Is Figure 2d perhaps yielding a misleading picture based on the way the data was binned? Furthermore, Figure 2d seems to be inconsistent with Figure 6a, where the largest number of profiles is indeed observed in the Southern Ocean.

Thanks for this. We agree Fig. 2d is somewhat misleading. As you suggest, this is mostly due to the way that the data were binned; and also due to the colour scale on the original figure. We have re-made Fig. 2d, binning the float profiles to 10x10 degree bins and normalizing the results by the surface area in each grid cell. We also adjusted the colour axis for that panel. Now, we believe that the large number of profiles from the Southern Ocean is better represented, as in Fig. 6a. Moreover, our heat map in Fig. 2d is consistent with a recent snapshot of BGC-Argo profile data obtained from the network status map (see below figure).

Revised Figure 2 and caption (revised text underlined):

[Figure]

**Figure 2. Coverage of GPP and NCP datasets, and BGC-Argo profiles. The upper row represents archived GPP and NCP data obtained from ships or moorings, while panel (c) shows the locations and durations of float- or glider-based GPP and NCP studies. Panel (d) shows a heatmap of the distribution of BGC-Argo profiles collected from 2010 through 2022. Data in panels (a) and (b) were binned to a five-by-five-degree grid. Data in panel (d) were binned to a ten-by-ten degree grid, and normalized by the surface area in each grid cell. A list of archived data sources is provided in the appendix.**

Network status map:

[Figure]

Near line 115 "…are driven by daytime net autotrophic production (GPP + CR)". Net autotrophic production is NPP (GPP-AR).

Near line 125 "NCP (i.e., GPP + CR)". Should be GPP – CR.

Section 2.1 is well documented and informative, but it has similar sign-problems in eq 5.3 (loss processes should be negative, as in eq. 5.1), and eq. 6 (again CR should be negative). This same correction applies for Figure 3, under assumptions, "GPP + CR" should be corrected to "GPP – CR".

Thank you for these comments. As described above, we will change the notation around CR to being negative, so that NCP = GPP-CR, and NPP = GPP-AR. Equation 6 becomes:

$$T(t_1,z) = T(t_0,z) + GPP(z) \int E(t) / \overline{E} dt - CR(z)(t_1-t_0)$$

Near line 160 "…derived from particle backscatter (bbp) or beam attenuation (cp, typically at 660 nm) measurements (both m-1) using regional (e.g., Loisel et al., 2011; Cetinić et al., 2012) or global (e.g., Graff et al., 2015) algorithms.". The Graff et al, 2015 algorithm is not global, it is based on samples from an Atlantic Meridional Transect (AMT-22) and a subsection of the Equatorial Pacific.

Thank you. We will clarify this statement to identify that the Graff et al. (2015) algorithm is based on a latitudinally-distributed dataset obtained from the Atlantic Meridional Transect and the equatorial Pacific.

Section 2.2 for NCP. As this manuscript aims to provide a complete overview of all float-based NCP estimates/methods available, it should also include those applied to the mesopelagic layer, mostly conducted in the Southern Ocean to infer respiration rates, and thereby NCP, from oxygen drawdown:

- Martz, Todd R., Johnson, Kenneth S., Riser, Stephen C., (2008), Ocean metabolism observed with oxygen sensors on profiling floats in the South Pacific, Limnology and Oceanography, 53, doi: 10.4319/lo.2008.53.5_part_2.2094.

- Hennon, T. D., Riser, S. C., and Mecking, S. (2016), Profiling float-based observations of net respiration beneath the mixed layer, Global Biogeochem. Cycles, 30, 920– 932, doi:10.1002/2016GB005380.

- Arteaga, L. A., Pahlow, M., Bushinsky, S. M., & Sarmiento, J. L. (2019). Nutrient controls on export production in the Southern Ocean. Global Biogeochemical Cycles, 33, 942– 956. https://doi.org/10.1029/2019GB006236

- Su, J., Schallenberg, C., Rohr, T., Strutton, P. G., & Phillips, H. E. (2022). New estimates of Southern Ocean annual net community production revealed by BGC-Argo floats. Geophysical Research Letters, 49, e2021GL097372. https://doi.org/10.1029/2021GL097372

We appreciate this feedback. However, mesopelagic respiration is beyond the intended scope of our manuscript, which is meant to focus on float-based PP methods relating to autotrophic production in the upper / euphotic ocean. As we describe in the manuscript's introduction, our intention is to describe the float-based PP methods that are seen as emerging alternatives to

traditional PP approaches which rely on ships or satellites. In this context, a description of processes occurring below the euphotic zone is out of scope for our manuscript.

We will, however, include some additional text in the introduction to identify that we are reviewing GPP/NCP methods for this specific purpose and noting that other recent literature (the references you provide) has presented float-based methods to evaluate NCP/respiration in the deeper water column. A modified final introductory paragraph is as follows, with revised text underlined.

*The primary objective of this paper is to demonstrate the potential of autonomous platforms, exemplified by BGC-Argo floats, for expanding the spatial and temporal coverage of PP estimates* *in the upper ocean**. This paper explores float-based approaches for estimating GPP and NCP, since those methods are more mature than emerging approaches for NPP quantification* *(Arteaga et al., 2022; Yang, 2021; Estapa et al., 2019; Long et al., 2021). While recent literature has presented float-based methods for quantifying PP metrics in the interior ocean (e.g., Martz et al., 2008; Hennon et al., 2016; Arteaga et al., 2019; Su et al., 2022), the focus of this manuscript is on methods that resolve processes occurring principally within the euphotic zone.* *To facilitate a full exploitation of these new opportunities, we take stock of the float-based tools currently available to researchers and identify their strengths and limitations. After providing an overview of the emerging float- and glider-based PP approaches, we present quantitative analyses to demonstrate the current application of these methods,* *as single or combined tools.*

References for the NPP literature cited in the previous paragraph:

Arteaga, L. A., Behrenfeld, M. J., Boss, E., and Westberry, T. K.: Vertical Structure in Phytoplankton Growth and Productivity Inferred From Biogeochemical-Argo Floats and the Carbon-Based Productivity Model, Global Biogeochemical Cycles, 36, e2022GB007389, https://doi.org/10.1029/2022GB007389, 2022.

Estapa, M. L., Feen, M. L., and Breves, E.: Direct Observations of Biological Carbon Export From Profiling Floats in the Subtropical North Atlantic, Global Biogeochemical Cycles, 33, 282–300, https://doi.org/10.1029/2018GB006098, 2019.

Long, J. S., Fassbender, A. J., and Estapa, M. L.: Depth-Resolved Net Primary Production in the Northeast Pacific Ocean: A Comparison of Satellite and Profiling Float Estimates in the Context of Two Marine Heatwaves, Geophysical Research Letters, 48, 1–11, https://doi.org/10.1029/2021GL093462, 2021.

Yang, B.: Seasonal Relationship Between Net Primary and Net Community Production in the Subtropical Gyres: Insights From Satellite and Argo Profiling Float Measurements, Geophysical Research Letters, 48, 1–8, https://doi.org/10.1029/2021GL093837, 2021.

In response to this comment, and a subsequent one, we will also specifically include references to the Arteaga et al. (2019) and Su et al. (2022) papers in the description of published float-based NCP studies. Please see below.

Section 3 suggests that examples of GPP and NCP will be shown at local and global scales. However, a local/regional example is shown for only NCP, and a global example is shown for

only GPP. These GPP and NCP analyses seem therefore disconnected between them and from the previous sections of the manuscript.

Thank you for this evaluation. Around L410 and L474 we attempted to describe why analogous analyses are not yet feasible for both PP metrics. For example, on L409-412, we state that "*To demonstrate the current capacity for float-based PP studies at local scales, we performed a case study analysis of float/glider NCP data from OSP. A similar analysis is not presently feasible for GPP, owing to the small number of localized studies using floats and gliders, and the currently insufficient number of profiles available to conduct GPP calculations from composite diurnal cycles*". Presently, there have not been enough float/glider PP studies in a single region to compile those data and perform an analysis similar to our local NCP analysis. We were also unable to perform our own GPP calculations at very fine spatial scales due to the high number of profiles required to make those calculations. We will clarify these point at the end of L412 by adding the following text:

*Indeed, there have not been enough published float-based GPP studies to date in a single region to compile those data and perform an analysis similar to our local NCP analysis. Moreover, we could not perform our own local GPP calculations due to the high number of profiles required to make those calculations. These factors currently preclude an analogous analysis of GPP methods at localized scales.*

In addition, a coarser NCP case study was not feasible because previous work evaluating NCP at basin scales is quite sparse and disparate - only a few studies (Johnson et al., 2017; Yang et al., 2019; Emerson and Yang, 2022, and the references you included on mesopelagic respiration/ANCP) have used compiled float data to evaluate NCP on coarser scales. Unfortunately, the limited number of studies preclude inter-comparisons of their results. Also, performing new global or basin scale NCP calculations in this manuscript is beyond our scope. To address this apparent inconsistency in our analyses, we will add the follow text to the opening paragraph of section 3.2.1, L482 (new text underlined):

*Building on recent work by Johnson and Bif (2021) and Stoer and Fennel (2022), we performed new global GOP and GCP calculations using the available BGC-Argo array. We summarize those calculations here and provide further details in the appendix. Presently, a similar analysis is not feasible for NCP, as global scale NCP calculations have not yet been attempted by the community, and only a small handful of studies have calculated NCP at basin scales (see section 3.1). As a result, intercomparisons of published results at these scales are not feasible, and new calculations of global NCP are beyond the scope of the present paper.*
*For our GPP calculations, we followed Stoer and Fennel (2022) by compiling all available high-quality BGC-Argo ΔO2 and bbp-POC data collected between January 2010 and December 2022,*

Near line 400 " Float-based NCP studies are somewhat more numerous than GPP studies (Table A2) but are similarly limited in their geographic extent. NCP has been well-studied around Ocean Station Papa (OSP; 50oN, 145oW) in the subarctic NE Pacific (sect. 3.1.1), and only a handful of localized studies have occurred elsewhere, such as in the S. China Sea (Huang et al., 2018) and the NW Atlantic (Alkire et al., 2014; Yang et al., 2021) (Fig. 2c)". This is incorrect, as it omits the Southern Ocean studies mentioned above.

Thank you for this feedback. We will include some of the Southern Ocean citations that you provided by including the following text immediately after the quoted section:

*"Several float-based studies have quantified ACNP in the Southern Ocean, however, that work has principally focused on processes occurring below the euphotic zone (e.g., Martz et al., 2008; Hennon et al., 2016; Arteaga et al., 2019; Su et al., 2022)"*

Near line 475: " No studies to date have estimated global NCP from floats. Johnson et al. (2017) (Southern Ocean), Yang et al. (2019), and 475 Emerson and Yang (2022) (both Subtropical Ocean) have, however, provided extensive assessments of (A)NCP from a compilation of multiple floats. Johnson et al. (2017) used BGC-Argo data to characterize ANCP in the Southern Ocean by compiling NO3- data from 24 floats deployed between 2009 and 2016. Similarly, Yang et al. (2019) and Emerson et al. (2022) compiled O2 data from multiple floats to estimate ANCP in the North and South Hemisphere Subtropical Ocean.". Again, the studies listed above also used a compilation of floats to infer NCP in large regions of the Southern Ocean and should be referenced here.

Thank you. We will include these references by modifying the referenced text as follows (revised text underlined):

*No studies to date have estimated global NCP from floats. Johnson et al. (2017) (Southern Ocean), Yang et al. (2019), and Emerson and Yang (2022) (both Subtropical Ocean) have, however, provided extensive assessments of (A)NCP from a compilation of multiple floats. Johnson et al. (2017) used BGC-Argo data to characterize ANCP in the Southern Ocean by compiling $NO_3^-$ data from 24 floats deployed between 2009 and 2016. Similarly, Yang et al. (2019) and Emerson et al. (2022) compiled $O_2$ data from multiple floats to estimate ANCP in the North and South Hemisphere Subtropical Ocean. Lastly, some recent work (e.g., Martz et al., 2008; Hennon et al., 2016; Arteaga et al., 2019; Su et al., 2022) compiled data from subsets of the Southern Ocean BGC-Argo array to quantify ANCP and respiration below the euphotic zone. Those studies, however, are out of scope for the present manuscript in which we focus on reviewing methods resolving PP metrics primarily within the euphotic zone.*

Section 3.1.1. I think this analysis would be better presented in a manuscript dedicated exclusively to NCP or productivity fluxes at OSP. This way, the methodology could be better explained and expanded in a section of its own.

Thank you. Please see our response to your suggestion to divide the manuscript into separate GPP and NCP papers above. We also feel that the details on the NCP case study provided in Appendix A are sufficient for explaining the methodology employed in that analysis.

Section 3.2. Again, this section hints at the presentation of global NCP and GPP estimates, but results are presented only for GPP. This type of inconsistency could be addressed by having separate manuscripts on GPP and NCP.

Thank you. Again, please see our response to this suggestion above. Please also note our comments above describing why global NCP analyses were not performed in this manuscript.

Near line 490 "Our calculations, we extend the work of ..". This sentence needs correction.

Thanks for catching this error. We will change the sentence from "*Our calculations, we extend the work of Johnson and Bif (2021) and Stoer and Fennel (2022)*" to "*Our calculations extend the work of Johnson and Bif (2021) and Stoer and Fennel (2022)*"

Near line 505 " There is generally good agreement between float O2- and bbp-based GPP and between the float estimates and independent GOP estimates derived from bottle sampling (Fig. 6b,c)". Also line 555 "Float-based GPP estimates have been shown to compare well with independent data, and well between O2- and POC-based estimates (see our global GPP case study, sect. 3.2, also Johnson and Bif, 2021; Stoer and Fennel, 2022)." I do not agree with these statements. On the contrary, I see a considerably disagreement between the zonally-averaged estimates presented in Figure 6b. From here on, most of the subsequent analyses are based on the premise of an agreement between independent GPP estimates, which is not supported by the presented analysis. The design and focus of the GPP and NCP analyses presented in section 3 do not seem to converge well together within one single manuscript. Therefore, I would strongly recommend having two different works for each topic. Overall, the review and novel analyses conducted with respect to GPP seem to be more mature than those for NCP. Perhaps the authors could consider approaching the topic of float-based GPP estimates first in a more concise manner.

Thank you for this feedback. Please note our response to your suggestion to divide the manuscript above.

We agree with your concerns about the comparison between O2 and bbp-based GPP estimates; there are important differences between these different metrics, particularly in the zonally-averaged estimates. To better address the apparent inconsistencies between O2 and bbp GPP estimates, we will revise the text starting on L555 as follows (revised text underlined):

*Float-based GPP estimates have been shown to compare well with independent data, and $O_2$ and $b_{bp}$-based estimates generally correlate with one another (p-value < 0.05 and $R^2$= 0.47 through paired data in upper 60 m; Fig. 7). With some exceptions (e.g., surface waters between 0-30ºN) offsets between $O_2$ and $b_{bp}$-based estimates are often within the standard error of the diurnal cycle approach (Fig. 6, and see results from Johnson and Bif, 2021; Stoer and Fennel, 2022). However, when compared directly, the ratio between $\Delta O_2$-GOP and POC-GCP is not always consistent with the expected relationships based on documented PQ and PER variability (Fig. 7). For example, given an estimated range of ~18-47% DOC production during photosynthesis (median PER value of 32.5% ± 14.4% standard deviation calculated from Moran et al., 2022), and a PQ range of 1-1.45 (Laws, 1991), the ratio between $\Delta O_2$-GOP and POC-GCP uncorrected for PER should be between ~1.2 and 2.6 (shaded region in Fig. 7). Considering an even broader PER range of ~2-50% (global confidence interval from Baines and Pace, 1991) results in an expected GOP:GCP ratio of ~1-2.9. Yet, in our depth-resolved, global GPP dataset, we derived a median ratio of ~3.1 ± 0.2 (median ± confidence interval) for estimates derived in the upper 60 m. When considering all depths (up to 200 m), the median ratio is ~4.1 ± 0.6, reflecting the lower signal-to-noise ratio of diurnal $O_2$ or $b_{bp}$ variability at depth. For comparison, Briggs et al. (2018) calculated a ratio of ~2.6 between mixed layer $O_2$-GOP and $c_p$-GCP during a NW Atlantic spring bloom. These results imply higher PQ values and/or DOC production rates and may indicate*

*that these terms are non-uniform across the global ocean. Using static PQ or PER values in GPP calculations (as in Stoer and Fennel, 2022 and in our global GPP case study) likely contributes to the uncertainty in the resulting GPP datasets, and partially explains the offsets we observed $O_2$- and POC-based GPP estimates, and differences between the float- and bottle sample GPP values. Other sources of uncertainty and causes for potential and apparent offsets between O2- and POC-based estimates are discussed in the following paragraphs.*

Please note that the remainder of section 4.1 describes a number of reasons for potential and apparent offsets between O2 and bbp based GPP estimates, including uncertainty in the bbp-to-POC relationship, diurnal variations in O2 or bbp not related to photosynthesis, and differences in the number and locations O2 and bpp profiles.

---

## Referee Report (RR1)

This manuscript provides a review on the recent progress on how BGC-Argo can be used to advance our understanding of marine productivity, with a special focus on the two key fluxes: gross primary production and net community production. The manuscript started with an explicit introduction about the fundamental concept of these two parameters, followed by a detailed recap on the main approaches used to constrain these two fluxes by using the high-frequency BGC-float observations. Subsequently, an in-depth analysis is performed using a compilation of datasets from prior studies to assess the natural variability of marine productivity, as well as the main uncertainties and challenges that persist in the float-based methodology. Lastly, the authors present a new estimate of the global meridional pattern of carbon export ratios by combining the float estimated GPP and NCP, demonstrating an encouraging agreement when compared with traditional estimates.

Overall, the manuscript is well-written and logically organized, and the figures and tables effectively support the conclusions. I believe this work is of great value and will be of interest to the broad community in the field of marine biogeochemistry, autonomous platforms, sensor technology, and climate. Additionally, it contributes to the ongoing global BGC-Argo project (GO-BGC) and provides valuable guidance for future float deployments.

During the revision, I suggest the authors consider the following comments to improve the clarity of the manuscript:

Figure 1: The PP profile displays the subsurface maximum, in contrast with the light attenuation as described by the authors. I believe this reflects the trade-off between light and nutrient availability, and it would be helpful to provide an explanation in the caption. Furthermore, consider adding an elementary equation describing the organic carbon production somewhere in the figure, as this can provide necessary context regarding why different tracers (O2, NO3, and DIC) are used to track productivity.

Table 1: Ensure that the literature citation format within the table is consistent.

Line 80: Provide information on the global range of GPP and NCP estimates, highlighting the large uncertainties in current estimates, which may exceed the magnitude of air-sea CO2 flux.

Figure 2d: Clarify how the number of BGC-floats was determined. Do you include all floats equipped with at least one chemical or bio-optical sensor? Please clarify this point.

Line 315: Note that DIC, TA, and POC are secondary-derived variables and not directly observed from floats. Describe how these variables are obtained from BGC-float observations to make the paper self-explanatory.

Line 315: Mention that salinity normalization is another commonly used approach to account for the EP term.

Figure 4: Spell out all abbreviations shown in the figures in the caption to enhance readability. This issue should be addressed for all figures throughout the manuscript.

Line 410: Consider adding a short sentence to introduce the background of OSP.

Figure 5: Add a legend to panel a to make the figure easy to interpret. Denote the geographic location of OSP in the caption. Also, consider using the carbon unit in all figures.

Line 585: Subscript "2" alongside the O2.

Figure 7: There appears to be no clear response of the GPP_O2:GPP_bbp ratio to depth. To explore potential geographic patterns. Based on the current knowledge, fractional contribution of DOC to the total carbon production is highly correlated with the NO3 concentration. I would suggest replacing the dot color with the latitude band or background NO3 concentration (i.e., derived from WOA2018) to see if we can derive some geographic pattern.

Line 720: Point out that the tracer budget approach typically assumes the float follows the same water mass, which is not always the case in reality.

Line 735: Change to something like "reflect the fraction of suspended particle organic carbon."

Line 735: I don't quite understand why the relative importance of new production (based on NO3-) versus recycled production can affect the coupling between O2 and NO3-based NCP estimates. The biological term solved from the NO3 budget reflects net production fueled by NO3, aligning with the original definition of NCP.

In contrast, the GPP_O2:GPP_DIC ratio (GPP_O2:GPP_N ratio) may be impacted by the relative importance of new production versus recycled production, as GPP is supported by the bulk inorganic nitrogen (=NO3+NH4), and the C:O ratio (or O:N) differs depending on the nitrogen sources (i.e., C:O=1.1 when the substrate is NH4 and C:O=1.4 when the substrate is NO3; for more details, see Laws et al., 1991, and Huang et al., 2021, GBC).

The extent of denitrification and N2 fixation indeed affects the consistency between O2 and NO3-based NCP estimates. On one hand, denitrification can lead to some degree of decoupling between O2 and NO3-based NCP because it generates NO3 without consuming O2. Regarding the influence of N2 fixation, it depends on whether we account for the external NO3 source inherited from N2 fixation in the NO3 tracer budget. If not, the NO3-based biological term solved from the tracer budget will be biased toward low values. This bias is particularly pronounced in the oligotrophic ocean (see Huang et al., 2023, PNAS). Additionally, it is worth pointing out that budgeting nitrate may be subject to considerable uncertainty in the oligotrophic ocean, as the magnitude of surface NO3 and associated seasonal evolution in this area is typically close to the instrument-to-noise ratio.

It would be also helpful to mention that the reliance on empirical estimate in TA will introduce error in TA-and DIC NCP.

Line 750: I noticed that many prior studies don't account for the effect of in situ sea-level pressure on oxygen solubility, leading to biases, particularly in high-latitude regions where the

sea-level pressure is lower than the standard pressure. Therefore, it is crucial to emphasize this point in the manuscript and call for attention to it in future work.

Line 825: NCP results from Johnson et al., (2017) represent the seasonal maximum of NO3 drawdown during the austral productive period, rather than true ANCP. I wonder if the authors applied any corrections to convert it to aNCP.

Line 835: "We present NCP and ANCP values integrated to the annual maximum mixed layer depth (MLD), scaling values from Huang et al. assuming constant NCP between 56 m and the maximum MLD." and "Accordingly, we scaled all independent NCP estimates to the annual average maximum MLD at OSP". Based on this statement, I still have difficulty understanding how you performed the depth conversion throughout the manuscript.

---

## Author Response (AR2)

**Response to Review #1 Comments**

This manuscript provides a review on the recent progress on how BGC-Argo can be used to advance our understanding of marine productivity, with a special focus on the two key fluxes: gross primary production and net community production. The manuscript started with an explicit introduction about the fundamental concept of these two parameters, followed by a detailed recap on the main approaches used to constrain these two fluxes by using the high-frequency BGC-float observations. Subsequently, an in-depth analysis is performed using a compilation of datasets from prior studies to assess the natural variability of marine productivity, as well as the main uncertainties and challenges that persist in the float-based methodology. Lastly, the authors present a new estimate of the global meridional pattern of carbon export ratios by combining the float estimated GPP and NCP, demonstrating an encouraging agreement when compared with traditional estimates.

Overall, the manuscript is well-written and logically organized, and the figures and tables effectively support the conclusions. I believe this work is of great value and will be of interest to the broad community in the field of marine biogeochemistry, autonomous platforms, sensor technology, and climate. Additionally, it contributes to the ongoing global BGC-Argo project (GO-BGC) and provides valuable guidance for future float deployments.

Thank you for reviewing our manuscript and providing these helpful comments. We have responded to your suggestions below in blue. In our responses, line numbers refer to the revised, unmarked copy of the manuscript.

During the revision, I suggest the authors consider the following comments to improve the clarity of the manuscript:

Figure 1: The PP profile displays the subsurface maximum, in contrast with the light attenuation as described by the authors. I believe this reflects the trade-off between light and nutrient availability, and it would be helpful to provide an explanation in the caption. Furthermore, consider adding an elementary equation describing the organic carbon production somewhere in the figure, as this can provide necessary context regarding why different tracers ($O_2$, $NO_3$, and DIC) are used to track productivity.

Thank you. We added the following equation to the figure, with additional text in the caption as follows: "The equation represents average oceanic aerobic photosynthesis, following Redfield nutrient stoichiometry. The reverse reaction represents respiration."

$$106\ CO_2 + 16\ HNO_3 + H_3PO_4 + 78\ H_2O + \textit{trace elements}$$

$$\downarrow \textit{light}$$

$$(C_{106}H_{175}O_{42}N_{16}P) + 150\ O_2$$

We also added details about why the subsurface maxima exist in the figure caption: "with a subsurface maximum due to photoinhibition".

The new figure and caption are as follows:

[Figure]

**Figure 1.** A conceptual schematic and definitions of the common primary productivity (PP) and respiration (R) metrics: gross primary production (GPP), net primary production (NPP), net community production (NCP), and autotrophic, heterotrophic and community respiration (AR, HR, CR, respectively). Panel (a) shows simplified reaction equations of organic matter production and R. The upper part of the figure represents a region of net autotrophic conditions (NCP > 0), while the lower part represents a region of net heterotrophic conditions (NCP < 0). Panel (b) represents idealized PP and CR profiles, where PP declines with depth due to the light dependency of photosynthesis with a subsurface maximum resulting from photoinhibition. The vertical axis represents water column depth, and the thin black line divides positive and negative rates. The equation represents average oceanic aerobic photosynthesis, following Redfield nutrient stoichiometry. The reverse reaction represents respiration.

Table 1: Ensure that the literature citation format within the table is consistent.

We have altered the citation format in Table 1 to ensure it is now consistent and added a new column to make the citations and the study abbreviations clearer. We also edited Figure 3 and Table A1, A2, and A3 to ensure it has consistent formatting with the rest of the manuscript. Figure 3 is now as follows:

[Figure]

| Sampling Platform | Single profiler, multiple profiles per day | Single profiler, multiple profiles per day | Single profiler, rapidly profiling,; multiple floats, profiling at ~5.2- or 10.2-d intervals |
|---|---|---|---|
| Variables used to date | $O_2$, $b_{bp}$-POC, $c_p$-POC | $b_{bp}$-POC, $c_p$-POC | $O_2$, $b_{bp}$-POC |
| Fit approach (Equation) | Difference between observed noontime $O_2$ or POC and linear regression extrapolation of nighttime data (Eq. 4) | Partial differential equation solved between $SS_0$ and $SR_1$ (CR), and between $SR_0$ and $SR_1$ (GPP + CR) (Eq. 5) | GPP vs light model (P-vs-E, sinusoidal or linear) fit to diurnal curve (Eq. 6) |
| Assumptions | | | $GPP = CR^{(1)}$; $\dfrac{GOP}{GPP} = \dfrac{PQ}{(1-PER)}$ $^{(2)}$ |
| | All approaches: night $d|T(z)|/dt$ = CR; day $d|T(z)|/dt$ = GPP – CR; CR constant over 24-hr | | |
| References | Briggs et al. (2018); Gordon et al. (2020) | Barbieux et al. (2022) | Nicholson et al. (2015); Barone et al. (2019); Henderikx Freitas et al. (2020); Johnson and Bif (2020); Stoer and Fennel, (2022) |

Line 80: Provide information on the global range of GPP and NCP estimates, highlighting the large uncertainties in current estimates, which may exceed the magnitude of air-sea CO2 flux.

We have added details on the uncertainty of GPP and NCP to line 73: "*Ultimately, the challenges associated with quantifying PP from the various in situ and ex situ methods has resulted in large uncertainties in global estimates of GPP and NCP. Reported estimates of GPP, for example, range from 8 to 14 Pmol y$^{-1}$ (Westberry and Behrenfeld, 2013; Huang et al., 2021), while estimates of NCP and carbon export range from 250 to 2650 Tmol y$^{-1}$ (Boyd and Trull, 2007; Henson et al., 2011; Siegel et al., 2016; Westberry et al., 2012).*"

We have added these new citations:

Westberry, T. K., and Behrenfeld, M. J.: Oceanic Net Primary Production, in: Biophysical Applications of Satellite Remote Sensing, edited by: Hanes, J.M., Springer, Berlin, Heidelberg, Germany, 205–230, https://doi.org/10.1007/978-3-642-25047-7_8

Boyd, P.W., Trull, T.W., 2007. Understanding the export of biogenic particles in oceanic waters: is there consensus? Prog. Oceanogr. 72, 276–312. https://doi.org/10.1016/j. pocean.2006.10.007.

Henson, S.A., Sanders, R., Madsen, E., Morris, P.J., Moigne, F.L., Quartly, G.D., 2011. A reduced estimate of the strength of the ocean's biological carbon pump. Geophys. Res. Lett. 38. https://doi.org/10.1029/2011GL046735.

Siegel, D.A., Buesseler, K.O., Behrenfeld, M.J., Benitez-Nelson, C.R., Boss, E., Brzezinski, M.A.,Burd,A.,Carlson,C.A.,D'Asaro, E.A., Doney, S.C., Perry, M.J., Stanley, R.H.R., Steinberg, D.K., 2016. Prediction of the export and fate of global ocean net primary production: the EXPORTS science plan. Front. Mar. Sci. 3. https://doi.org/ 10.3389/fmars.2016.00022

Figure 2d: Clarify how the number of BGC-floats was determined. Do you include all floats equipped with at least one chemical or bio-optical sensor? Please clarify this point.

Thank you. We have clarified in the figure 2 caption that "Only floats equipped with at least one biogeochemical sensor and registered in the international BGC-Argo program were included in (d)". The new figure caption is as follows:

[Figure]

**Figure 2. Coverage of gross primary productivity (GPP) and net community productivity (NCP) datasets, and biogeochemical-Argo (BGC-Argo) profiles. The upper row represents archived GPP and NCP data obtained from ships or moorings, while panel (c) shows the locations and durations of float- or glider-based GPP and NCP studies. Panel (d) shows a heatmap of the distribution of BGC-Argo profiles collected from 2010 through 2022. Data in panels (a) and (b) were binned to a five-by-five-degree grid. Data in panel (d) were binned to a ten-by-ten degree grid, and normalized by the surface area in each grid cell. Only floats equipped with at least one biogeochemical sensor and registered in the international BGC-Argo program were included in (d). A list of archived data sources is provided in the appendix.**

Line 315: Note that DIC, TA, and POC are secondary-derived variables and not directly observed from floats. Describe how these variables are obtained from BGC-float observations to make the paper self-explanatory.

We changed the language on line 310 to remove the word "observations" in the sentence so DIC, TA, and POC are not directly referred to as being observed. We also edited the caption for Figure 4 so that "tracer observations" is now "tracer concentration"; we removed references to "observations" from lines 561, 636 and 643. We also removed the word 'observed' in front of "DIC:salinity ratio" from caption of Table A3.

We describe how DIC, TA, and POC are estimated in the last paragraph of Section 2 "Overview of approaches and application details" (lines 180 to 188). The description for POC reads:

*"POC concentrations (typically mg m$^{-3}$) for GCP and NCP calculations are derived from particle backscatter ($b_{bp}$) or beam attenuation ($c_p$, typically at 660 nm) measurements (both m$^{-1}$) using regional algorithms (e.g., Loisel et al., 2011; Cetinić et al., 2012) or those derived from latitudinally distributed datasets (e.g., Graff et al., 2015 based on data obtained from the Atlantic Meridional Transect and equatorial Pacific) (see Table A4 for a list of selected POC algorithms). Many algorithms estimate POC from $b_{bp}$ at 700 nm ($b_{bp,700}$), the wavelength that is most commonly measured by BGC-Argo floats. For algorithms that rely on different $b_{bp}$ wavelengths (e.g., $b_{bp}$ at 470 nm, as in the algorithm of Graff et al., 2015), a power-law equation is required to convert between $b_{bp,700}$ and $b_{bp}$ at other wavelengths (Boss et al., 2013; Boss and Haëntjens, 2016). Only a subset of floats directly measures $b_{bp,470}$ or $c_{p,660}$."*

Furthermore, to make the explanation for TA and DIC clearer, we changed the final sentences of that section (L188-192) from:

*"Lastly, NCP estimates derived from TA and DIC budgets rely on float pH measurements and an empirical TA function (Huang et al., 2022), where TA is estimated from float $O_2$ and hydrographic observations using a neural network algorithm (e.g., Bittig et al., 2018; Carter et al., 2021). DIC is subsequently calculated from pH and TA based on known seawater carbonate system relationships (Gattuso et al., 2022)."*

to:

*"Lastly, because TA and DIC are not directly measured by BGC-Argo floats, NCP estimates derived using those variables rely on calculations of their concentrations using float measurements and an empirical TA function (Huang et al., 2022). Total alkalinity is estimated from float pH, $O_2$ and hydrographic observations using a neural network algorithm (e.g., Bittig et al., 2018; Carter et al., 2021), and DIC is subsequently calculated from float-pH and derived-TA based on known seawater carbonate system relationships (Gattuso et al., 2022)." – lines 186 to 190)*

Line 315: Mention that salinity normalization is another commonly used approach to account for the EP term.

Thank you for the suggestion. We modified the sentence beginning on line 342 from:

*"T:S is the ratio of tracer T to salinity, $\frac{d[S(t,z)]}{dt}$ is the observed change in salinity over time, and $\frac{d[S(t,z)]}{dt}_{phys}$ is the change due to physical processes"*

to:

*"The evaporation/precipitation term (Eq. 7.5) is typically estimated by normalizing tracer concentrations to the observed salinity during each time step, and multiplying by the measured time-dependent change in salinity (Fassbender et al., 2016; Huang et al., 2022). In Eq. 7.5, T:S is the ratio of tracer T to salinity, $\frac{d[S(t,0)]}{dt}$ is the observed change in salinity over time, and $\frac{d[S(t,0)]}{dt}_{phys}$ is the change due to physical processes"*

Figure 4: Spell out all abbreviations shown in the figures in the caption to enhance readability. This issue should be addressed for all figures throughout the manuscript.

We spelled out the abbreviations in the caption for Figure 4. We also corrected Figures 1-3, Figures 5-10, Table 1, and Tables A1 to A4.

Line 410: Consider adding a short sentence to introduce the background of OSP.

Two sentences were added at the beginning of Section 3.1.1 to add this detail (starting on L448). We changed some sentence structure in the original sentences so that it reads better with these new details.

Figure 5: Add a legend to panel a to make the figure easy to interpret. Denote the geographic location of OSP in the caption. Also, consider using the carbon unit in all figures.

Thank you for your suggestions. However, because the figure is already quite busy, we have elected to not include a legend in panel (a). As described in the figure caption, the colours in (a) correspond with those in panels (b)-(d). Our intention in the figure was to demonstrate the discrepancies between approaches, rather than highlight the magnitude or seasonality of any one study – for this reason, we feel it is unnecessary to add additional details to identify the different NCP time series in (a). Moreover, we have elected to maintain $O_2$-based units for consistency with the rest of the paper, which primarily describes NCP and GOP in $O_2$ equivalents.

We added the geographic location OSP to the caption.

Line 585: Subscript "2" alongside the O2.

We corrected this typographical error with $O_2$ as well as with $b_{bp}$.

Figure 7: There appears to be no clear response of the GPP_O2:GPP_bbp ratio to depth. To explore potential geographic patterns. Based on the current knowledge, fractional contribution of DOC to the total carbon production is highly correlated with the NO3 concentration. I would suggest replacing the dot color with the latitude band or background NO3 concentration (i.e., derived from WOA2018) to see if we can derive some geographic pattern.

Thank you for this suggestion. We agree that it would be interesting to explore geographic patterns in the GOP:GPP relationship. However, our analysis only includes regions (provinces and latitude bands) where the number of floats exceeds the bootstrap threshold for both O2 and $b_{bp}$-based calculations – after we filter out calculations where the bootstrap threshold is not met, only three geographic regions remain. As a result, we are not able to perform an analysis involving latitude or $NO_3^-$ concentration. We have thus elected to keep the figure as-is, but added the following text to line 629:

*"We note that our analysis presented in Fig. 7 is, unfortunately, unable to discern geographic patterns in or predictors of the GOP:GCP relationship due to an insufficient number of floats available for calculations in most geographic regions (see next section). However, future work should use float data to explore potential relationships between the GOP:GCP ratio and $NO_3^-$ concentrations (a predictor of the fractional contribution of DOC-to-total carbon production) or latitude."*

Line 720: Point out that the tracer budget approach typically assumes the float follows the same water mass, which is not always the case in reality.

Thank you for this note – we agree that this is an important distinction. We added the following text as a new paragraph after line 817 (section 4.2):

*"Similarly, it is important to note that budget-based NCP calculations assume that the float follows the same water mass over the duration of the calculation period. However, floats may often transition into adjacent water masses, making the interpretation of observed tracer changes somewhat challenging. The resulting uncertainty in NCP calculations may be important, but is difficult to constrain. In some cases, if floats are judged to transition between different water masses (e.g., by assessing water mass temperature and salinity properties), NCP calculations may be precluded altogether."*

We also added the following text to line 350 to identify this assumption in the description of the NCP method (section 2.2):

*"Importantly, when evaluating NCP following Eq. 7, it is assumed that the float remains in a single water mass, such that tracer changes strictly represent temporal variations due to NCP*

*and the processes described in Eqs. 7.3-7.9. In reality, however, this may not always be the case, and the resulting effect on NCP calculations remains a source of uncertainty that is difficult to constrain."*

Line 735: Change to something like "reflect the fraction of suspended particle organic carbon."

We have changed "reflect the particulate organic fraction" to "reflect the fraction of suspended POC" on line 768.

Line 735: I don't quite understand why the relative importance of new production (based on NO3-) versus recycled production can affect the coupling between O2 and NO3-based NCP estimates. The biological term solved from the NO3 budget reflects net production fueled by NO3, aligning with the original definition of NCP

In contrast, the GPP_O2:GPP_DIC ratio (GPP_O2:GPP_N ratio) may be impacted by the relative importance of new production versus recycled production, as GPP is supported by the bulk inorganic nitrogen (=NO3+NH4), and the C:O ratio (or O:N) differs depending on the nitrogen sources (i.e., C:O=1.1 when the substrate is NH4 and C:O=1.4 when the substrate is NO3; for more details, see Laws et al., 1991, and Huang et al., 2021, GBC).

The extent of denitrification and N2 fixation indeed affects the consistency between O2 and NO3-based NCP estimates. On one hand, denitrification can lead to some degree of decoupling between O2 and NO3-based NCP because it generates NO3 without consuming O2. Regarding the influence of N2 fixation, it depends on whether we account for the external NO3 source inherited from N2 fixation in the NO3 tracer budget. If not, the NO3-based biological term solved from the tracer budget will be biased toward low values. This bias is particularly pronounced in the oligotrophic ocean (see Huang et al., 2023, PNAS). Additionally, it is worth pointing out that budgeting nitrate may be subject to considerable uncertainty in the oligotrophic ocean, as the magnitude of surface NO3 and associated seasonal evolution in this area is typically close to the instrument-to-noise ratio.

It would be also helpful to mention that the reliance on empirical estimate in TA will introduce error in TA-and DIC NCP.

Thanks for the feedback. Regarding the impact of new versus recycled production, $O_2$- and $NO_3^-$-based estimates would be decoupled because $O_2$-based estimates would reflect $O_2$ production during photosynthesis, while $NO_3^-$ would be un-impacted under photosynthesis based entirely on $NH_4^+$. For photosynthesis based on $NO_3^-$, $O_2$- and $NO_3^-$-based estimates should be consistent, within the uncertainty of the $C:O:NO_3^-$ stoichiometry. Similarly, as you have noted, the extent of denitrification and $N_2$-fixation would affect the decoupling between $NO_3^-$-based estimates and estimates based on other tracers due to the consumption/production of $NO_3^-$ during those processes.

We have modified the text, starting on line 769 to explain this line of reasoning:

*"Differences between $O_2$ and $NO_3^-$-based estimates, moreover, are sensitive to the relative importance of new production (based on $NO_3^-$) versus recycled production (based on $NH_4^+$), and, to a lesser degree, denitrification or $N_2$-fixation. For example, under fully recycled production, $O_2$-based NCP estimates would reflect $O_2$ production during photosynthesis, while $NO_3^-$ concentrations would be unchanged. As a result, $O_2$-based estimates would exceed $NO_3^-$-based values. Similarly, denitrification and $N_2$-fixation would affect the decoupling between $NO_3^-$-based estimates and estimates derived using other tracers if the consumption/production of $NO_3^-$ during those processes is unaccounted for in the NCP budget calculations. Indeed, if the $NO_3^-$ source of $N_2$-fixation is unaccounted for, the resulting NCP estimated will be biased low. This bias is particularly problematic oligotrophic waters (e.g., Huang et al., 2023)."*

We agree that it is worth noting the uncertainty in $NO_3^-$ in oligotrophic regions due to measurement noise. We have added the following text on line 804 (new text underlined):

*"...particularly during the transition seasons. Moreover, $NO_3^-$ budget calculations may be subject to considerable uncertainty in oligotrophic regions when the $NO_3^-$ concentration is close to the sensor's signal-to-noise ratio. In some cases, erroneous float data should preclude NCP calculations altogether (Plant et al., 2016), and, in general, NCP calculations cannot be performed reliably on unadjusted BGC-Argo data."*

Line 750: I noticed that many prior studies don't account for the effect of in situ sea-level pressure on oxygen solubility, leading to biases, particularly in high-latitude regions where the sea-level pressure is lower than the standard pressure. Therefore, it is crucial to emphasize this point in the manuscript and call for attention to it in future work.

Thank you for pointing this out. We added the following text on line 807:

*"Another potential source of NCP uncertainty resulting from the air-sea flux parameterization is the impact of sea-level pressure on gas solubility. In the diffusive air-sea flux equation described by Eq. 7.3, the term $[T(t,0)]_{eq}$ refers to the gas saturation concentration at ambient sea level pressure ($P_{SLP}$), which can be calculated from empirical solubility algorithms (e.g., Garcia & Gordon, 1992). These algorithms describe the saturation concertation at one atmosphere, yet, the saturation concentration in situ is impacted by $P_{SLP}$, such that $T_{eq}(P_{SLP}) = T_{eq}(1\,atm.)\frac{P_{SLP}-P_{H2O}}{1\,atm-P_{H2O}}$, where $P$ is pressure and $P_{H2O}$ is the pressure due to water vapour. In temperate and high-latitude regions where $P_{SLP}$ is typically lower than one atmosphere, neglecting to account for this effect may lead to an overestimate in the importance of the diffusive air-sea flux term, and a corresponding underestimate in NCP. Such impacts will only be relevant to gas-based budget calculations, and will be most important for those based on $O_2$. Future work should thus endeavour to address this important detail."*

Line 825: NCP results from Johnson et al., (2017) represent the seasonal maximum of NO3 drawdown during the austral productive period, rather than true ANCP. I wonder if the authors applied any corrections to convert it to a NCP.

Thanks for pointing this out. We did not make a correction and have identified this in the Figure 10 caption. In addition, we included results from Huang et al. (2023) in the analyses presented in Fig. 10 and updated the caption accordingly. Figure 10 is now as follows:

[Figure]

**Figure 10. The latitudinal distribution of float-derived annual-average gross primary production (GPP), annually-integrated net primary production (ANCP), and the export ratio (equal to GPP divided by ANCP). GPP estimates in (A) are gross oxygen productivity estimates from oxygen measurements ($\Delta O_2$-GOP) integrated within the euphotic zone and converted to carbon equivalents using a photosynthetic quotient value of 1.4. ANCP values are from various data sources, as indicated in the figure legend or from the compilation of Ocean Station Papa data in section 3.1 (Fig. 5). Note that values from Johnson et al. (2017) represent NCP calculated from $NO_3^-$ drawdown over the austral productive period; we did not perform any corrections to adjust those values to represent annually-integrated NCP. The data from Huang et al. (2023) represent the average of $NO_3^-$and DIC-based ANCP estimates. The black line and shading represent average ± one standard deviation values in 10° latitude bands. In (B), a float-based estimate of the export ratio was derived by dividing average float-based ANCP by float-based net primary productivity (NPP), using a GOP-to-NPP ratio of 2.7, as in Johnson and Bif (2021) and Stoer and Fennel (2022). Independent estimates of the export ratio from Laws et al. (2011) and Henson et al. (2012) are also shown. The dotted black lines north of 30°S indicate poorer latitudinal representation of float-based ANCP, and therefore lower confidence in the derived export ratio.**

Huang et al. 2023 citation has been added:

Huang, Y., Fassbender, A. J., and Bushinsky, S. M.: Biogenic carbon pool production maintains the Southern Ocean carbon sink, PNAS, 120, 18, https://doi.org/10.1073/pnas.2217909120, 2023.

Line 835: "We present NCP and ANCP values integrated to the annual maximum mixed layer depth (MLD), scaling values from Huang et al. assuming constant NCP between 56 m and the maximum MLD." and "Accordingly, we scaled all independent NCP estimates to the annual average maximum MLD at OSP". Based on this statement, I still have difficulty understanding how you performed the depth conversion throughout the manuscript.

We have clarified what we mean by this by modifying the following text on line 901:

"*We present NCP and ANCP values integrated to the annual maximum mixed layer depth (MLD), scaling values from Huang et al. to maximum MLD (i.e., NCP estimates from Huang et al. were scaled by dividing values from that publication by 56 m, and then multiplying by an annual maximum MLD of 120 m for OSP). We appreciate that this approach may result in an over-estimate in maximum MLD-integrated NCP values from Huang et al. (2022) as it assumes constant NCP between 56 m and the maximum MLD, which is likely not the case.*"

We also modified the text on line 911 as:

"*Accordingly, we scaled all independent NCP estimates to the annual average maximum MLD at OSP, as described above, using MLD estimates obtained from the Argo Mixed Layers climatology (Holte et al., 2017).*"

In addition to your comments above, we made additional corrections or clarifications to the text as follows.

We have modified the text starting on line 521:

*"Two Southern Ocean studies (Johnson et al., 2017; Huang et al. 2023) and two subtropical ocean studies (Yang et al. 2019; Emerson and Yang, 2022) have, however, provided extensive assessments of (A)NCP from a compilation of multiple floats. Johnson et al. (2017) used BGC-Argo data to characterize ANCP in the Southern Ocean by compiling $NO_3^-$ data from 24 floats deployed between 2009 and 2016. Similarly, Huang et al. (2023) provided basin-scale estimates of NCP in different biogenic carbon pools in the Southern Ocean, derived using a compilation of floats and multiple tracers (DIC, TA, $NO_3^-$, POC)."*

We have also used consistent terms for $b_{bp}$-GCP (rather than POC-GCP on line 615 and 618) in Section 4.1.

We added more to the foot note in Table 1 to explain what 'surface' and profile' means.

We changed $O_2$/GOP and $b_{bp}$/GPP in the legend of Fig. 6d to $\Delta O_2$-GOP and $b_{bp}$-GCP, respectively.

The x-axis label in Figure 7 was changed from "particulate $b_{bp}$-GCP" to "$b_{bp}$-GCP".

In the caption of Figure 10, "NPP-to-GOP ratio" was corrected to "GOP-to-NPP ratio".

We have also updated Stoer and Fennel (2022) to Stoer and Fennel (2023) (in accordance with the publication date)

We have also corrected the Data Availability Statement:

1) To give the appropriate links to the raw float data starting on line 970:

"BGC-Argo data were collected and made freely available by the International Argo Program and the national programs that contribute to it Argo (2023). The Argo Program is part of the Global Ocean Observing System. Float data are available from the Argo Global Data Assembly Centers in Brest, France (ftp://ftp.ifremer.fr/ifremer/argo) and Monterey, USA (ftp://usgodae.org/pub/outgoing/argo)."

2) To reference the Zenodo data repositories of Stoer and Fennel (2022) and Izett et al. (2023), which have also been added to the citation list.

Stoer, A. C. and Fennel, K.: Processing and Data for "Estimating ocean net primary productivity from daily cycles of carbon biomass measured by profiling floats", Zenodo [dataset], https://doi.org/10.5281/zenodo.6977161, 2022.

Izett, R. W., Haskell, W., Huang, Y., Pelland, N., Plant, J., and Yang, B.: An archive of net community production estimates derived from autonomous profiler observations at Ocean Station Papa [Dataset], Zenodo, https://doi.org/10.5281/zenodo.7667521, 2023.